# NODULIN HOMEOBOX is required for heterochromatin homeostasis in *Arabidopsis*

Zsolt Karányi[1,2,8], Ágnes Mosolygó-L[1,8], Orsolya Feró[1], Adrienn Horváth[1], Beáta Boros-Oláh[1,3,4], Éva Nagy[1], Szabolcs Hetey[1], Imre Holb[5], Henrik Mihály Szaker [6,7], Márton Miskei[1], Tibor Csorba [6] ✉ & Lóránt Székvölgyi [1,3] ✉

*Arabidopsis* NODULIN HOMEOBOX (NDX) is a nuclear protein described as a regulator of specific euchromatic genes within transcriptionally active chromosome arms. Here we show that NDX is primarily a heterochromatin regulator that functions in pericentromeric regions to control siRNA production and non-CG methylation. Most NDX binding sites coincide with pericentromeric het-siRNA loci that mediate transposon silencing, and are antagonistic with R-loop structures that are prevalent in euchromatic chromosomal arms. Inactivation of NDX leads to differential siRNA accumulation and DNA methylation, of which CHH/CHG hypomethylation colocalizes with NDX binding sites. Hi-C analysis shows significant chromatin structural changes in the *ndx* mutant, with decreased intrachromosomal interactions at pericentromeres where NDX is enriched in wild-type plants, and increased interchromosomal contacts between KNOT-forming regions, similar to those observed in DNA methylation mutants. We conclude that NDX is a key regulator of heterochromatin that is functionally coupled to het-siRNA loci and non-CG DNA methylation pathways.

Functionally, the eukaryotic genome is divided into transcriptionally active euchromatin and silent heterochromatin (so-called 'open' and 'closed' chromatin). In *Arabidopsis*, euchromatin is located along chromosome arms containing most transcriptionally active genes, whose expression depends on tissue/cell type, developmental phase, and environmental conditions. Heterochromatin is depleted in protein coding genes and typically associates with telomeres, (peri)centromeres, transposable elements, silenced rDNA, and small heterochromatic islands interspersed along euchromatic arms. Key functions of heterochromatin include: (i) protection of the genome against unscheduled transposon activities that would lead to genomic instability, (ii) regulation of cell differentiation and cell-type identity by hindering transcription factor-mediated cellular reprogramming, (iii) controlling mitotic cell division by ensuring kinetochore assembly at pericentromeres and sister chromatin cohesion.

Several molecular pathways interact to regulate heterochromatin status, including DNA methylation, histone modifications, and RNA silencing[1,2]. DNA methylation in plants occurs in three sequence contexts (CG, CHG, CHH) driven by different genetic pathways[3,4]. MET1 is responsible for CG methylation, the chromomethylases CMT3 and, to lesser extent, CMT2 catalyze CHG methylation, while DRM1/2 and CMT2 perform CHH methylation. De novo methylation of CHG/CHHs

[1]MTA-DE Momentum, Genome Architecture and Recombination Research Group, Department of Biochemistry and Molecular Biology, Faculty of Medicine, University of Debrecen, Debrecen H-4032, Hungary. [2]Department of Internal Medicine, Faculty of Medicine, University of Debrecen, H-4032 Debrecen, Hungary. [3]Department of Pharmaceutical Technology, Faculty of Pharmacy, University of Debrecen, H-4032 Debrecen, Hungary. [4]Doctoral School of Molecular Cell and Immune Biology, University of Debrecen, H-4032 Debrecen, Hungary. [5]Faculty of Agronomy, University of Debrecen, Böszörményi út 138, 4032 Debrecen, Hungary. [6]MATE University, Genetics and Biotechnology Institute, Gödöllő Pest H-2100, Hungary. [7]Institute of Plant Biology, Biological Research Centre, Szeged H-6726, Hungary. [8]These authors contributed equally: Zsolt Karányi, Ágnes Mosolygó-L. ✉e-mail: Csorba.Tibor.Levente@uni-mate.hu; lorantsz@med.unideb.hu

is established by the RNA-dependent DNA methylation pathway (RdDM) involving the generation of specific 24nt heterochromatic small interfering RNAs (het-siRNA)[1,2]. In the upstream phase of RdDM, precursor transcripts are produced by RNA Polymerase IV (Pol IV) that mature into functional het-siRNAs by RNA-dependent RNA polymerase 2 (RDR2) and DICER-Like 3 (DCL3) endoribonuclease[5,6]. In the downstream phase, het-siRNAs are incorporated into the central component of the RNA-Induced Transcriptional Silencing Complex (RITSC), ARGONAUTE4 (AGO4) or its homologs (AGO6, AGO9). RITSC is then recruited to target loci by scaffold transcripts produced by another plant-specific RNA polymerase, Pol V, which ultimately tethers the DRM2 methylase to perform de novo cytosine methylation[5,7].

In *Arabidopsis*, CHG and CHH methylation is mainly associated with heterochromatin and is functionally linked to histone H3 lysine 9 methylation. CHG/CHH methylation reinforces H3K9me1/2 and vice versa: CMT3/CMT2 DNA methylases read the H3K9me2 mark, while SUVH4/5/6 histone methylases (see below) read the mCHG/mCHH signal[8,9]. This mechanical coupling creates a self-amplifying loop between CHH/CHG methylation and histone methylation to silence transposable elements. H3K9me1/2 is catalyzed by KYP/SUVH4, SUVH5, and SUVH6 lysine-methyltransferases, which recognize mCHG/mCHH sites in constitutive heterochromatin[10,11]. (The abundance of H3K9me3 modification is low in *Arabidopsis*.) The H3K9me1/2 signal is recognized by specific reader proteins, of which AGDP1 (also known as ADCP1) has only recently been identified[12,13] and further members await exploration.

Another crucial repressive histone mark is H3 lysine 27 methylation (H3K27me) that occurs independently of DNA methylation pathways[14]. This suggests that the two repressive histone modifying systems act autonomously on heterochromatin formation. H3K27me1 is typically located in chromocenters and deposited by ATXR5/6, while H3K27me2/3 form heterochromatic patches in chromosome arm-associated gene loci, catalyzed by the Polycomb Repressive Complex 2 (PRC2). Polycomb Group (PcG) proteins also form the Polycomb Repressive Complex 1 (PRC1) that has H3K27me3-reader and histone H2A E3 ubiquitin ligase activities[15] and mediates gene repression through histone H2A lysine 119 mono-ubiquitination (H2AK119ub1). Several factors have emerged in recruiting PRC1 to chromatin, such as the PRC2 component CLF[16], the CAF-1 histone chaperone[17], the AL6 PhD finger protein[18], the LHP1 chromodomain protein[16], and more recently, the homeodomain protein NDX (NODULIN HOMEOBOX), which is the only transcription factor to date that interacts with PRC1[19].

NDX is a nuclear transcription factor that appeared early in evolution and is present in vascular plants[20]. It contains an atypical homeobox domain (HD) and two additional NDX-A/B domains, of which the HD and NDX-B domains were shown to bind double-stranded and single-stranded DNA in vitro[19,21,22]. The specific role of these domains is not known in detail. NDX has been shown to associate with the PRC1 core components RING1A/B, making a functional link with the E3 ubiquitin ligase module of the complex[19]. Previously, NDX was shown to regulate the expression of *FLOWERING LOCUS C* (*FLC*), a central integrator of flowering transition, by stabilizing an R-loop structure at the 3′-end of the locus[21]. This region serves as a terminator of sense (*FLC*) transcription and a promoter of antisense (*COOLAIR*) transcription, the latter being inhibited by an unknown mechanism that suppresses *FLC* transcription and accelerates flowering transition. A recent study identified FCA (an RNA binding protein) and FY (a component of the cleavage polyadenylation specificity factor complex) to recognize and resolve the R-loop in *FLC* and a subset of other loci[23,24]. However, the role of NDX in these processes was not studied in a mechanistic detail and was omitted from the working model of co-transcriptional chromatin silencing by R-loop resolution[23]. The genome-wide association of NDX with R-loops and its global effect on ncRNA transcription remains a fundamental question that needs to be addressed. In addition, NDX may also affect gene expression by

modifying spatial chromatin organization, as enhanced DNA looping interactions were observed at *FLC* in an *ndx* mutant[21]. A similar mechanism has been hypothesized on the regulation of *ABI4* through spatial interactions between NDX and PRC1[19].

The possibility of NDX-mediated genome structural changes seems realistic given that in animals PRC1/RING1B (i) causes chromatin compaction through its non-enzymatic function[25,26], (ii) mediates long-range promoter-promoter interactions between developmentally regulated genes[27], and (iii) orchestrates estrogen-induced enhancer-promoter looping interactions[28]. In *Drosophila*, the NDX-related zeste protein[22] was shown to be required for long-range communication between promoters and for insulator bypass during gene activation[29,30]. In addition, a recent study in *Arabidopsis* showed that removal of the PRC1 component LHP1 from chromatin (through *APOLO* lncRNA-mediated R-loop formation) induces 3D chromatin conformational changes[31].

Herein, we characterized the global role of NDX in chromatin-based gene regulatory processes using cytological, molecular, and high-throughput approaches. Our results show that NDX is a key regulator of heterochromatin accessibility and chromatin packing in pericentromeric regions that are functionally coupled to non-CG methylation pathways.

## Results

### NDX is a heterochromatin-associated factor

To understand the genome-wide regulatory roles of NDX, we first analyzed its genomic distribution. For this, we performed chromatin immunoprecipitation sequencing (ChIP-seq) in 10-day old *Arabidopsis* seedlings expressing N-terminally and C-terminally tagged NDX fusion proteins (flag-NDX/*ndx1-1*(FRI)/*flc-2* and NDX-GFP/*ndx1-1*(FRI)/*flc-2*, respectively) expressed from their endogenous promoter[21,32]. The two ChIP-seq profiles were highly correlated (Pearson $r > 0.85$, Supplementary Fig. 1), and 2243 flag-NDX and 583 NDX-GFP binding sites were identified (Fig. 1a) showing a statistically significant overlap compared to random overlaps ($p < 0.0001$, prop.test). Enrichment of NDX was validated by ChIP-PCR at selected genomic loci (Fig. 1b). Importantly, binding of NDX to both the *ABI3* and *ABI4* downstream regions[19] and the *FLC* 3′-terminator[21] was also confirmed (Fig. 1c), indicating that our analysis is robust and reliable. We then visualized the density of ChIP peaks along the entire length of *Arabidopsis* chromosomes and revealed a significant colocalization of NDX with centromeric and pericentromeric regions (Fig. 1d), in marked contrast to the five chromosomal arms and telomeres, where the density of peaks was much lower (16 peaks/Mb in arms *vs.* 84 peaks/Mb in pericentromeres). This non-random distribution suggests that NDX is mainly associated with gene-poor heterochromatic regions and is rare in transcriptionally active euchromatin.

To confirm the genomic binding of NDX by a sequencing-independent approach, we applied confocal laser scanning microscopy (CLSM) coupled to fluorescence correlation spectroscopy (FCS) and fluorescence recovery after photobleaching (FRAP). CLSM performed in live root tips (expressing NDX-GFP) showed strong green fluorescence near the nuclear periphery (typically associated with perinuclear heterochromatin) and in the nucleolus (Fig. 2a and Supplementary Fig. 2). Nucleolar staining was visible in <20% of cells and it is currently unclear whether this fraction represents a functional or non-functional population that acts as a "storage depot". In contrast, peripheral nuclear localization was consistently observed and was further reinforced by the overlap between NDX binding sites and NUCLEOPORIN1 (NUP1)-enriched chromatin[33], which marks nuclear periphery (Supplementary Fig. 3). Of the NUP1 peaks classified as pericentromeric and arm-associated, only the former group showed significant colocalization with NDX. Pericentromeric enrichment was confirmed microscopically in a subset of cells that showed a typical chromocenter structure (Fig. 2b), which represents a small fraction of

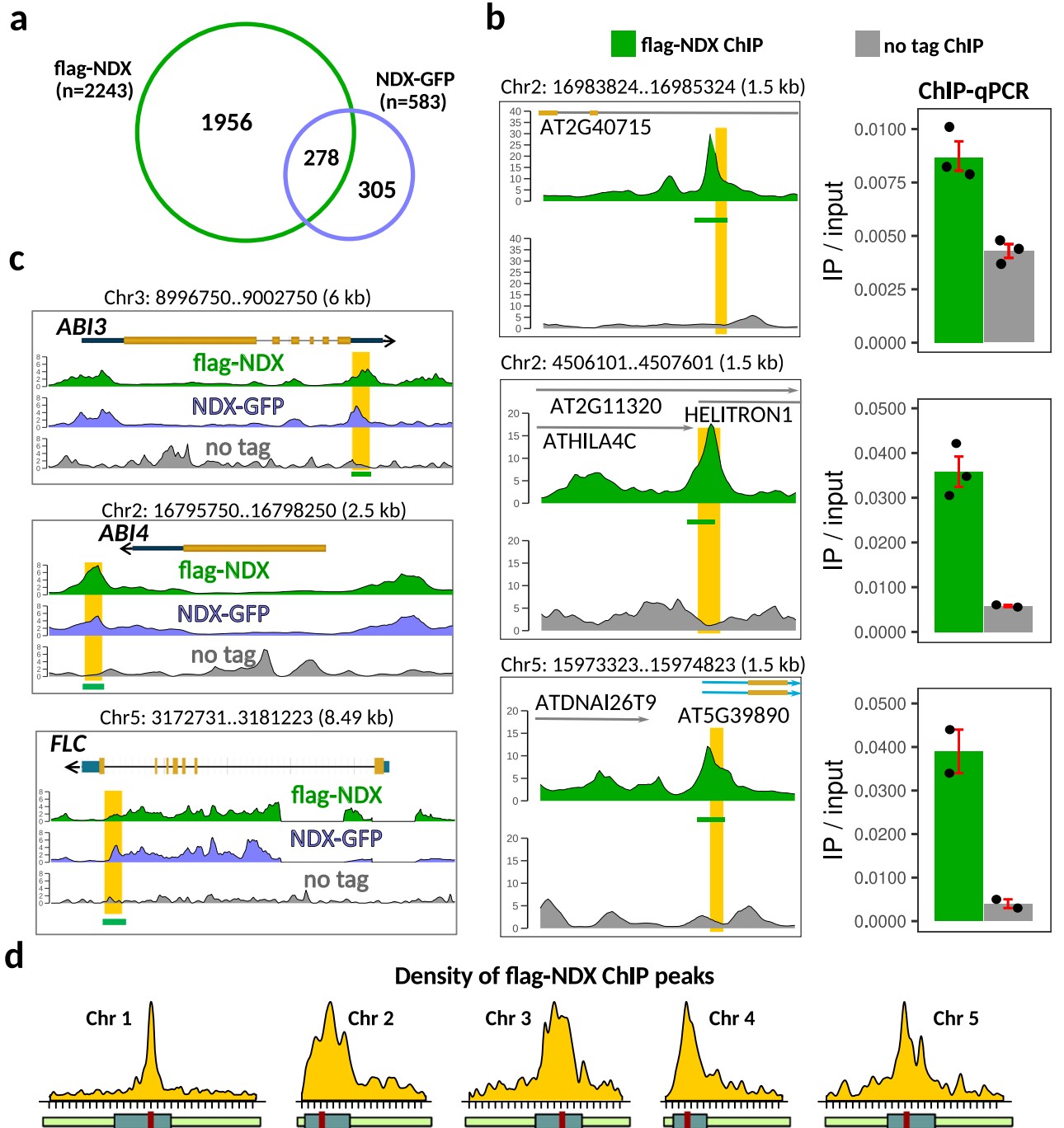

**Fig. 1 | Genome-wide mapping of chromosomal binding sites of NODULIN HOMEOBOX (NDX). a** Overlap of ChIP-seq peaks identified in N-terminally and C-terminally tagged flag-NDX and NDX-GFP lines, respectively. The proportion of common peaks is significantly higher than expected from a computer randomized distribution (***$p < 0.0001$, prop.test, two-sided; Bonferroni correction). **b** ChIP-qPCR validation of representative flag-NDX binding sites from different chromosomes. Genomic positions are indicated on the top of each panel. Specific ChIP signal and background signal ("no tag") is shown in green and black, respectively. Positions of qPCR amplicons are highlighted in yellow. Error bar: SEM. Sample size $n = 3$ biologically independent replicates. **c** Genome browser tracks showing the distribution of flag-NDX and GFP-NDX ChIP signals over *ABI3*, *ABI4*, and *FLC* loci that were previously shown by qPCR to bind NDX. Positions of qPCR amplicons are highlighted in yellow. Sample size $n = 2$ biologically independent samples for each track. **d** Density of ChIP peaks along the five *Arabidopsis* chromosomes (peak count/Mb). Most NDX peaks are enriched near centromeres, while less peak density is characteristic of arms.

root tip cells in which chromocenters can be visually detected. The NDX signal was weak in the nuclear interior, which typically coincides with euchromatin.

We then used the FCS approach to calculate the effective diffusion coefficient of the mobile fraction of NDX (Fig. 2c). From the FCS parameters, the time-dependent autocorrelation function was derived ($G(\tau)$ or ACF; Fig. 2d, e) from which we calculated the average diffusion coefficient at the nuclear periphery ($16.72 \pm 7.1\,\mu m^2$/sec) and the nucleolus ($5.00 \pm 0.9\,\mu m^2$/sec; Fig. 2f, g). Since the diffusion parameters did not differ significantly between nuclear and nucleolar NDX, these compartments are likely to have closely related NDX-binding properties. In the FRAP setting (Fig. 2h, left), GFP fluorescence was bleached within the nuclear periphery and nucleolus, and then fluorescence intensities were tracked (i) in the bleached region ($F_{bleach}$), (ii)

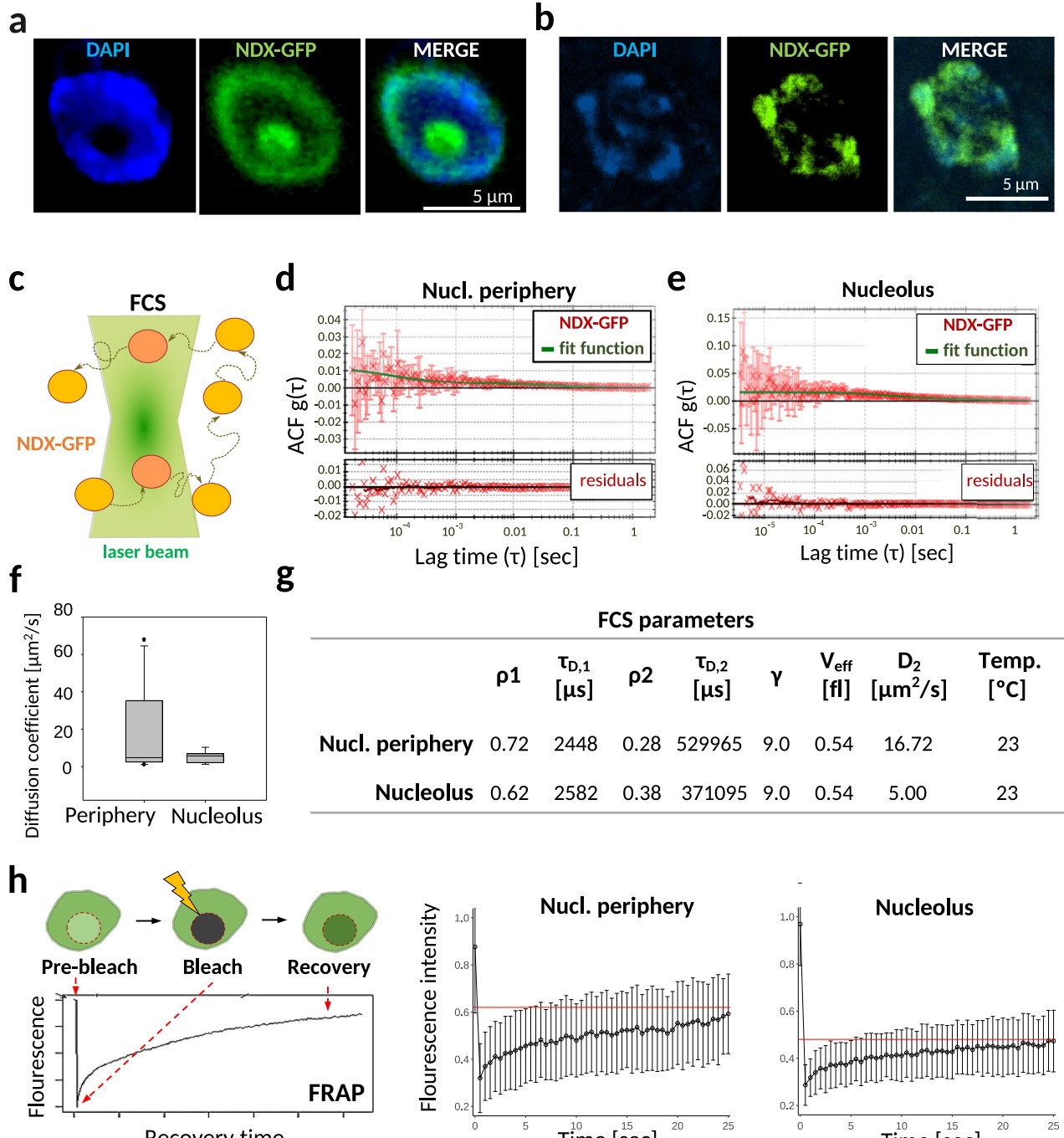

**Fig. 2 | Microscopic localization and molecular diffusion of NDX in Arabidopsis root tips. a**, **b** Representative confocal images showing the pattern of NDX-GFP distribution in DAPI stained nuclei. The experiments were repeated four times with similar results. **a** The GFP signal is enriched in the nuclear periphery and in the central (DAPI-negative) region, corresponding to the nucleolus. Nucleolar staining is detected in 5–20% of cells. **b** A minor fraction of cells with typical chromocenter structure (DAPI foci) shows strong NDX-GFP enrichment at chromocenters. **c** Scheme of fluorescence correlation spectroscopy (FCS) measurement. **d**, **e** Representative time-dependent autocorrelation functions (ACFs) with the estimated diffusion parameters and residuals. ACF curves represent the correlation coefficient between the fluctuation from the mean GFP fluorescence intensity at time I(0) and the fluctuation from the mean intensity at some time later I(t). The curves were fitted with a two-component 3D normal diffusion model. Sample size *n* = 16 individual cells examined over three independent experiments. **f** Distribution of diffusion coefficients (D) of fast components of NDX-GFP at the nuclear periphery and nucleolus. *D* values do not differ significantly (*p* = 0.820, Mann–Whitney test, two sided). Bounds of boxes describe the interquartile range with the median; whiskers indicate 95% confidence interval; dots show outliers. **g** Main FCS parameters and calculated values. ρ1: slow FCS component; $\tau_{D,1}$: diffusion time of slow component; ρ2: fast FCS component; $\tau_{D,2}$: diffusion time of fast component; γ: structure factor of the confocal volume; $V_{eff}$: effective measurement volume; $D_2$: diffusion coefficient of the fast component in μm²/s unit. The number of cells analyzed were 50. **h** Left: scheme of fluorescence recovery after photobleaching (FRAP) measurement. Right: FRAP curves showing slow fluorescent recovery of NDX-GFP at both the nuclear periphery and the nucleolus. Immobile (non-moving) fractions correspond to the area above the horizontal red line. Sample size *n* = 56 individual cells examined over 3 independent experiments. Error bars: SD.

in total nuclei ($F_{total}$), and (iii) in randomly selected regions outside the nucleus, used for background subtraction ($F_{bckgr}$). Recovery curves were obtained by normalizing the background-subtracted signal to the mean prebleach signal and, at the same time, considering the decrease of total fluorescence due to the initial high-intensity laser pulse and bleaching upon post-bleach imaging[34]. For the mobile fraction of NDX, we obtained a very slow recovery rate that recapitulates the slow kinetics and tight chromosome binding of nuclear core histone (H3/H4) proteins[35]. GFP fluorescence did not return to the initial pre-bleach value, leaving 40–60% of NDX in the slowly exchanging (static) fraction (Fig. 2h, area above the horizontal red line). The latter immobile fraction most likely represents NDX molecules that bind directly and permanently to chromatin. Taken together, the above quantitative microscopic data obtained in living cells are consistent with our ChIP-seq results and extend them to different spatial resolutions and timescales, indicating that NDX is a chromatin-binding factor that is stably incorporated into chromosomes.

## NDX shows opposite genomic distribution to R-loops

To address the possibility that NDX is a general R-loop regulator, we compared the chromosomal distribution of NDX binding sites with the genomic profile of R-loops. For this, RNA-DNA hybrids were immunoprecipitated with the S9.6 antibody[36,37] and mapped by DNA-RNA hybrid immunoprecipitation (DRIP) sequencing[38,39]. The specificity of S9.6 immunoselection was monitored by RNaseH treatment, which degrades the RNA strand of RNA-DNA hybrids (Fig. 3a–c). Our peak prediction identified 14124 R-loops in Col-0 seedlings that followed an antagonistic distribution to NDX (Fig. 3a, c) as 97% of DRIP peaks ($n = 13798$) and 85% of NDX peaks ($n = 1910$) showed no overlap (Fig. 3b, c). Differential association of NDX and R-loops was also analyzed in gene-rich euchromatic regions by a peak-independent (metagene) approach: strong DRIP signal enrichment was found over promoters and 5′-UTRs while NDX was enriched at transcriptional termination sites (Fig. 3d). The predicted R-loops were validated by DRIP-qPCR at selected loci that appeared as positive (enriched) and negative (depleted) sites according to our analysis (Fig. 3e). The R-loop structure previously described at the 3′-end of *FLC*[21] was also detected and further validated by qPCR (Supplementary Fig. 4), confirming the quality of our DRIP-seq data. Importantly, despite the opposite genomic distribution of NDX binding sites and R-loops, 15% of NDX peaks ($n = 333$, green) and 2.3% of DRIP peaks ($n = 326$, red) still overlapped (Fig. 3b, c, "common peaks"). This overlapping fraction is statistically significant compared to random enrichment (*$p < 0.001$, prop.test) and suggests that the NDX-stabilized R-loop model described for the *FLC* locus[21] may still be true in some cases, though not in general. Accordingly, secondary ssDNA structure prediction showed that NDX binding sites have a lower propensity to form ssDNA than DRIP peaks or common NDX/DRIP binding sites (Fig. 3f), which tend to adopt a single-stranded conformation. In other words, NDX binds primarily to double-stranded DNA regions and, to a lesser extent, to the classical R-loop structure containing ssDNA (represented by the common DRIP/NDX peak set). This is consistent with previous reports demonstrating a strong binding affinity of NDX for dsDNA templates[19,22]. However, mechanistic understanding of NDX binding to dsDNA, ssDNA, and R-loop structures remains unknown.

## NDX binds to het-siRNA loci associated with pericentromeric heterochromatin

Functional annotation of the identified ChIP and DRIP peaks revealed a significant enrichment of flag-NDX and NDX-GFP in pericentromeric regions and transposable elements (Fig. 4a). The Knob region, a constitutive heterochromatin domain located on the short arm of Chr4 rather than the pericentromere[40,41], also showed preferential NDX enrichment, suggesting that NDX is generally associated with this type

of chromatin. Of the annotated TEs, some of the families showed significant NDX-enrichment characterized by diverse transposition mechanisms, internal structure, and chromosomal distribution (Fig. 4b). NAD transposons, which are localized cytologically in the nucleolus and participate in forming so-called 'nucleolus associated chromatin domains' (NADs)[42], also showed NDX association (Fig. 4a), consistent with our microscopic data. We also performed a family-independent transposon analysis based on TE targeting by non-CG methylation pathways[43] and compared TE groups controlled by RdDM (RdDM TEs) to those targeted by the CMT2 pathway (CMT2 TEs) or both pathways (intermediate TEs) (Fig. 4c). The above TE categories were further classified by chromosomal location as pericentromeric and arm-associated. The results show that NDX is primarily enriched at TEs regulated by the CMT2 pathway as well as the common pathway and is depleted from RdDM-targeted TEs, regardless of pericentromeric or arm location. These functional relationships link NDX to non-CG methylation mediated by CMT2 and/or the CMT2/RdDM common pathway. This conclusion is reinforced by the association of NDX with repressive chromatin modifications (H3K9me2, H3K27me1) required to silence TEs (Fig. 4a, b). Of note, there was no colocalization between NDX and H2A ubiquitination (H2Aub) mediated by PRC1, which is not unexpected as this repressive mark is constantly removed by ubiquitin proteases to drive PRC2-catalyzed H3K27 trimethylation[44].

R-loops were mostly associated with 5′-UTRs, but also with promoters, specific transposons (e.g. NAD/En-Spm/MuDR/Copia), and euchromatic histone marks located in active chromatin (Fig. 4a–d). It is noteworthy that NAD TEs and "canonical" TEs (located outside the nucleolus) show differential R-loop association as the former group is enriched with DRIP peaks while the latter show depletion (Fig. 4a). Thus, R-loops appear to associate with nucleolar TEs and have less preference for extranucleolar TEs. These associations are consistent with previous research that found high R-loop abundance in the nucleolus[45–49]. Interestingly, NAD TEs associated with NDX and DRIP peaks overlap significantly ($p < 0.001$, prop.test), suggesting their possible functional interaction on a subset of nucleolar transposons.

Next, we correlated the localization of NDX and DRIP peaks with nine different chromatin states of *Arabidopsis* (Fig. 4e, left) reconstituted from the combinatorial pattern of epigenomic landscapes and transcriptome maps (based on ref. 50). NDX binding sites showed preferential associations with transcriptionally silent heterochromatin (states 8–9) enriched in H3K9me2, H3K27me1, CG/CHG/CHH methylation, transposons and transposon genes (Fig. 4e, right). In contrast, RNA-DNA hybrids mainly localized to transcriptionally active euchromatin, which corresponds to state 1 (high levels of transcription, H3K4me2/3, H3K36me3, H2Bub, H3.3, H2A.Z, and H3K9ac), state 2 (high levels of H3K4me2/3, H3.3, H2A.Z, and H3K27me3; moderate levels of active chromatin marks)[50]. Notably, NDX binding sites were highly anticorrelated with states 1–2 (where R-loop structures are typically enriched). Interestingly, states 1/2 and 8/9, the most extreme chromatin states, lie furthest apart on the linear scale of the genome[50], consistent with the reciprocal chromosomal distribution of NDX peaks and R-loops (Fig. 5a, top panel).

Functionally distinct small RNAs (sRNAs) were previously classified into nine functional groups and linked to different chromatin states[51]. Utilizing this sRNA database, NDX-binding sites showed a particularly strong colocalization with class 6–9 sRNA loci (Fig. 5a, b) that code for centromeric and pericentromeric 21–24nt siRNAs participating in CHH/CHG methylation via the CMT2/3 pathway (classes 6,8,9) or the RdDM pathway (class 7)[51]. However, NDX peaks were anticorrelated with class 1–3 euchromatic sRNA loci that tend to associate with genes. Nevertheless, NDX enrichment at class 6–9 sRNA loci was rather heterogeneous as NDX bound strongly to a subset of targets (Fig. 5c). Our metaanalysis shows that NDX does not bind to

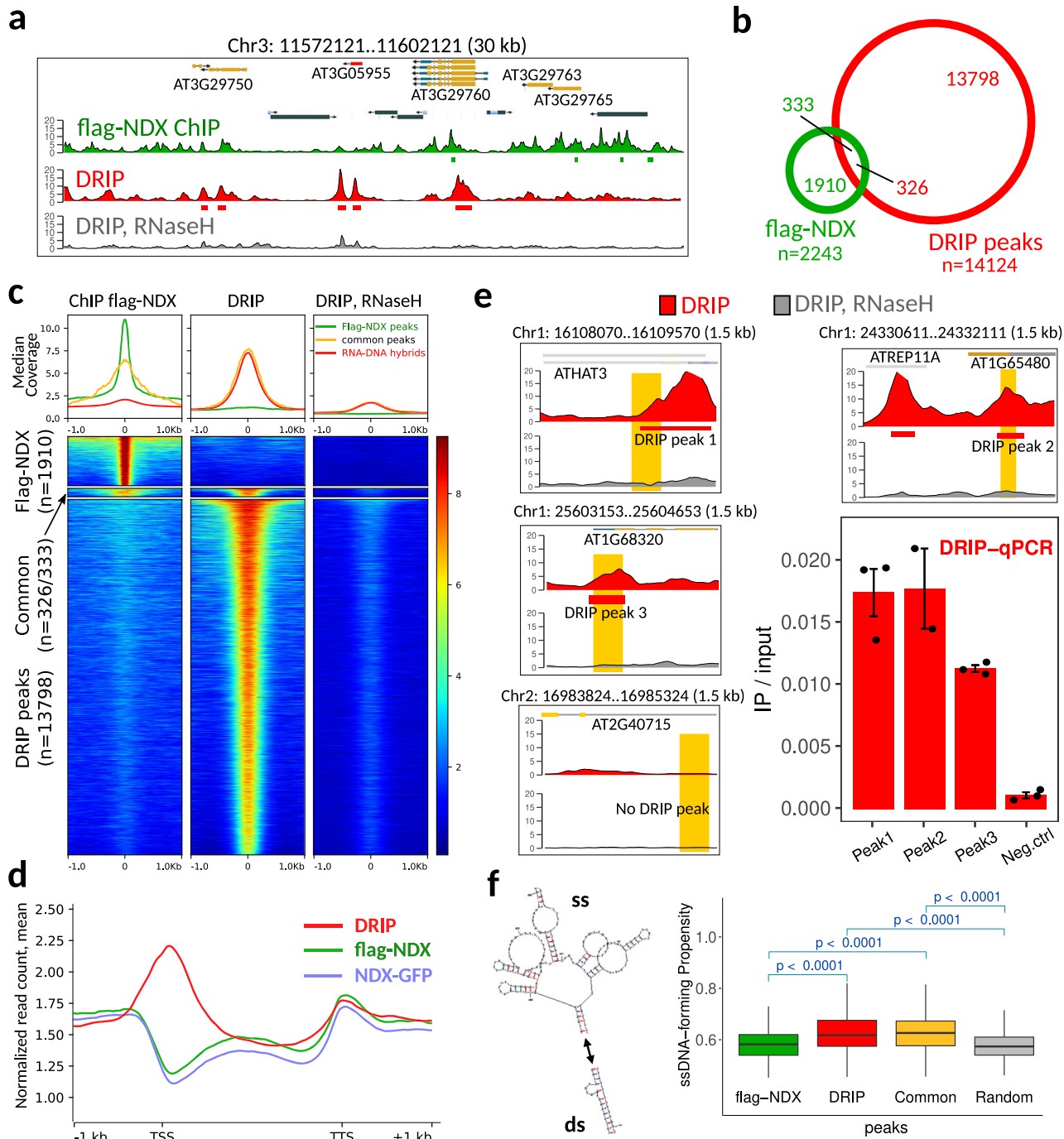

**Fig. 3 | Antagonistic genomic distribution of NDX binding sites and R-loops.**
**a** Genome browser snapshot showing the enrichment of flag-NDX and DRIP signals along chromosome 3. RNase H-treated sample represents the background DRIP signal. RNA-DNA hybrids were immunoprecipitated by S9.6 monoclonal antibodies. **b** Venn diagram showing the overlap of flag-NDX peaks and DRIP peaks (R-loops). The number of common peaks is not equal because ChIP peaks can overlap with multiple DRIP peaks and vice versa. **c** Pile up plot showing the antagonistic distribution of ChIP and DRIP signal intensities over the predicted peak positions. RNase H treatment is shown as a negative control for DRIP. Color scale corresponds to RPGC (reads per genomic content) values. Peak summits were aligned to zero positions. **d** Metagene profile of flag-NDX (light blue), NDX-GFP (light green) and DRIP signal (red) intensities (mean values) over protein coding ORFs. TSS: transcription start site. TTS: transcription termination site. **e** DRIP-qPCR validation of

representative DRIP peaks. Specific DRIP signal and background (+RNase H) signal are shown in red and black, respectively. Genomic positions and qPCR amplicons are indicated. Error bars represent SEM. **f** Secondary ssDNA structure prediction over NDX binding sites and RNA-DNA hybrids. The plot shows the propensity of single-strand formation of individual flag-NDX peaks and DRIP peaks, common NDX/DRIP peaks, and randomly selected regions. Sample size *n* = 300 peaks randomly selected from DRIP and ChIP peak lists, and *n* = 300 random peaks. The ssCount values were calculated from the primary nucleic acid sequence of the peaks using the mfold algorithm. Statistical significance and p values are indicated (Mann-Whitney U test, two-sided). Bounds of boxes describe the interquartile range with the median; whiskers indicate minimum and maximum values; outliers are not shown.

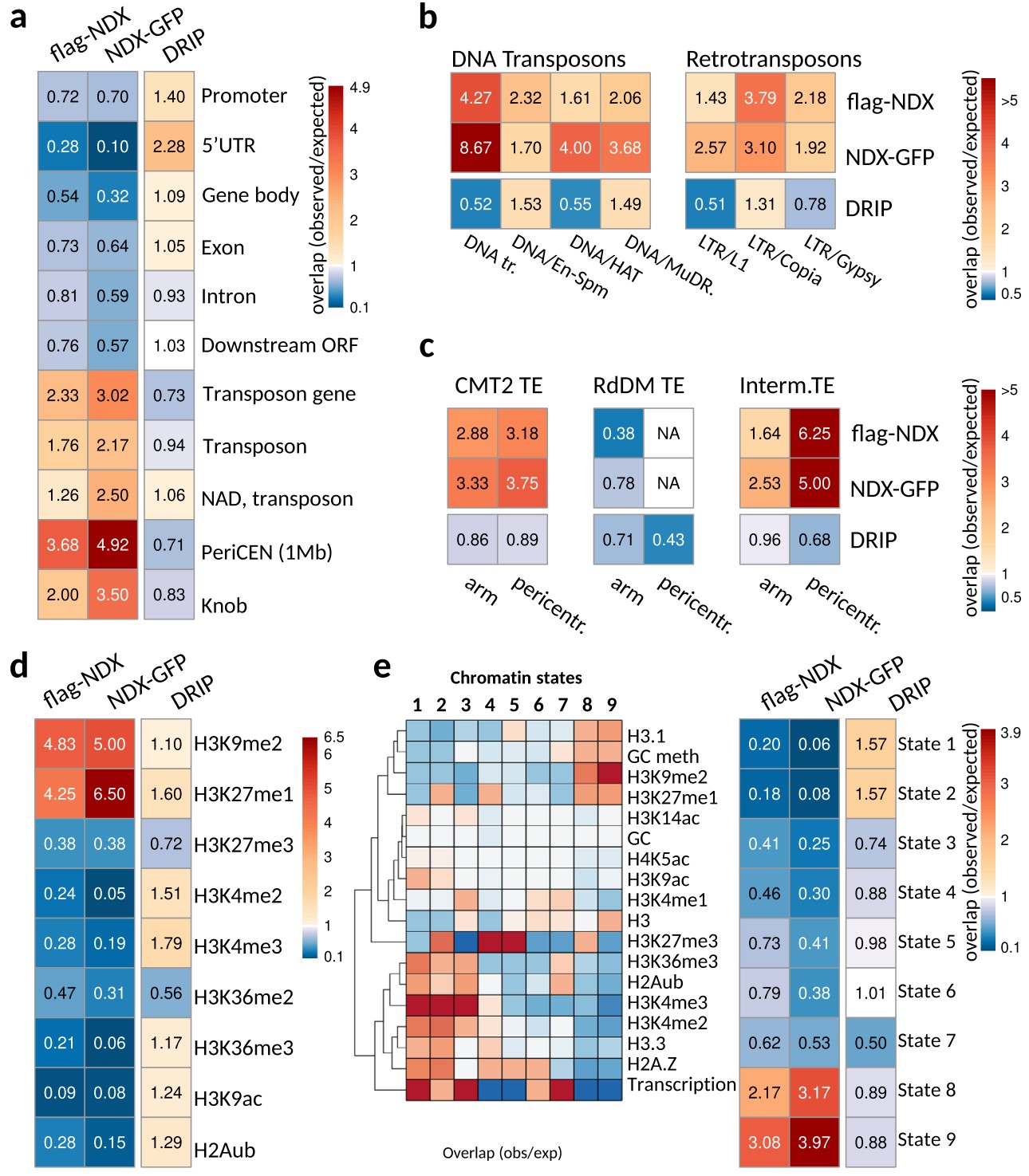

**Fig. 4 | NDX is preferentially associated with constitutive heterochromatin.**
**a** Functional annotation of flag-NDX, NDX-GFP, and DRIP (R-loop) binding sites over genomic features. Cells contain observed/expected ratios for peak counts. Warmer colors represent higher enrichment. **b** DNA transposon and retrotransposon families associated with NDX and DRIP peaks. (Only families showing significant enrichment of ChIP or DRIP peaks are shown.) **c** Enrichment of NDX and DRIP peaks over the functional categories of TEs. CMT2 TEs were compared to those controlled by the RdDM pathway or both pathways (intermediate TEs). TE categories were further classified as pericentromeric and arm-associated. NA: lack of statistical power due to very low peak count in the indicated category. **d** The same annotation as above for histone modifications, and **e** chromatin states. The nine chromatin states were defined by the combinatorial pattern of epigenetic modifications and transcription[50].

sRNA #1–5 loci (as expected), nor does it bind to sRNA #6–9 loci located on chromosomal arms, however, pericentromerically localized sRNA #6–9 loci bind significantly but heterogeneously to NDX (Fig. 5c). NDX enrichments were validated at selected class 6–9 sRNA loci using ChIP-qPCR (Fig. 5d). In contrast to NDX, RNA-DNA hybrids were mainly associated with sRNA classes 1–2 that are related to protein coding ORFs and promoters[51]. The genome-wide enrichment of NDX over het-siRNA loci is fully consistent with previous data that identified two sRNA molecules at the *FLC* locus (24 nt and 30 nt) that colocalized with NDX in a heterochromatic patch of H3K9me2[21,52].

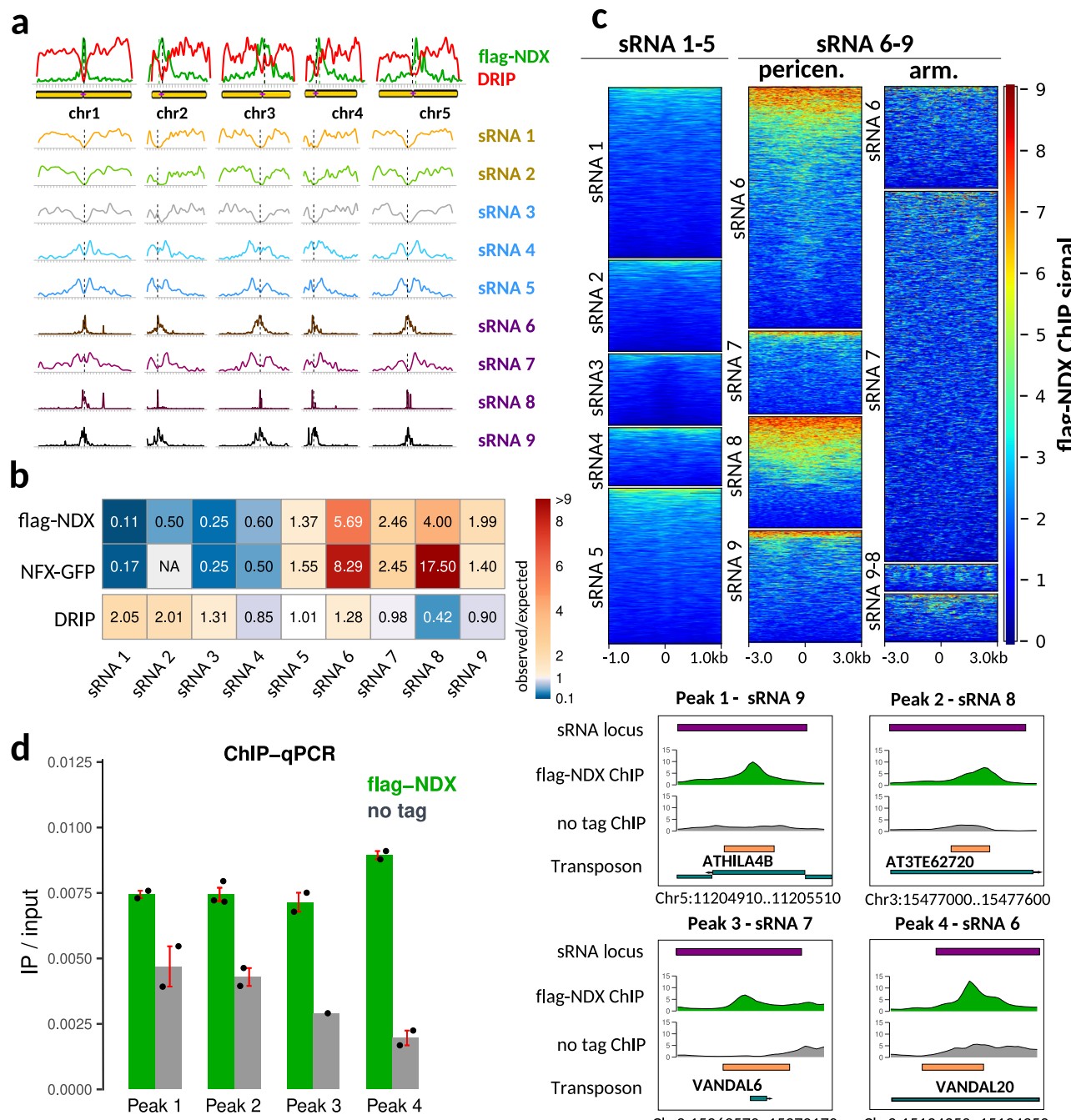

**Fig. 5 | NDX binds to heterochromatic small RNA loci (het-siRNA).**
**a** Chromosomal distribution of flag-NDX binding sites (green), RNA-DNA hybrids (red), and small RNA loci. sRNA loci are classified into nine functional clusters based on their genetic and epigenetic relationships and expression profiles[51]. NDX-binding sites correlate with cluster 6–9 sRNA loci while R-loops are correlated with cluster 1–2 sRNA loci. **b** Overlap of flag-NDX, NDX-GFP, and DRIP peaks with sRNA clusters relative to random associations. Cells contain observed/expected ratios for peak counts. Warmer colors indicate higher degree of association. **c** Metaplot analysis of NDX ChIP signal over sRNA loci 1–5 and 6–9. sRNA loci 6–9 have been divided by chromosomal location as pericentromeric and arm associated. The NDX ChIP signal is preferentially enriched in pericentromerically located sRNA 6–9 loci, however, their binding affinity for NDX is heterogeneous. **d** Left: ChIP-qPCR validation of representative sRNA loci that are bound by NDX. Specific ChIP signal and background signal ("no tag") is shown in green and grey. Error bar: SEM. Sample size $n$ = 2 biologically independent experiments. Right: JBrowse snapshot of the test regions.

## Loss of NDX function results in increased het-siRNA expression and R-loop formation

To gain functional insights into the association of NDX with het-siR-NAs, we performed sRNA deep sequencing (sRNA-seq) in an *ndx* T-DNA insertion line (*ndx1–4*) to assess global changes in the production of sRNAs (Fig. 6). Our analysis revealed significantly higher sRNA abundance in the *ndx1–4* mutant than in wild-type (Col-0), however, the size distribution and the ratio of dominant sRNA fractions (24nt/21nt) were similar (Fig. 6a, left). Of the nine functional groups of sRNA loci, only class 4–9 showed increased sRNA levels in *ndx1–4*, whereas class 1–3 did not differ from the wild type (Fig. 6a, right). These heterogeneities are in accordance with the primary division between sRNA loci that occurs between classes 1–3 and 4–9, determined by their associations with protein coding ORFs and promoters (sRNAs 1–3), and with epigenetically activated small interfering RNAs (easiRNAs) and non-CG methylation pathways (sRNAs 4–9)[51]. Differential expression analysis

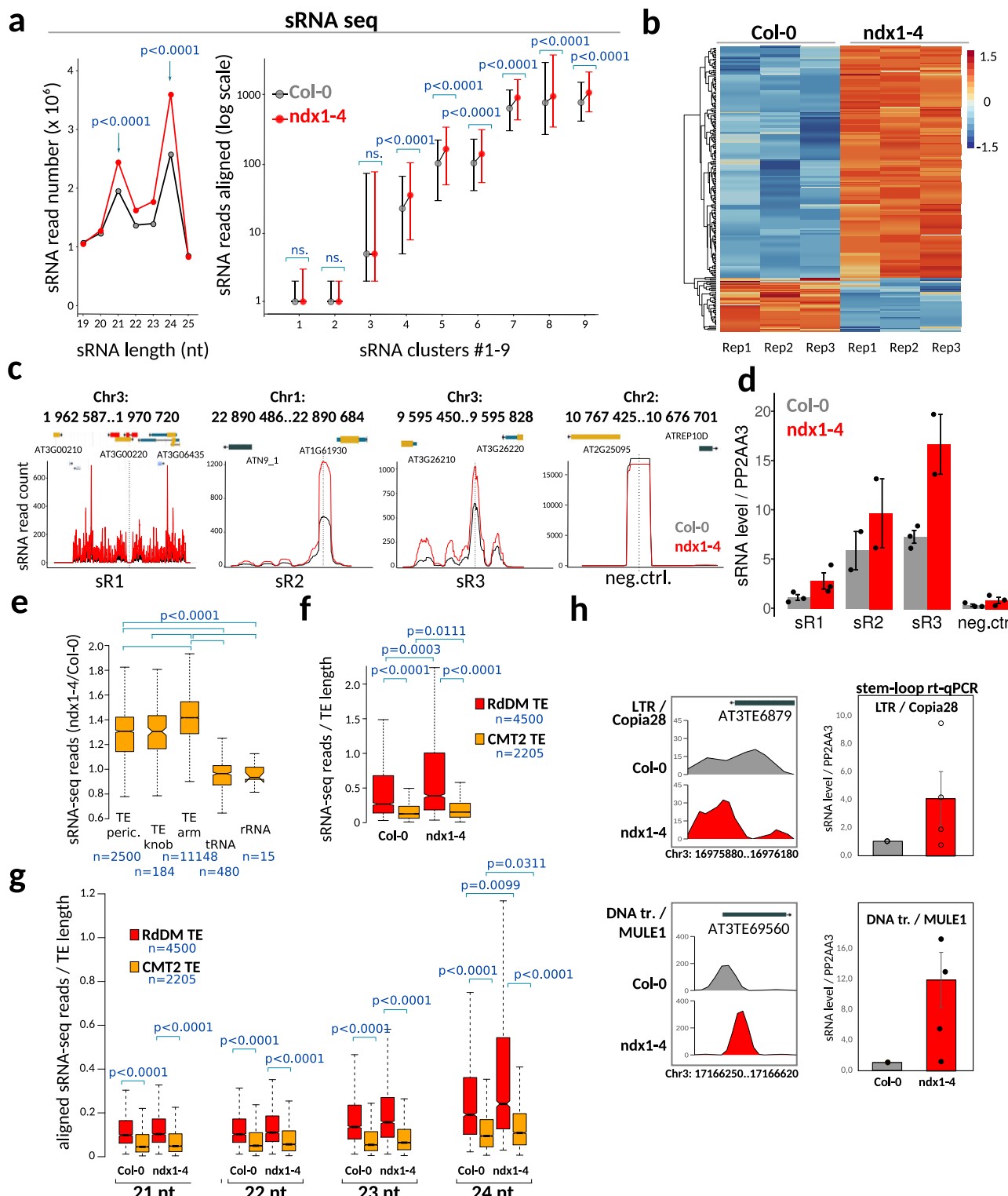

identified 286 upregulated and 165 downregulated sRNA loci ($n = 451$) that showed significantly increased or decreased sRNA levels in the *ndx1−4* mutant (Fig. 6b). About 85% of these sRNAs belong to classes 4–9 and thus depend on Pol IV activity (Supplementary Fig. 5) such that the proportion of classes 4 and 5 was doubled compared to non-differential ones. These associations demonstrate that Pol IV-derived/dependent siRNAs are specifically altered in the *ndx1−4* mutant.

Regarding the expression of annotated microRNAs (that are known to act post-transcriptionally), we found only 10 up- and 2 downregulated miRNAs from all known 326 precursor hairpin RNAs[53].

The small effect of *ndx1−4* mutation on miRNA transcription and the lack of association between NDX ChIP peaks and miRNA loci preclude the possibility that NDX is a central regulator of miRNA expression. In contrast, differential expression of class 6–9 siRNAs is probably due to the direct binding of NDX to these sRNA loci, which follows from their particularly strong association (Fig. 5b). Representative sRNA loci upregulated in *ndx1−4* and their validation by stem-loop rt-qPCR[54] are shown in Fig. 6c, d.

Regarding transposon-derived siRNA production, TE siRNAs also showed a clear increase in the absence of NDX, independent of their

**Fig. 6 | Loss of NDX function induces increased sRNA expression and R-loop formation. a** Left: sRNA expression levels are significantly increased in the *ndx1-4* mutant compared to Col-0. The size distribution of sRNA molecules peaks at 21-nt and 24nt, respectively. Sample size *n* = 3 biologically independent experiments. Statistical sifnificance is indicated (prop.test, two sided, Bonferroni correction). Right: Distribution of sRNA levels in Col-0 and *ndx1-4* plants over the nine functional sRNA clusters identified by[51]. Box plots show the medians and 95% confidence intervals. Clusters 1–3 are associated with protein coding ORFs and miRNA loci and sRNA levels show no difference between *ndx1-4* and Col-0. Clusters 4–9 are associated with Pol IV, and in part, Pol V and sRNA levels are significantly increased in *ndx1-4* (*p* < 2.2e-16, Kruskal–Wallis test with Mann–Whitney test (for multiple pairwise comparison) and Benjamini–Hochberg correction). Sample size *n* = 3 biologically independent experiments. **b** Heatmap showing upregulated (orange) and downregulated (blue) sRNA loci in *ndx1-4*. sRNA reads were aligned to sRNA loci from[51]. Three independent biological replicates are shown in the diagram. *P* < 0.0001, Mann–Whitney rank sum test. **c**, **d** Representative examples of upregulated sRNA loci and their validation by stem loop rt-qPCR. The bar chart shows positive strand expression levels normalized to PP2AA3 RNA expression. Sample size *n* = 3 biologically independent experiments. Error bar: SEM. **e** Increase of siRNA-levels in the absence of NDX function. The ratio of aligned siRNA reads (*ndx1-4*/Col-0) is significantly increased in pericentromeric TEs, Knob TEs and chromosome arm TEs relative to "negative control" regions (tRNA and rRNA genes). Statistics: Kruskal–Wallis test with Mann–Whitney test (for multiple pairwise comparison) and Benjamini–Hochberg correction. Sample size (*n*) is indicated. **f** The number of aligned sRNA reads show a significant increase in *ndx1-4* over the functional categories of TEs (RdDM TEs vs. CTM2 TEs). sRNA read counts were normalized to TE length. The siRNA levels from RdDM loci are also significantly higher than those from CMT2-only loci (both in Col-0 and *ndx1-4*; Statistics: Kruskal–Wallis test with Mann–Whitney test (for multiple comparison) and Benjamini–Hochberg correction. Sample size (*n*) is indicated. **g** The same as **f** but sRNAs were sized as 21, 22, 23, 24 nt siRNAs. In the 24nt class, there is a statistically significant difference between the expression status of *ndx1-4* and Col-0 samples. Bounds of boxes in **e**–**g** describe the interquartile range with the median; whiskers indicate minimum and maximum values; outliers are not shown. **h** Stem-loop rt-qPCR validation of increased sRNA production from Copia28 and MULE1 transposons (normalized to PP2AA3 expression). Error bar: SD. Sample size *n* = 4 biologically independent replicates.

pericentromeric, Knob, or arm association (Fig. 6e). We observed a significant increase in siRNA levels over RdDM TEs and CTM2 TEs such that higher expression was detected in the *ndx1-4* mutant relative to Col-0 (Fig. 6f; *ndx1-4* vs. Col-0 comparison). (We note that siRNA levels from RdDM TEs were substantially higher than those from CMT2 TE both in Col-0 and *ndx1-4* backgrounds, which is expected from the known mechanism of sRNA control, in agreement with published data[55].) When siRNAs were sized into 21, 22, 23, 24 nt classes, 24 nt siRNAs (representing het-siRNAs generated by Pol IV) were specifically upregulated in the *ndx1-4* mutant at both RdDM TEs and CMT2 TEs, as opposed to 21–22 nt siRNAs (so called epigenetically activated siRNAs, easiRNAs) that are related to TE expression (Fig. 6g). These associations further reinforce the link between NDX and Pol IV-dependent siRNA expression from TEs. The above siRNA changes were confirmed by stem-loop rt-qPCR at two representative transposons (Fig. 6h), suggesting that NDX is involved in transcriptional silencing of these loci. Interestingly, the pericentromeric ncRNA *siR1003*, expressed from the silenced 5 S rDNA array on chr3 (cytologically in the chromocenter), showed mildly increased siRNA levels by northern blot hybridization (Supplementary Fig. 6). This suggests that NDX not only affects the sRNA expression profile of transposons but may also silence other gene types in pericentromeric regions.

To elucidate whether the observed siRNA changes are associated with similar (or opposite) changes in the abundance of R-loops, we globally mapped RNA-DNA hybrids in the *ndx1-4* mutant (Fig. 7a). The DRIP-seq signal detected in both *ndx1-4* and Col-0 samples was reduced to background by RNaseH treatment, indicating the specificity of RNA-DNA hybrid detection. The amount of R-loops showed a moderate but statistically significant increase in the absence of NDX (fold change: >1.5; *p* < 0.0001), as evidenced by the higher intensity of DRIP peaks in the *ndx1-4* mutant, calculated by AUC analysis (Fig. 7b). Elevated genomic R-loop levels were also confirmed by slot blot hybridization (Fig. 7c, d) and DRIP-qPCR at selected genomic regions (Fig. 7f). Based on these data, NDX appears to prevent, rather than stabilize or stimulate, the formation of R-loops in the *Arabidopsis* genome. Moderately increased R-loop levels in *ndx1-4* could be due to an indirect consequence of the mutation rather than the direct effect of NDX binding, since R-loops and NDX follow opposite genomic distributions.

**Loss of NDX results in global transcriptional changes that may directly or indirectly affect heterochromatin status**

To assess the global effect of NDX loss on nuclear transcription, including euchromatin- and heterochromatin-derived transcripts, we performed an mRNA transcriptome analysis.

This analysis identified 1984 differentially expressed protein coding transcripts (*p* < 0.01) involving 864 up- and 1120 downregulated genes in the *ndx1-4* mutant compared to Col-0 control (Supplementary Fig. 7). GO term analysis showed that upregulated genes were primarily involved in the formation of ribonucleotide complexes, cold-response, flowering, and organ development, while downregulated genes were implicated in general stress response, postembryonic development, lipid storage, and binding of ribonucleotides (Supplementary Fig. 8). Our results were highly correlated with previously published data[19].

We also analyzed the activity of transposons at a global level. The NGS results show that loss of NDX had a mild but significant effect on TE activity: 28 and 21 transposons were up- or downregulated in the *ndx1-4* mutant (DESeq2 analysis; *p* < 0.05; Supplementary Fig. 9). This may be an underestimate since mRNA-seq captures only a portion of active TEs due to poly(A) selection[56]. Notwithstanding, the number of reactivated transposons is comparable to the number of upregulated TEs detected in *cmt2* and *drm1/drm2* heterochromatin mutants[9].

In addition, we sought to find chromatin regulators that are differentially expressed in *ndx1-4*. Indeed, we identified a group of genes that may play a direct or indirect role in the regulation of heterochromatin status. Overexpressed genes involve known RNAi factors such as (1) NRPD1b, NRPE3b, and NRPB/D/E9a, which represent the structural and regulatory subunits of Pol IV and Pol V, the core *trans* factors of RdDM; (2) RDR1, which participates in non-canonical RdDM[57]; (3) IDN2, required for siRNA accumulation and binding to dsRNA and lncRNA[58]; (4) AGO9, which is normally expressed in the ovule to interact with siRNAs transcribed from pericentromeric retrotransposons[59]; (5) ROS1, which demethylates several genomic targets to restrict non-CG methylation activity[60]. These RdDM genes are typically silent in seedlings (or expressed at low level) but become induced at high levels in the *ndx1-4* mutant. Downregulated chromatin factors include HTA4 (histone H2A), HON4 (linker histone like protein), and HMGB1 (high mobility group B1), which are involved in the assembly of nucleoprotein complexes. RT-qPCR validation of representative genes is shown in Supplementary Fig. 10. Misregulation of the above heterochromatin regulators are likely to contribute to the molecular phenotype of *ndx1-4*, however, causative relationships remain to be explored.

**NDX is required for proper non-CG methylation levels**

Since loss of NDX influences heterochromatin behavior (siRNA accumulation at hundreds of genomic loci and TEs), we performed bisulfite sequencing (BS-seq) to reveal the involvement of NDX in DNA

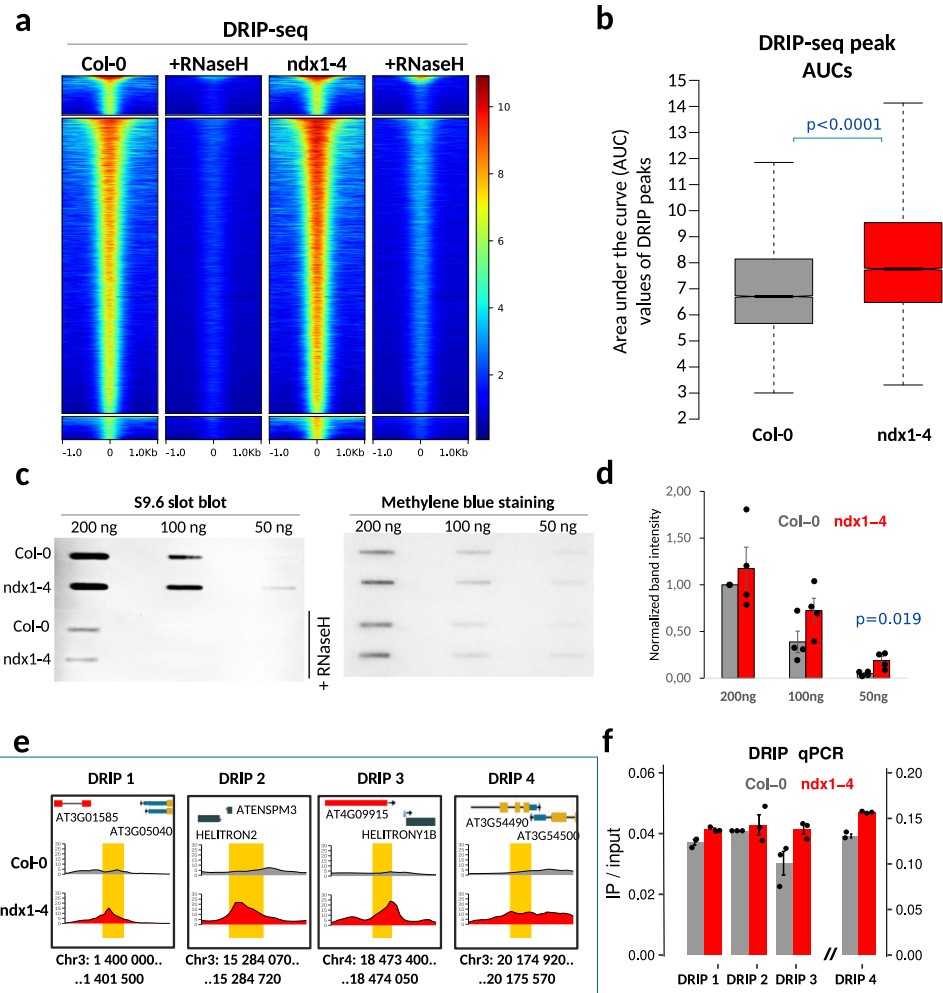

**Fig. 7 | Loss of NDX is associated with increased rather than decreased R-loop levels. a** Pile up plots showing DRIP signal intensities over the identified peaks in Col-0 and *ndx1–4* plants. RNase H-treated samples are also shown. Color scale corresponds to RPGC (reads per genomic content) values. **b** Statistical evaluation of DRIP peak intensities in Col-0 and *ndx1–4* plants. AUC (area under curve) values were calculated for each peak and their distributions were compared. AUCs, which are proportional to DRIP peak intensities, were significantly higher in the *ndx1–4* mutant (*p* < 0.0001, Mann–Whitney *U* test, two-sided). Bounds of boxes describe the interquartile range with the median; whiskers indicate minimum and maximum values; outliers are not shown. Sample size *n* = 15,049 peaks identified in Col-0 and *ndx1–4* plants. **c** Detection of RNA-DNA hybrids in Col-0 and *ndx1–4* plants using slot blot hybridization. 200 ng, 100 ng, and 50 ng of gDNA were slotted onto a nitrocellulose membrane with and without RNase H-treatment, stained with S9.6 antibody and goat anti-mouse-HRP secondary antibody. Equal loading was determined by methylene blue staining. **d** Band intensities were quantified by ImageJ (average values are shown). Error bars: SD. Statistics: Student's t test, two-sided. Sample size *n* = 3 biologically independent replicates. **e, f** Genome browser tracks and DRIP-qPCR validation of DRIP peaks identified in Col-0 (red) and *ndx1–4* (blue) samples. Error bar: SEM. Sample size *n* = 2 biologically independent experiments.

methylation (Supplementary Figs. 11, 12). Comparison of *ndx1–4* and Col-0 samples revealed 2449 hypomethylated and 1597 hypermethylated regions (DMRs) in the CHH/CHG sequence context (Fig. 8a), of which hypo-CHH/CHG sites were co-localized with NDX (Fig. 8b, c). In the CG context, 1353 hypo- and 1624 hyper-CG DMRs were detected, showing no enrichment for NDX binding (Fig. 8b). As expected, CHG/CHH DMRs significantly overlapped with TEs located in pericentromeric regions (Fig. 8d), and with chromatin states 8-9 that mark constitutive heterochromatin (Fig. 8e). In contrast, CG DMRs followed a more even distribution between the different annotation categories and chromatin states (Fig. 8d, e). These associations suggest that NDX influences transposon CHH/CHG methylation in pericentric heterochromatin regions. Grouping transposons by their regulatory classes and genomic positions highlighted decreased CHH and CHG methylation in CMT2 TEs in the *ndx1–4* mutant, regardless of pericentromeric or arm association (Fig. 8f). However, RdDM TEs showed reduced DNA methylation only in the CHG context located in chromosomal arms (mCG levels did not change in either class). Since NDX

binds directly only to CMT2 TEs (Fig. 4c), the above changes are likely due to the direct effect of NDX, whereas CHG methylation changes observed in RdDM TEs may be the indirect effect of *ndx1–4* mutation. Importantly, sRNA expression changes detected at hypo CHH/CHG DMRs appear to be independent of NDX binding, since there was no difference between hypo CHH/CHGs classified as "NDX-enriched" and "non-enriched" (Supplementary Fig. 13). This suggests that the *ndx1–4* mutation plays an indirect role in the production of siRNAs at hypomethylated CHH/CHG sites, and that the DNA methylase and RNAi systems can function without NDX binding (underscoring their epistatic relationship).

To explore hierarchical relationships between the DNA methylome changes of *ndx1–4* and heterochromatin mutants, we analyzed BS-seq data from a collection of mutants using the hcDMR pipeline[61]. The identified high-confidence DMRs confirmed well-known genetic interactions (e.g., between mutants operating in the RdDM and CMT2 pathways; Supplementary Fig. 14) and also identified new associations between *ndx1–4*, *drm2*, and *met1* as these groups were clustered

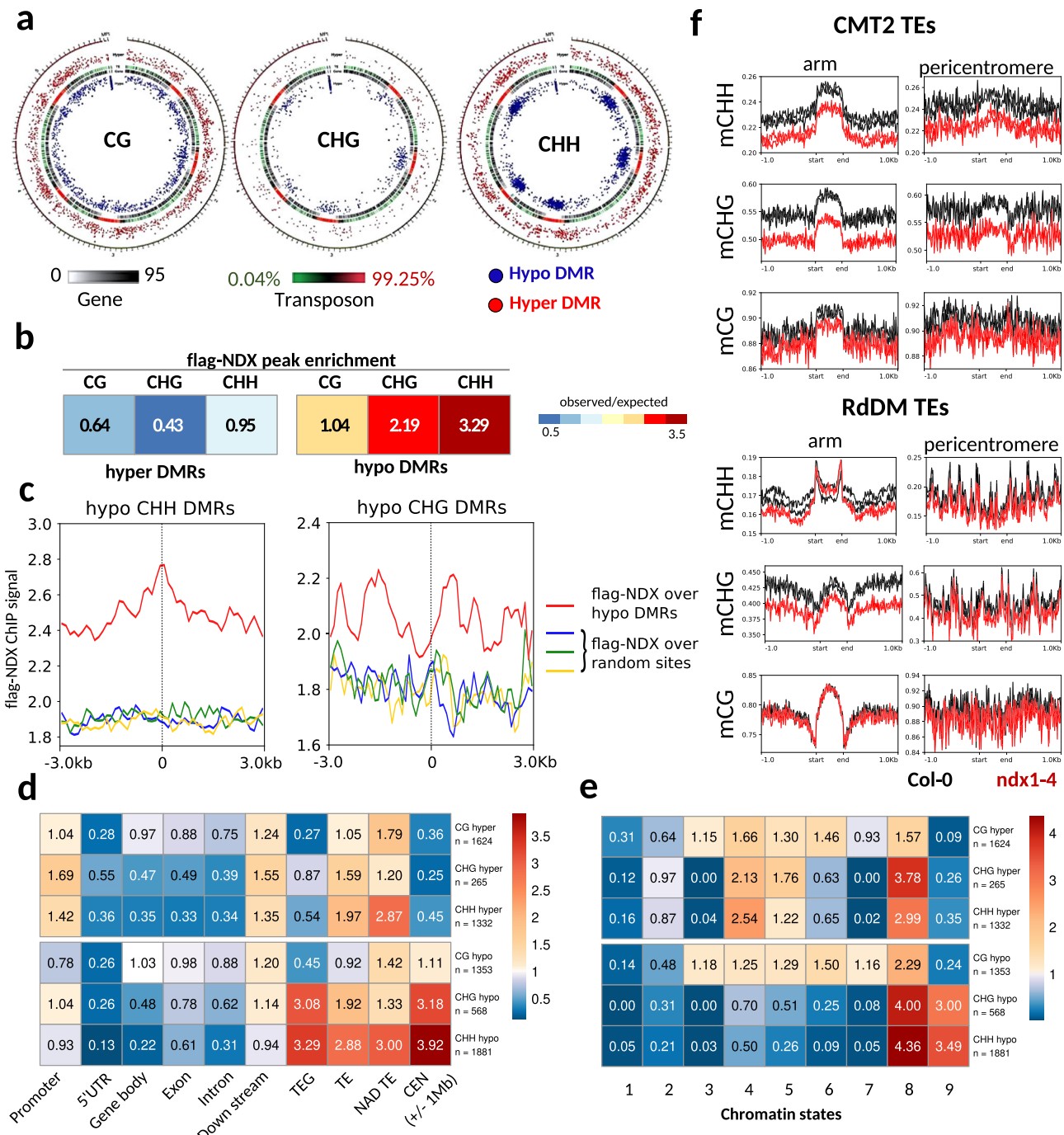

**Fig. 8 | Non-CG DNA methylation is affected in the absence of NDX. a** Circos plot showing the genomic distribution of differentially methylated regions (DMRs) in CG/CHG/CHH nucleotide contexts. Red and blue spots indicate hyper- and hypo-methylated DMRs in the *ndx1–4* mutant. Outer circle represents the five chromosomes and the mitochondrial and plastid genomes. Color and black-and-white heatmaps show gene density and transposon (TE) density. **b** Overlap of flag-NDX ChIP peaks with hyper- and hypo-DMRs. Cells contain observed/expected ratios for peak counts. Warmer colors represent higher enrichment. **c** Anchor plot of flag-NDX ChIP signal over hypo CHH and CHG DMRs (red line) and random sites (blue, green, yellow lines). The NDX signal is enriched in the middle of the CHH DMRs and

near the center of hypo CHG DMRs. **d, e** Annotation of hyper- and hypo-DMRs on **d** functional genomic features and **e** chromatin states. Cells contain observed/expected ratios for peak counts. Warmer colors represent higher enrichment. **f** CMT2 TEs and RdDM TEs were classified by chromosomal location as pericentromeric and arm associated. Metaplots show average DNA methylation levels in CHH/CHG/CG contexts in Col-0 (black) and *ndx1–4* (red) plants. CHH and CHG methylation is significantly decreased CMT2 TEs regardless of their arm- or pericentromeric location (mCG levels do not change). CHG methylation levels show reduction at RdDM TEs located in chromosome arms. (mCHH and mCG levels do not change.).

together. However, we could not clearly determine if NDX influences non-CG methylation via the CMT2 or RdDM pathways, or, more possibly, by a less characterized independent pathway (e.g., Knot linked silencing). As DNA methylation pathways show significant redundancies, appropriate classification is not obvious in terms of NDX function.

For instance, MET1 (the main CG methylase) and DRM2 (the main RdDM methylase) are both required to maintain CHH methylation in CMT2 targeted heterochromatin. The potential crosstalk between DRM2, MET1, and NDX remains to be explored by combining classical and molecular genetics.

## Loss of NDX results in rearrangement of chromatin organization in pericentromeric regions

Given the marked changes in siRNA accumulation, RNA transcription, R-loop formation, and DNA methylation in the *ndx1−4* mutant, we asked whether the observed molecular phenotype is related to underlying chromatin structural changes. We therefore examined the genome-wide chromatin conformations of *ndx1−4* and Col-0 plants using in situ Hi-C, and sequenced ~200 million valid reads for each genotype. Normalized contact matrices resembled published Hi-C maps[62–64] and showed similar patterns for *ndx1−4* and Col-0 at first glance (Supplementary Fig. 15). This indicates that loss of NDX does not lead to extensive restructuring of three-dimensional chromosome architecture in *Arabidopsis*. Nevertheless, we found remarkable quantitative changes between the two Hi-C maps during differential analysis of contact matrices (Supplementary Fig. 16). Several genomic regions were intensified (red) or diminished (blue) in the *ndx1−4* mutant, indicating a global rearrangement of spatial interactions in the absence of NDX. Circos plot analysis of the identified differential contacts revealed the topography of weakened (blue) and enhanced (red) Hi-C interaction network, highlighting that intrachromosomal contacts are mostly decreased, while interchromosomal contacts tend to increase in the *ndx1−4* mutant (Fig. 9a). Another striking change was the rearrangement of the Knot structure, which consists of an entanglement of ten genomic regions in wild-type plants[62,65]. Knot-forming regions showed increased interchromosomal contact frequencies in *ndx1−4*, with the entry of new regions and parallel loss of other intra- and interchromosomal connections (Fig. 9b). Interestingly, the T-DNA insert inactivating the *NDX* locus was also anchored to the Knot in *ndx1−4* plants, consistent with recent Hi-C data showing that transgene integration can induce novel Knot interactions[65]. We also analyzed Hi-C data from available epigenetic mutants (*met1*, *ddm1*, *clf28 swn7*) and compared them to *ndx1−4*. The results show that the Hi-C pattern of *ndx1−4* was similar to that of the *met1* and *ddm1* mutants, however, overall interaction frequency changes were milder in *ndx1−4* (Fig. 9b and Supplementary Fig. 16b, c). Several new regions joined the Knot in all three mutants that do not appear in Col-0 (Fig. 9b); however, in the *ddm1* and *met1* mutants additional regions also interact with the Knot that are not present in *ndx1−4* (Fig. 9c). Therefore, chromatin structure changes observed in *ddm1* and *met1* are more severe than in *ndx1−4*. The *clf28 swn7* mutant lacks all H3K27me3[66] and previous Hi-C analysis showed that spatial interactions of H3K27me3-enriched minidomains were reduced in *clf28 swn7*, however, no Hi-C change was found in the Knot[63]. Our analysis gave similar results as the Hi-C pattern of the *clf28 swn7* mutant was different from all three mutants (Supplementary Fig. 16d) and the de novo Knot interactions were absent from *clf28 swn7*. Therefore, in terms of genome organization, *clf28 swn7* appears to operate in a different pathway than *ddm1*, *met1*, and *ndx1−4*, which all showed somewhat similar Knot interaction patterns. These associations reinforce the link between NDX and DNA methylation pathways. Another important Hi-C change in *ndx1−4* occurred at pericentromeric regions, with a general decrease in local intra-centromeric interactions (Fig. 9e, upper panel) and an increase in inter-centromeric interactions (except for CEN chr1/CEN chr4, Fig. 9e, lower panel). The same trend was true for all intra- and inter-chromosomal interactions of centromeres that were significantly weakened (blue) or enhanced (red) in the *ndx1−4* mutant (Fig. 9f). We propose that reduced intracentromeric interactions in the absence of NDX results in chromatin decompaction at pericentromeric regions (where NDX is enriched in wild-type plants), which strongly correlates with the location of sRNA expression changes and CHH/CHG methylation changes observed in the *ndx1−4* mutant. Whether the above chromatin changes are directly or indirectly mediated by NDX remains unclear. Nevertheless, chromatin compaction analysis[67] suggests that NDX has a direct effect on the condensation state of hypomethylated CHH regions (Supplementary Fig. 17) as NDX-enriched hypo-CHHs

show significantly higher Hi-C compactness than hypo CHHs not bound by NDX (in wild-type plants). This indicates that NDX binding at these sites can directly promote chromatin condensation, which is inversely changed in the *ndx1−4* mutant due to the loss of NDX (see Hi-C data).

Finally, it should be noted that the chromatin binding of NDX does not scale with 3D chromatin compactness, i.e., NDX shows preferential enrichment in pericentromeric heterochromatin that is not due to the condensed state or high local DNA concentration of this chromatin type. To demonstrate this, we compared the Hi-C compactness of pericentromeric, Knob, and chromosomal arm regions in terms of NDX binding (Supplementary Fig. 18). The analysis showed that the compactness of all three regions differ significantly such that the most condensed is the heterochromatic Knob region, followed by pericentromeres and chromosome arms. Pericentromeric and Knob regions appear to bind more NDX per unit length than chromosome arms, however, the Knob showed similar NDX enrichment as pericentromeric regions despite its greater compactness. This suggests that NDX binding is disproportionate to compactness. When comparing regions with similar compactness (by random sampling from the above regions), chromosome arms still showed significantly lower NDX enrichment despite their same condensation state (Supplementary Fig. 18c). It follows that NDX preferentially interacts with constitutive heterochromatin, regardless of local chromatin density.

## Discussion

Heterochromatin is an essential structural feature of eukaryotic genomes, conferring special functional properties on different chromosomal regions[68]. The ability of heterochromatin to restrain DNA recombination events and limit the activity of transposons is critical to maintaining genomic stability. Therefore, it is crucial to identify specific heterochromatin regulators that control the formation and maintenance of repressive chromatin states associated with DNA methylation pathways, siRNA biogenesis, histone modifications, and 3D chromatin structure.

The homeodomain protein NDX was previously described as a transcriptional and/or epigenetic regulator of two euchromatic genes (*FLC*, *ABI4*) located in transcriptionally active chromosome arms[19,21]. In this work, we show that NDX is primarily a heterochromatin regulator that functions in pericentromeric regions to control the production of het-siRNAs and deposition of repressive CHH/CHG DNA methylation. There are multiple lines of evidence to support these claims: (i) ChIP-seq measurements using two different tags in independent transgenic lines (N-terminal flag and C-terminal GFP) show that NDX is strongly associated with pericentromeric regions; (ii) quantitative microscopy in live cells consistently show that NDX is a chromatin-binding factor with very slow nuclear dynamics, stably incorporated into peripheral heterochromatin; (iii) NDX preferentially associates with pericentromeric het-siRNA loci involved in non-CG methylation pathways, with significant sRNA transcriptional changes; (iv) CHH/CHG hypo-methylation of pericentromeric regions in *ndx1−4* significantly overlaps with NDX binding sites; and (v) loss of NDX function results in extensive 3D chromatin structural changes in pericentromeric regions.

The previously described NDX-associated *FLC* and *ABI4* loci are found in heterochromatin islands interspersed in euchromatic genomic regions[19,21]. Consequently, NDX may also control the activity of these "heterochromatin-marked" euchromatic genes, consistent with its proposed role as a heterochromatin regulatory protein. Integrating the NDX cistrome data with transcriptomic/epigenomic maps is expected to reveal similar mechanisms for other euchromatic genes. However, our DRIP-seq data strongly argue against the notion that NDX is a common factor that directly regulates widespread genomic R-loops. Instead, NDX appears to negatively regulate R-loop formation throughout the *Arabidopsis* genome, in contrast to its previously proposed positive role as an R-loop stabilizing factor at the *FLC* locus[21].

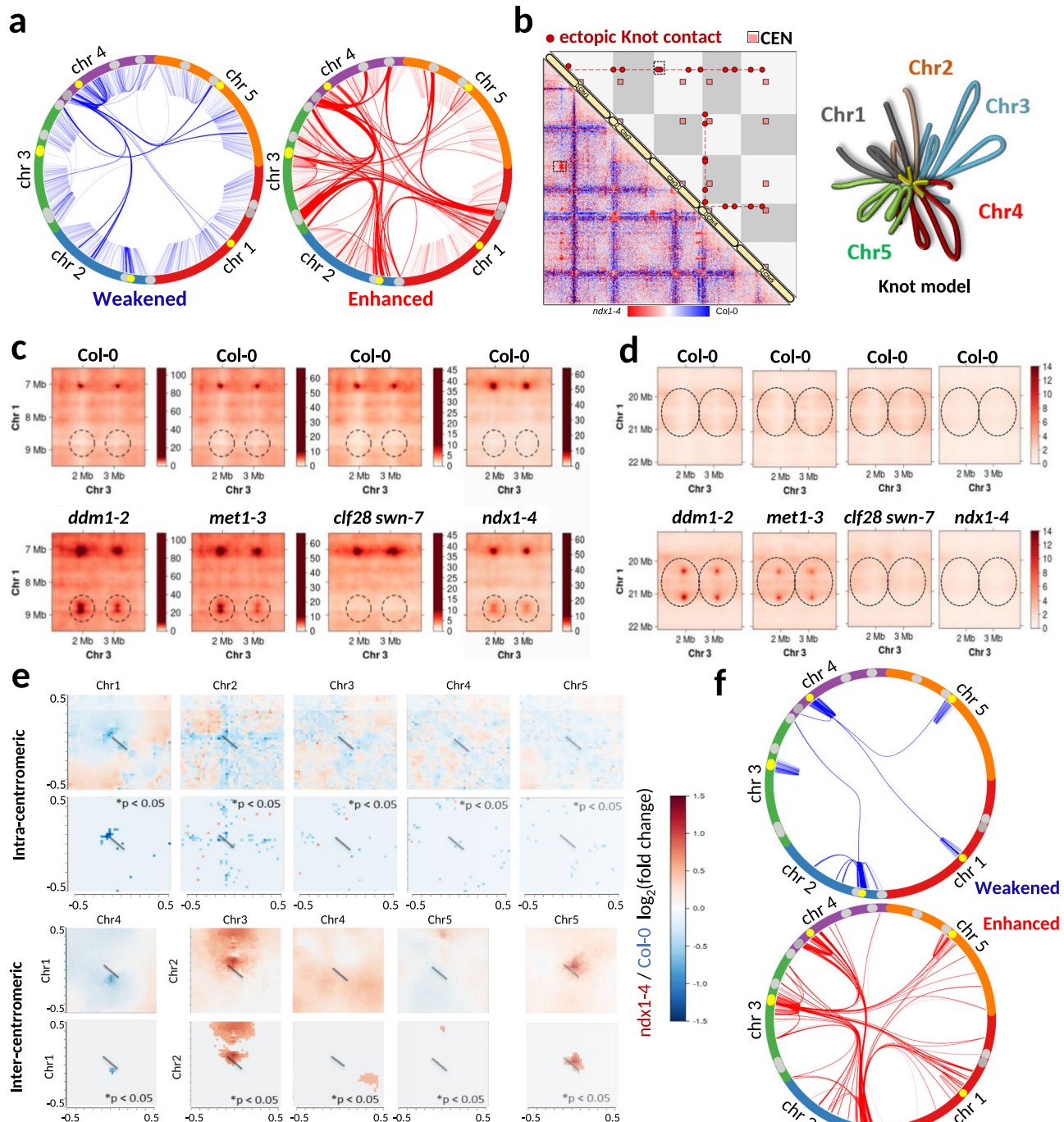

**Fig. 9 | Loss of NDX function causes rearrangement of chromatin organization.** **a** Circos plot representation of intra- and interchromosomal interactions that are significantly weakened (left) or enhanced (right) in *ndx1–4* (*p* < 0.05, Slitherine pipeline). Color intensity of the arcs is proportional to the fold-change in interaction frequencies (*ndx1–4*/Col-0). Outer circle represents the five *Arabidopsis* chromosomes. Centromeres are highlighted in yellow. Knot engaged elements (KEEs) are shown in gray. **b** Knot interactions in the absence of NDX. Diagonal: five *Arabidopsis* chromosomes. Lower left triangle (below the diagonal): differential Hi-C matrix calculated from the ratio of *ndx1–4* (red) and Col-0 (blue) interaction frequencies in Juicer. Resolution: 250 kb. Upper right triangle (above the diagonal): scheme of Knot interactions in the *ndx1–4* mutant. Knot-associated regions (circles) and centromeres (squares) and are highlighted. The T-DNA insertion site inactivating the *NDX* locus (red line in chr4) also participate in forming an ectopic Knot interaction. Right panel: Proposed 3D model of the Knot in the *ndx1–4* mutant. The structure involves de novo formation of inter-chromosomal contacts (highlighted in yellow). **c, d** Representative Knot contacts in *ddm1-2, met1-3, clf28 swn7*, and

*ndx1–4* mutants. Panel c shows de novo formed Knot contacts in the *ddm1-2, met1-3*, and *ndx1–4* mutants that are missing from Col-0 and *clf28 swn7*. Panel d shows a *ddm1-2* and *met1-3* specific Knot contact that is missing from *ndx1–4*. **e** Pericentromeric chromatin shows local decompaction and enhanced interchromosomal interactions in the absence of NDX. Upper: Differential intra-centromeric Hi-C interactions represented by a pileup plot. Red and blue colors show enrichment or depletion of intrachromosomal interactions of centromeres in the *ndx1–4* mutant. Diagonal bars show the exact locations of centromeric regions. Significantly different regions are shown below (identified by Slitherine; permutation test; p-value adjustment: Benjamini & Hochberg (FDR) method). Lower: The same as above but differential inter-centromeric interactions are shown. **f** Circos plot showing all intra- and interchromosomal interactions of centromeric regions that are significantly weakened or enhanced in the *ndx1–4* mutant (*p* < 0.05, Slitherine pipeline). Color intensity of the arcs is proportional to the fold change in Hi-C interactions (*ndx1–4*/Col-0). Outer circle represents the chromosomes. Centromeres are highlighted in yellow.

Because NDX binding sites and R-loops follow an antagonistic genomic distribution, we propose that NDX inhibits R-loop accumulation indirectly. Increased global R-loop levels in *ndx1–4* is likely due to the secondary effect of the mutation, but exact molecular mechanisms remain to be explored. One possibility is that an aberrant Pol II activity or overloading of R-loop resolving pathways may cause increased R-loop levels in the absence of NDX. This is supported by the association of R-loops with gene promoters and transcription terminators. Another possibility arises from the fact that plants contain vast amounts of free RNA molecules[69] that are capable of generating unscheduled R-loops under certain conditions. We speculate that upregulated het-siRNAs may be incorporated into chromatin as homologous RNA-DNA hybrids, resulting in an overall increase in the number of R-loops. Accordingly, when searching for sequence homologies between upregulated siRNAs and R-loops (using the psRNATarget algorithm[70]), we found a significant correlation in the number of siRNA targets homologous to R-loops compared to random sequences ($p < 0.0001$; Supplementary Fig. 19). This is consistent with the observation that class 1–2 sRNAs (where most R-loops are located) do consist not of true sRNA loci, but are targets for other sRNAs acting in trans[51]. These associations support a model in which upregulated siRNAs hybridize to homologous target sites that in turn raise global R-loop levels. The mechanistic link between NDX activity, siRNA biogenesis, and R-loop production awaits further experimental validation.

Another important question concerns the mechanism of action of NDX and potential causal relationships between structural and functional changes induced by loss of NDX. In *Drosophila*, the NDX-related zeste protein has the capacity to self-associate to form protein aggregates, consisting of several hundreds of monomers[71]. This self-assembly mechanism may also apply to *Arabidopsis* NDX with a potential impact on higher-order chromatin architecture. Based on our Hi-C data, NDX appears to mediate long-range chromatin interactions through chromocenters and transcriptionally silent chromatin. The chromatin structural changes in pericentromeric regions are probably specific for *ndx1–4* plants, as no similar genome architecture rearrangements were detected so far in other *Arabidopsis* mutants. However, the identified Knot interactions reflect genome structural changes observed in epigenetic mutants involving DNA methylation (e.g., *ddm1*, *met1*)[62,63,65], which further strengthen the link between NDX and DNA methylation pathways. It seems relevant in this regard that heterochromatin has been shown to induce significant chromosome compaction at centromeres in *Sc. pombe*, providing basic mechanical constraints for proper genome structure and function[72].

Based on our data, the following model is proposed to summarize the function of NDX in heterochromatin homeostasis: (i) inactivation of NDX results in chromatin decompaction at highly condensed pericentromeric regions (by an as yet unknown mechanism), (ii) relaxed heterochromatin structure leads to het-siRNA accumulation and DNA methylation changes, (iii) consequently, a subset of chromatin regulatory genes and transposons become derepressed or repressed. Increased global R-loop formation is likely due to the collateral effect of NDX loss and chromatin conformational changes. Whether the NDX-mediated nuclear chromatin organization is related to the CMT2 pathway (as may be suggested by some of our observation, Figs. 4c, 8f) or represents a completely independent route, needs to be further investigated. Genetic analysis of the above and other factors is expected to lead to a better understanding of heterochromatin homeostasis as the driver of genome organization and stability and its response to developmental signals and environmental stimuli.

## Methods

### Plant material

The following *Arabidopsis thaliana* genotypes were used in our study: Col-0, *ndx1–4*, flag-NDX (genotype: flag-NDX/*ndx1-1*(FRI)/*flc-2*), and NDX-GFP (genotype: NDX-GFP/*ndx1-1*(FRI)/*flc-2*). The NDX fusion proteins were expressed from their endogenous promoter. All transgenic plant lines were described previously[21,73]. Seeds were sterilized and sown on agar-MS plates, kept at 4 °C in dark for 2 days. After stratification, plates were placed to 21 °C, long day (LD, 16 h light, 8 h dark) conditions for 10 days. For immunofluorescence, NDX-GFP seedlings were grown for 10 days (14 h light, 10 h dark at 16 °C).

### FRAP and FCS microscopy

FRAP measurements were performed using an Olympus FluoView 1000 confocal microscope, based on an inverted IX-81 stand with an UPlanAPO 60 × 1.2-numeric-aperture (NA) water immersion objective. GFP was excited by the 488-nm argon-krypton laser line and fluorescence was detected through a 500–550 nm band-pass filter. For the optimal bleach depth, laser power was set to maximal intensity (-900 µW) with 500 ms duration. Cells were selected randomly in the root tip region of the plant and were bleached (in rectangular ROIs for 25 seconds) separately either in the nucleolus or in the nuclear periphery, where the NDX-GFP protein was highly expressed. Post-bleach images were acquired in 30 s intervals using low laser intensities. FRAP curves were normalized using Phair's double-normalization protocol[34,74]. In the FCS measurements, we recorded images and carried out autocorrelation analysis (at room temperature) within selected femtoliter-sized confocal volumes using an Olympus FluoView 1000 confocal microscope combined with FCS module[75]. NDX-GFP molecules passing through the confocal volume were excited with the 488 nm line of an Ar-Kr laser with minimized laser intensity (-0.9 µW). The emitted photons were detected from 500 nm to 550 nm. Fluorescence autocorrelation curves were calculated online by an ALV-5000E correlator card, which recorded the time course of the signal and calculated the autocorrelation function real-time. Data acquisition time was set to $8 \times 10$ s at each selected point. Autocorrelation analysis was performed by the QuickFit 3.0 software (Krieger, Jan; http://www.dkfz.de/Macromol/quickfit/) applying a 3D normal diffusion model for two-component fitting. The triplet state formation and slower dark states due to protonation were considered in the model function[76].

### DNA-RNA hybrid immunoprecipitation

2 g of 10-day-old seedlings of Col-0, and *ndx1–4* plants were harvested, and cross-linked under a vacuum chamber by phosphate buffered saline (PBS) containing 1% formaldehyde for $2 \times 5$ min. Excess formaldehyde was quenched by adding glycine to a final concentration of 0.125 M. Chromatin was purified and resuspended in Nuclei lysis buffer (50 mM Tris-HCl, 10 mM EDTA, 20 mM Hepes KOH pH 7.4, 10 mM MgCl$_2$, 0.5% Triton X-100, 5 mM DTT and Protease Inhibitors) followed by Proteinase K treatment (Thermo Fisher Scientific) and crosslink-reversal in the same step, at 65 °C, overnight. Total nucleic acid was prepared by phenol-chloroform extraction and precipitated with 2-volumes of ice cold isopropanol and 1/10 volume 4 M ammonium-acetate. Samples were then divided into two equal parts: half of the samples were treated with 8 µl of RNaseH (5000 U/ml, New England Biolabs) at 37 °C, overnight. Nucleic acid preps were diluted 4-fold in a ChIP dilution buffer (16.7 mM Tris-HCl pH 8, 1.2 mM EDTA, 1.1% Triton X-100, 167 mM NaCl) and DNA was fragmented by sonication to an average fragment size of 250–500 bp (low intensity, $3 \times 5$ min; 30 sec ON, 30 sec OFF, Bioruptor UCD-300, Diagenode). Dynabeads Protein A magnetic beads (Thermo Fisher Scientific) were pre-blocked with PBS/EDTA containing 0.5% BSA. The RNA-DNA hybrid specific S9.6 monoclonal antibody was prepared from the Hb-8730 mouse hybridoma cell line (Atcc) using protein A/G purification and stored in PBS/0.05% (w/v) Sodium Azide buffer. To immobilize the S9.6 antibody, 50 µl pre-blocked Dynabeads Protein A slurry was incubated with 10 µg of S9.6 antibody in IP buffer (50 mM Hepes/KOH at pH 7,5; 0,14 M NaCl; 5 mM EDTA; 1% Triton X-100; 0,1% Na-Deoxycholate, ddH$_2$O) at 4 °C for 4 h, with rotation. Six micrograms of fragmented genomic DNA were added to the mixture and gently rotated at 4 °C, overnight. Beads were

washed two times with 1 ml of ChIP lysis buffer (low salt, 50 mM Hepes/KOH pH 7.5, 0.14 M NaCl, 5 mM EDTA pH 8, 1% Triton X-100, 0.1% Na-Deoxycholate), 1 ml of high-salt ChIP lysis buffer (50 mM Hepes/KOH pH 7.5, 0.5 M NaCl, 5 mM EDTA pH 8, 1% Triton X-100, 0.1% Na-Deoxycholate), 1 ml of wash buffer (10 mM Tris-HCl at pH 8, 0.25 M LiCl, 0.5% NP-40, 0.5% w/v sodium deoxycholate, 1 mM EDTA at pH 8) and 1 ml of TE buffer (100 mM Tris-Cl at pH 8, 10 mM EDTA at pH 8) at 4 °C. Elution was performed in elution buffer (50 mM Tris-HCl pH 8, 10 mM EDTA, 1% SDS) for 15 min at 65 °C. After purification by a NucleoSpin Gel and PCR Clean-up kit (Macherey-Nagel), DRIP samples were analyzed by quantitative real-time PCR (qPCR) performed with a Light-Cycler 480 SYBR Green I Master mix (Roche) and a QuantStudio 12 K Flex Real-Time PCR System (Thermo Fisher Scientific). Primer sequences are listed in Supplementary Data 14. qPCR data were analyzed using the comparative CT method. RNA-DNA hybrid enrichment was calculated based on the IP/Input ratio.

### Chromatin immunoprecipitation (ChIP)

2 g of 10-day old flag-NDX (flag-NDX/*ndx1-1* (FRI)/*flc-2*), NDX-GFP (NDX-GFP/*ndx1-1*(FRI)/*flc-2*), and *ndx1–4* seedlings (no tag control) were harvested, and cross-linked under a vacuum chamber for 2 × 5 min in PBS containing 1% formaldehyde. Excess formaldehyde was quenched by adding glycine to a final concentration of 0.125 M. Nuclei were isolated and lysed in 300 μl of Nuclei lysis buffer (50 mM Tris-HCl, 10 mM EDTA, 20 mM Hepes KOH pH 7.4, 10 mM MgCl$_2$, 0.5% Triton X-100, 5 mM DTT and Protease Inhibitors) followed by a 1-minute centrifugation. The pellet was resuspended in Nuclei lysis buffer and the chromatin was fragmented by sonication to an average fragment size of 250–500 bp (low intensity, 3 × 5 min; 30 sec ON, 30 sec OFF, Bioruptor UCD-300, Diagenode). Immunoprecipitation and reverse crosslinking were performed as described previously[39]. For ChIP, 2 μg of antibodies against flag and GFP epitopes were used (anti-flag antibody - #2044 New England Biolabs; anti-GFP antibody - #ab290 Abcam). Dynabeads Protein A magnetic beads (Thermo Fisher Scientific) were pre-blocked with PBS/EDTA containing 0.5% BSA. To immobilize the anti-flag or anti-GFP antibody, 50 μl pre-blocked Dynabeads Protein A slurry was incubated with 2 μg of anti-flag or anti-GFP antibody in IP buffer (50 mM Hepes/KOH at pH 7,5; 0,14 M NaCl; 5 mM EDTA; 1% Triton X-100; 0,1% Na-Deoxycholate, ddH$_2$O) at 4 °C for 4 h, with rotation. Six micrograms of fragmented chromatin were added to the mixture and gently rotated at 4 °C, overnight. Beads were washed two times with 1 ml of ChIP lysis buffer (low salt, 50 mM Hepes/KOH pH 7.5, 0.14 M NaCl, 5 mM EDTA pH 8, 1% Triton X-100, 0.1% Na-Deoxycholate), 1 ml of high-salt ChIP lysis buffer (50 mM Hepes/KOH pH 7.5, 0.5 M NaCl, 5 mM EDTA pH 8, 1% Triton X-100, 0.1% Na-Deoxycholate), 1 ml of wash buffer (10 mM Tris-HCl at pH 8, 0.25 M LiCl, 0.5% NP-40, 0.5% w/v sodium deoxycholate, 1 mM EDTA at pH 8) and 1 ml of TE buffer (100 mM Tris-Cl at pH 8, 10 mM EDTA at pH 8) at 4 °C. Elution was performed in elution buffer (50 mM Tris-HCl pH 8, 10 mM EDTA, 1% SDS) for 15 min at 65 °C. Reverse cross-linking was carried out by adding 200 mM NaCl and 5 mM EDTA to the eluted product, incubated at 65 °C, overnight. After RNA and protein digestion by RNaseA and Proteinase K, DNA was isolated by a NucleoSpin Gel and PCR Clean-up kit (Macherey-Nagel). QPCR validations were performed by quantitative real-time PCR (qPCR) performed with a LightCycler 480 SYBR Green I Master mix (Roche) and a QuantStudio 12 K Flex Real-Time PCR System (Thermo Fisher Scientific).

### ChIP-seq and DRIP-seq library preparation

ChIP-seq and DRIP-seq libraries were prepared according to Illumina's TruSeq ChIP Sample Preparation protocol except that the uracil tolerant KAPA HiFi HotStart DNA Polymerase (Kapa Bio) was used. Briefly, the enriched DRIP and ChIP DNA samples were end-repaired and indexed adapters were ligated to the inserts. Purified ligation products were then amplified by PCR. Libraries were sequenced using 150 nt paired end read with Illumina HiSeq 2500 (EMBL Genomics Core Facility, Heidelberg, Germany) and 150 nt paired end read with Illumina NextSeq500 (Genomic Medicine and Bioinformatics Core Facility of the University of Debrecen).

### ChIP-seq and DRIP-seq data analysis

Sequenced reads were aligned to the *A. thaliana* reference genome (TAIR10; NCBI; Ecotype: Columbia-0) using the bowtie2[77] algorithm. Samtools[78] was used for making.bam files and indexing. Low mapping quality and PCR-duplicated reads were omitted from downstream analysis (http://broadinstitute.github.io/picard/). Deeptools[79] bamCoverage was used to create RPKM (Reads Per Kilobase per Million mapped reads) normalized coverage files (.bedgraph and.bigwig). Read densities were calculated for 20 bp bins (−binSize 20−operation ratio−smoothLength 60 −normalizeUsing RPKM) and IP/input bigwig ratios were made by bigwigCompare. Heatmaps were generated for 10 bp bins with computeMatrix and were plotted by plotHeatmap functions of deepTools. For metagene profiles, mean normalized read coverage were calculated in 20 bp windows with computeMatrix and were plotted using plotProfile (deepTools). MACS2[80] was used with default settings to identify ChIP peaks in input normalized flag-NDX, NDX-GFP, and DRIP peaks in input normalized and RNaseH-corrected Col-0 and *ndx1–4* samples, respectively. The identified peaks are listed in Supplementary Data 2–5. ChIP and DRIP data were visualized in JBrowse.

### Genomic annotation of DRIP and ChIP peaks

Enrichment of ChIP and DRIP peaks over functional genomic categories was performed by bedtools intersectBed to calculate overlap ratios[81]. Random peak sets were generated using the bedtools shuffleBed function. Observed/expected (random) ratios were plotted over genomic categories as a heatmap (in R project). The analyzed genomic categories were downloaded from Araport11 (.gff3 file). List of sRNA loci and sRNA clusters[51], mircoRNAs (https://www.mirbase.org/), chromatin states[50] were downloaded from the corresponding publications.

### Calculating ssDNA forming propensity

Propensity to form single-stranded DNA was estimated for NDX binding sites and RNA-DNA hybrids. We took a random sample of flag-NDX peaks, DRIP peaks and common (NDX/DRIP) peaks (*n* = 300 each) and then resized each peak to 400 bp. We also selected 400 bp long random regions as control (*n* = 300). We used the command line version of mfold software[82] to predict secondary structures for the nucleotide sequences of each peak. Overall ssDNA forming propensity was defined as the average proportion of individual bases adopting a single-stranded conformation, considering all computed models.

### Small RNA sequencing

Total RNA was purified from Col-0 and *ndx1–4* seedlings (3-3 biological replicates) using phenol-chloroform extraction. The quality of total RNA samples was checked by an Agilent bioanalyzer (RIN > 9). Small DNA libraries were prepared according to Illumina's NEBNext® Multiplex Small RNA Library Prep protocol. sRNA-seq libraries were sequenced using an Illumina NextSeq500 instrument with 1 × 50 bp reads. The results were analyzed using the sRNAnalyzer pipeline[83] as follows: Illumina adapters were trimmed by Cutadapt and reads were then size selected in the 19–25 nt range. Using sRNAnalyzer, size selected reads were aligned to the miRbase database (https://www.mirbase.org/) and a recently published comprehensive sRNA locus database[51]. Differential sRNA expression between Col-0 and ndx1–4 samples was determined by DEseq2 (*p* < 0.05, abs(log2(fc)) >log2(1.5)) and results were plotted in R. For sRNA target site prediction of differentially expressed sRNA reads over the identified DRIP-

seq peaks, the psRNATarget algorithm was applied using default settings[70].

## Stem-loop qRT-PCR analysis of sRNA expression

For sRNA qRT-PCR, we treated 5 µg total RNA with DNaseI (NEB, M0303) according to the manufacturer's instructions. RNA samples were precipitated in absolute ethanol and resuspended in sterile water. One microgram of total RNA was reverse transcribed using an sRNA-specific primer and U6- or PP2AA3 reference RNA specific primer according to the manufacturer's instructions (NEB, E6560). First, the stem-loop RT primer was hybridized to the sRNA molecule and then reverse transcribed in a pulsed RT reaction. Next, the RT product was PCR amplified using an sRNA-specific forward primer and a universal reverse primer. Specific primers for LTR/Copia28 and MULE1 transposon-derived sRNA molecules and reference RNA were designed according to Varkonyi-Gasic and Hellens (2010). qPCRs were performed by a LightCycler 96 Real-Time PCR machine (Roche) using Master Mix NEB M3003. Data was processed with a LightCycler® 96 software (Version 1.1.0.1320). The specific qPCR primers for sRNAs and reference RNA detection can be found in[84].

## mRNA sequencing (RNA-seq) and rt-qPCR

Total RNA was purified from Col-0 and *ndx1–4* seedlings using the phenol-chloroform extraction method. The quality of total RNA samples was checked by an Agilent bioanalyzer (RIN > 9). cDNA libraries were prepared according to Illumina's TruSeq RNA Library Preparation Kit v2 protocol. NGS libraries were sequenced using an Illumina MiSeq instrument with 1 × 50 bp reads (Biological Research Center of Szeged, Hungarian Academy of Sciences). We used Salmon to map RNA-seq reads and get transcript quantities for genes and transposable elements. Reads were mapped to the *A. thaliana* reference transcriptome (Araport11) to quantify gene expression whilst reads were mapped to transposable elements (Araport11) themselves to estimate their expression. using Salmon. Transcript quantities were corrected for GC bias to reduce isoform quantification errors. Differentially expressed genes and differentially expressed transposable elements (ndx1–4 vs. Col-0 wild type) were identified by DESeq2. The level of significance was defined by the adjusted *p*-values with independent hypothesis weighting[85] at *p*(adjusted) <0.01 for differentially expressed genes and at *p*(adjusted) <0.05 for differentially expressed transposable elements (DESeseq2). For data visualization, RNA-seq reads were aligned to the TAIR10 reference genome using HISAT2 allowing for reporting spliced alignments. We used deepTools bamCoverage to create RPKM (Reads Per Kilobase per Million) normalized bedgraph files. For rt-qPCR validation of differential mRNA expression changes, contaminating DNA was eliminated from total RNA preps by DNaseI digestion (RQ1 RNase-Free DNase, Promega). DNA-free RNA samples were reverse transcribed with random hexamers using SuperScript IV reverse transcriptase (Thermo Fisher Scientific). Real time qPCR was performed with a LightCycler 480 SYBR Green I Master mix (Roche) using a QuantStudio 12 K Flex Real-Time PCR System (Thermo Fisher Scientific). mRNA expression levels were normalized to UBQ10 gene expression.

## Gene ontology (GO) analysis

Singular enrichment analysis (SEA) of differentially expressed genes was performed by agriGO. Plant GO slim terms with a probability of *p* < 0.01 were called significant.

## Northern blot

For sRNA northern blot hybridizations, total RNA was extracted from 30 mg 10-day old seedlings. Plant material was homogenized, resuspended in an extraction buffer (0.1 M glycine-NaOH, pH 9.0, 100 mM NaCl, 10 mM EDTA, 2% SDS). The extracts were treated with phenol pH 4,3, phenol-chloroform and chloroform, precipitated in ethanol and

resuspended in sterile water. 30 µg of total RNA was separated by 12% PAGE (8.6 M urea, 1× Tris-borate-EDTA). The RNA was transferred onto Hybond-NX membranes and fixed by chemical crosslinking. The membrane was sequentially hybridized with 32P-ATP end-labeled (T4 PNK, NEB) cDNA oligos (IDT) that detect siR1003 siRNA and U6 snRNA, respectively. Quantification of band intensities was performed by ImageJ such that siR1003 expression was normalized to U6 levels.

## S9.6 slot blot

Genomic DNA was purified from 14-old Col-0 and *ndx1–4* seedlings using phenol/chloroform extraction. 50, 100, and 200 ng of gDNA preps were slotted onto nitrocellulose membranes with and without RNaseH treatment (Hybond-N + , GE Amersham) using a Bio-Dot SF Microfiltration Apparatus. Membranes were crosslinked with UV(2000) two times, and blocked in milk-TTBS (5% milk in 1× TTBS; 17 mM Tris, 130 mM NaCl, PH7.5, 1%Triton-X-100) for 1 h at room temperature. The S9.6 antibody was diluted in milk-TTBS at 1:1500 and membranes were incubated overnight at 4°C. Membranes were rinsed in milk-TTBS 3 times, each time for 10 min. The secondary antibody (goat anti-mouse-HRP) was added for 1 h at room temperature, followed by washes in milk-TTBS for 5 min and 1× TTBS 3 times, each time for 10 min. The S9.6 signal was detected by ECL reagent on the film. Equal loading of samples was determined by methylene blue staining of the membranes.

## Whole genome bisulfite sequencing (BS-seq)

Genomic DNA was prepared by standard phenol-chloroform extraction from 14-old seedlings (Col-0 and ndx1–4). 1 µg of gDNA was sent for whole genome bisulfite sequencing (BS-seq) to Novogene Ltd. Two independent biological replicates were analyzed from each background. DNA samples were fragmented into 200–400 bp using Covaris S220 and end-repaired, dA-tailed and ligated to sequencing adaptors containing only methylated cytosines. Then the DNA fragments were sodium-bisulfite treated with EZ DNA Methylation Gold Kit (Zymo Research) after which cytosines without methylation changed to U (after PCR amplification to T), while cytosines with methylation remained unchanged. 1 ng/l of PE150 library was prepared and sequenced on an Illumina NovaSeq 6000 instrument resulting in 15–17 million raw reads. The Bismark software[86] was used to align the bisulfite-treated reads to the TAIR10 reference genome. For the methylated sites, the methylation level is calculated using the following formula: $ML = mC/(mC + umC)$, where ML represents the methylation level, mC and umC represent the number of methylated and unmethylated cytosines, respectively. Methylation levels were determined for CpG, CHH, and CHG sequence contexts as percentage of methylated cytosines in their contexts and integrated with JBrowse. Differentially methylated regions (DMRs) were identified by the DSS-single (DSS) pipeline considering the variance among biological replicates. To compare the DNA methylome changes of *ndx1–4* plants to a large collection of DNA methylation mutants functioning in different pathways, we used the hcDMR pipeline[61]. High-confidence DMRs were identified by comparing the methylation changes of each mutant to the methylation levels of 54 control libraries. HcDMRs were clustered with the S-MOD method (statistical measurement of overlapping of DMRs), allowing the identification of hierarchical relationships between *ndx1–4* and DNA methylation mutants. SRA IDs of the mutants used in the analysis are listed in Supplementary Data 15.

## In situ Hi-C

Hi-C experiments were performed by the Arima HiC Kit (Arima Genomics). 10-day old Columbia wild type (Col-0) and *ndx1–4* seedlings were crosslinked with 1% formaldehyde as described in DRIP-seq experiments. According to the Arima User Guide for Plant tissue (https://arimagenomics.com/), 1 g of crosslinked plant tissue was used as a starting material for isolation of nuclei. Plant samples were ground

in liquid nitrogen using a mortar. 20 ml of PTNI buffer was then added (250 mM Sucrose, 20 mM HEPES pH 8.0, 5 mM KCl, 1 mM MgCl2, 40% glycerol, 0.1 mM PMSF, 1% protease inhibitor cocktail (Sigma), 0.25% Triton X-100, 0.1% mercaptoethanol) and samples were purified two times by using a double-layer Miraclot (Merk). Nuclei were washed several times in PTNI buffer as described in the Arima-HiC User Guide. Proximity ligated in situ Hi-C libraries were then constructed according to the Arima User Guide. Hi-C samples were fragmented to an average size of 400 bp (Bioruptor UCD-300, Diagenode; low intensity mode, 3 × 5 min; 30 sec ON, 30 sec OFF). NGS libraries for Illumina sequencing were prepared using the Accel-NGS 2 S Plus DNA Library Kit and Accel-NGS 2 S Indexing Kit (Swift Biosciences). To estimate the proper number of PCR cycles for library amplification, Arima-QC2 values were determined using the KAPA Library Quantification Kit (Roche). NGS library amplifications were performed until the estimated PCR cycle numbers (usually between 5 and 9) using the Kapa Library Amplification Kit (Roche). NGS libraries were then sequenced on an Illumina NextSeq 500 platform (2 × 150 nt, paired end reads) using a NextSeq 500 High Output v2 kit (Illumina). Raw Hi-C libraries (.fastq files) were processed by the Juicer toolbox (Durand et al. 2016) using default parameters, except that appropriate restriction enzyme cutting sites - provided by the Arima-HiC kit (GATC and GANTC) - were introduced into the script files. Hi-C reads were mapped to the *A. thaliana* TAIR10 reference genome and MAPQ ≥ 30 reads were retained for further analysis. Hi-C alignment statistics is summarized in Supplementary Data 12. We used the HiGlass tool[87] for visual exploration and analysis of the interaction maps. Quantitative differences in Hi-C interactions (Col-0 (wild type) vs. *ndx1–4* mutant) were identified at 25 kb resolution by the Slitherine pipeline (https://gitlab.pasteur.fr/gmillot/slitherine). Slitherine runs the Serpentine tool[88] to smooth local noise in Hi-C interaction maps and identifies statistically significant differences between Hi-C contact matrices. Regions that have significantly different Hi-C interactions in wild type vs. *ndx1–4* mutant are listed in Supplementary Data 13. For chromatin compactness analysis, we calculated the Hi-C condensation states of chromosomal regions following the method of Zhu et al.[67], with the difference that average Hi-C interaction frequencies were used within the specified genomic ranges to calculate compactness (instead of the sum of Hi-C interactions).

### Reporting summary

Further information on research design is available in the Nature Research Reporting Summary linked to this article.

## Data availability

Datasets generated for this study can be accessed in Supplementary Data 1–16 and via JBrowse (https://geneart.med.unideb.hu/pub/2021-ndx). Raw data are available at GEO GSE201841. External datasets: TAIR10 gene annotation files (gene list, splice junctions) were obtained from The *Arabidopsis* Information Resource. Promoter and downstream regions were defined as the arbitrary extension of transcription start sites (TSS) and termination sites (TTS) by 2000 base pairs. All other datasets used in this study are summarized in Supplementary Data 15. Source data are provided with this paper.

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

## Acknowledgements

This work was funded by HAS-Lendület-LP2015-9/2015, NKFIH-NNE-130913, GINOP-2.3.2-15-2016-00024, and the Thematic Excellence Programme (TKP2021-EGA-18) of the National Research, Development and Innovation Fund of Hungary. L.Sz was supported by the Bolyai Janos fellowship of the Hungarian Academy of Sciences and the UNKP-21-5-DE-11 and UNKP-22-5-DE-3 new national excellence program of the Ministry For Innovation and Technology from the source of the National Research, Development and Innovation Fund. M.M received support from NKFIH-K137678. T.Cs received grants from NKFIH-K129283, K137722, and K136513. We thank Caroline Dean (John Innes Centre, UK) for providing all the plant lines utilized in this study. We thank Csaba Máthé and Tamás Garda (Dept. of Botany, University of Debrecen) for their contribution to microscopic measurements. We are grateful for the Genomic Medicine and Bioinformatics Core Facility (University of Debrecen) for the NGS service.

## Author contributions

Á.M, A.H., B.B., Sz.H., É.N., H.Sz, T.Cs. performed the research, Zs.K, M.M., O.F., T.Cs., L.Sz analyzed the data, L.Sz., T.Cs., I.H. secured funding and supervised the work, L.Sz and T.Cs. wrote the manuscript.

## Funding

## Competing interests

The authors declare no competing interest.
