## [Peer Review File · Nature Communications]

NODULIN HOMEBOX is required for heterochromatin homeostasis in ArabidopsisEditorial Note: Parts of this Peer Review File have been redacted as indicated to remove third-party material where no permission to publish could be obtained.

REVIEWER COMMENTS

Reviewer #1 (Remarks to the Author):

This is a review of the manuscript entitled "Nodulin homeobox (NDX) is required for heterochromatin homeostasis in Arabidopsis" submitted to Nature Communications. In this work, Karányi et al perform a number of techniques that demonstrate that the previously established role of the protein NDX is not correct, and they define a new (and very different) role for NDX in the regulation of pericentromeric heterochromatin and suppression of transposable element expression by RNA-directed DNA methylation.

The strength of this work is the ChIP-seq of two different tagged NDX lines that for the first time identifies where in the genome this protein binds. This acts to focus the rest of the work onto the pericentromere. A second strength is the clear demonstration that NDX does not function in the formation or stabilization of R-loops, which had previously been reported. Below I detail the weaknesses of the work, split into the categories of major flaws and small changes to be made.

Major flaws

1. Since the pericentromere is condensed chromatin, many general DNA-binding factors would be expected to be concentrated there. Please factor in the condensation of this region of the genome to calculate if NDX is present there due to the increase amount of DNA at these regions, or if there is a specificity towards NDX's interaction with this chromatin type. For example, use the knob on chromosome 4 to determine if NDX goes where ever chromatin is condensed.
2. This work is over-generalizing the results of RT-PCR of two transposable elements to infer that many or all transposons are regulated in this manner. If the authors would like to make an argument on the whole-genome scale, RNA-seq is needed to quantify how many transposons are regulated by NDX.
3. Many of the figures require additional description, clarity or controls. For example:
 - a. In Figure 1C, please show the no-tag ChIP track as a negative control.
 - b. In Figure 3B-C, are the number of overlap between DRIP-seq R-loops and ChIP seq peaks expected by chance given datasets of this size, or is this an enrichment over the number expected by chance?
 - c. In Figure 4D there is a close relationship to siRNA regions #6-9 and NDX binding. Please include a meta analysis of NDX binding (via the available ChIP data) to these regions of the genome such as the analysis in figure 3C. It will be important to determine if NDX binds all of these regions, or just a subset strongly.
 - d. Figure 6B identifies regions that lose CHH/CHG methylation in ndx mutants AND bind NDX in wild-type. Please continue analyzing these regions to probe the mechanism of NDX activity. Particularly in Hi-C experiments and small RNA-seq experiments, what occurs at these regions that are likely the direct binding and action sites of NDX? More focus should be put on analyzing this "direct target" set in the manuscript.
4. The analysis of transposons was not up to par for a paper that significantly focuses on their regulation. For example:
 - a. The authors describe how their ChIP overlaps with transposons from Fig 4A, but there is a lot more of this type of analysis that should be done. For example, which transposon types? All copies of that TE family, or only the ones in the centromere?
 - b. It was not clear what "NAD" transposons are, or why they are important. It seems that NAD transposons are enriched at both NDX ChIP sites and R-loops, but these two datasets do not overlap much. Please expand to clarify.
 - c. Most transposons in the pericentromere are not regulated by RdDM. Instead, they are regulated by the maintenance of CHG and CHH methylation via CMT3 and CMT2, respectively. The authors should split the transposons into their regulatory classes (Pol V/RdDM vs. CMT2/3) and determine if NDX functions on one or both of these classes. Since NDX functions on the pericentromere, I would predict that these are CMT3/2 regulated, and not RdDM.
5. The Hi-C experiments do not separate the cause from consequence of ndx mutation. The authors

argue that the molecular phenotype is caused by underlying chromatin changes, but this may be a consequence, rather than the cause.

6. The authors argue that the Hi-C changes are similar to those seen in other chromatin modifiers such as *met1* and *ddm1*. However, the molecular changes in *met1* and *ddm1* are significantly more severe compared to *ndx*. For example, the CHH methylation loss in *ndx* is very subtle (Fig 6E). Therefore, the authors need to directly compare the Hi-C changes, expression changes and methylation changes to those mutants to make such comparisons. It is likely that the authors will find that *ndx* has similar patterns, but at a much lower strength compared to *met1* and *ddm1*. I recommend a direct data comparison whenever the authors wish to compare them in the Results or Discussion section.

Minor changes to be fixed

1. In the abstract it is claimed that Hi-C determines chromatin compaction. This is incorrect and should be fixed.
2. The Introduction is lengthy, and in particular the length of the section on H3K27 should be reduced, as this is not assayed or really discussed beyond the introduction in the manuscript.
3. Clear introduction and conclusion sentences to paragraph should be added to help guide the reader. For example, starting off a paragraph in the introduction "This possibility seems realistic given..." does not establish what "this" refers to.
4. Please clarify if the FLAG- and GFP- tagged versions of NDX complement the mutant phenotype before using them for ChIP.
5. For the DRIP sequencing, it was not immediately clear whether the authors had performed this experiment or downloaded the data from a previously-described work. Either way, please clarify if the DRIP-seq was performed in the same tissue type and developmental stage compared to the ChIP. This is important because they would be expected to overlap only if from the same tissue.
6. It was unclear why siRNA1003 is used in Fig. 6? Is this a transposon, and if so, what type?
7. In Fig 6G, it is unclear what the browser shots refer to? The Results section says transcript levels, but if this is RNA-seq data, a lot more needs to be done with this data in this manuscript. The production of RNA-seq is not described in the methods section. See major point #2.
8. Figure 1D leaves out knobs, NORs and other important features. In addition, it would be very interesting to introduce and bring in the different chromatin states into Figure 1D for overlap with their ChIP data.
9. Figure 2B is difficult to follow. I suggest putting different panel letters on the 5 different display items that are currently all labelled as "B".
10. Since much of the DRIP-seq data is negative with respect to overlap with NDX, I suggest confining the DRIP-seq analysis to a single main figure, rather than having it in several figures.
11. In Figure 4E, it was unclear to which category from part D each peak corresponds to.
12. In all browser shots in figures, please add the surrounding annotation or something that helps the reader understand what type of a region of the genome this is. Currently they lack this information content. For example, in Figure 5C&H, what are these regions of the genome that make small RNAs?
13. Figure 5A needs statistical testing
14. Hi-C data should be confined to one main figure plus other supplementals as needed.

Reviewer #2 (Remarks to the Author):

In this study the authors explore the function of the transcription factor NDX that has previously been identified to play a role in ABA signaling and to directly interact with the PRC1 components RING1A and B (Zhu, Plant Cell 2020). NDX has also been reported to be involved in stabilizing R-loop structures at the FLC locus (Sun, Science, 2013).

In contrast to these initial observations that could suggest a function in gene regulation, the authors find here by ChIP-seq that NDX is particularly enriched in pericentromeric heterochromatin and its binding sites are rather anti-correlated with positions of R-loops identified by DRIP-seq.

Instead NDX is enriched at regions enriched in chromatin states 8/9 and at class 6-9 sRNA loci involved in RdDM. In *ndx* mutants, siRNA loci of class 4-9 and some transposons are upregulated. The authors also determine DNA methylation changes in the mutant by BS-seq and find hypo CHH/CHG sites co-localizing with NDX. Finally they show that by Hi-C that loss of NDX leads to more interchromosomal contacts.

This study brings new insights on the role of NDX in pericentromeric heterochromatin and refutes a more general genome-wide role in R-loop stabilization. It further establishes a link with siRNA production, DNA methylation and transposon control as well as higher-order genome organization. While this is an overall thorough study including several genome wide analyses, bringing important new insight, it remains vague how NDX affects these different chromatin features.

Do the authors suggest, as the phenotypes are unlikely to involve R-loops, that the interaction with the PRC1 complex plays an important role? Indeed, several studies have suggested an ancestral role of Polycomb in transposon regulation (reviewed in D  l  ris, Trends in Genetics, 2021). Is a similar reactivation of transposons / siRNAs observed in *ring1ring1b* mutants?

Are any other chromatin regulators among the genes misregulated by loss of NDX?

How do the authors envisage that NDX would control HC compaction?

Can an NDX overexpressor rescue HC decompaction in other mutants such as *suvh456* or *atxr56*?

More specific comments:

Figure 2. I am a bit surprised that the authors look only in root cell nuclei, which are not the perfect cell type to investigate pericentromeric heterochromatin. Root cells have a big nucleolus and the chromocenters are not as well visible compared to other tissues. It would be good to see NDX-GFP enrichment also in other cell types - as here a clear enrichment in heterochromatin as would suggest the ChIP-seq results is not visible.

From the ChIP-seq data, it seems that NDX is enriched in pericentromeric HC, but in the conclusion they suggest that it is closely associated with peripheral heterochromatin. Are they suggesting also silent euchromatin domains associated with the nuclear lamina? They could also compare their enrichment data with RE-ChIP data from Bi et al, where they have been looking at NUCLEOPORIN1 (NUP1) enriched chromatin.

I would strongly suggest to also investigating NDX localization by immunofluorescence on the Flag-tagged version of the NDX protein. Often tags will alter the localization and the presence of a big fraction of NDX in the nucleolus might point in this direction.

It is not clear whether the NDX nucleolus fraction is a functional one or just storage of non-functional protein? Can the authors show that it is also present in the Flag-NDX line? What do the authors suggest NDX binds to in the nucleolus, as there is no chromatin except active rDNA copies? Does it fit with their conclusion of a chromatin-binding factor?

Figure 6: How does the altered interactome in *ndx1-4* correlate with changes observed in *ref6* (Huang, Genome Research, 2020)? Can the authors show a more detailed comparison between *ndx* and *clf* or *swi* interactome ?

Can the authors also look at heterochromatin organization by an alternative approach? E.g. a FISH study of 180bp repeats to visualize changes in the organization of chromocenters?

Minor comments:

Please indicate in the text and Methods whether the NDX fusion proteins are expressed from an endogenous promoter.

Figure 3d: Color of lines does not correspond to legend.

Figure 3f: Could same colors be used as in d to ease understanding for the reader.

Figure 4d: The sRNA6 peaks seem not to overlap with the position of the pericentromeric heterochromatin for chromosome 5?

Does the increased KNOT interactions involve the NDX locus itself (with its T-DNA)?

REVIEWER COMMENTS

Reviewer #1

This is a review of the manuscript entitled “Nodulin homeobox (NDX) is required for heterochromatin homeostasis in Arabidopsis” submitted to Nature Communications. In this work, Karányi et al perform a number of techniques that demonstrate that the previously established role of the protein NDX is not correct, and they define a new (and very different) role for NDX in the regulation of pericentromeric heterochromatin and suppression of transposable element expression by RNA-directed DNA methylation.

The strength of this work is the ChIP-seq of two different tagged NDX lines that for the first time identifies where in the genome this protein binds. This acts to focus the rest of the work onto the pericentromere. A second strength is the clear demonstration that NDX does not function in the formation or stabilization of R-loops, which had previously been reported. Below I detail the weaknesses of the work, split into the categories of major flaws and small changes to be made.

We sincerely thank the Reviewer for the favourable assessment and constructive suggestions to improve the manuscript. In our revision we addressed all the issues mentioned, and answered all the questions. Please note that we have considered *Nature Communications'* formatting guidelines (<https://www.nature.com/documents/ncomms-formatting-instructions.pdf>) and modified the original figures and text accordingly. The revised version contains 9 figures and 18 supplementary figures, which extend and confirm our major conclusions about the role of the NDX protein.

Major flaws

1. Since the pericentromere is condensed chromatin, many general DNA-binding factors would be expected to be concentrated there. Please factor in the condensation of this region of the genome to calculate if NDX is present there due to the increase amount of DNA at these regions, or if there is a specificity towards NDX's interaction with this chromatin type. For example, use the knob on chromosome 4 to determine if NDX goes where ever chromatin is condensed.

We analyzed chromatin condensation states from our Hi-C data following the method of Zhu *et al.* (Genome Biology 2017; <https://pubmed.ncbi.nlm.nih.gov/28830561/>), with the difference that average Hi-C interaction frequencies were used within the specified genomic ranges to calculate compactness (instead of the sum of Hi-C interactions). First, compactness of pericentromeric, Knob, and chromosomal arm regions was compared (**Revision Figure 1a**). The analysis shows that the

compactness of all three regions differed significantly such that the most condensed was the Knob region, followed by pericentromeric and chromosome arm regions. Next, we compared the NDX ChIP signal distribution along the above regions and found that the pericentromeric and Knob regions show significantly higher NDX association than chromosome arm regions (**Revision Figure 1b**). Importantly, the Knob showed similar NDX-association relative to pericentromeric regions, however, its compactness was significantly greater than that of pericentromeres. This suggests that NDX binding does not scale with compactness. When comparing regions with *similar* compactness scores (by random sampling from pericentromeric, Knob and arm regions), chromosome arm regions still showed significantly lower NDX enrichment despite their similar compactness (**Revision Figure 1c**). From these associations we conclude that NDX preferentially interacts with constitutive heterochromatin characteristic of the pericentromere and the Knob region, regardless of local chromatin density.

Revision Figure 1. The relationship between chromatin compactness and NDX binding. (a) Distribution of compactness scores calculated from Hi-C data. Pericentromeric, Knob (Chr4:1,560,561-2,082,885; PMID: 32312999), and chromosome arm regions are compared in 50 kb bins. (b) Distribution of NDX ChIP signal (average values in 50 kb bins) over pericentromeric, Knob and chromosome arm regions. (c) Distribution of NDX ChIP signal over genomic regions with similar Hi-C compactness, randomly sampled from pericentromeric, Knob and arm regions. CEN+Knob regions were merged since the number of relevant bins were very low in the Knob region to assign statistical significance. Statistical significance: ** $p \leq 0.01$; **** $p \leq 0.0001$; n.s.: non-significant, $p > 0.05$ (Mann-Whitney test).

The above data have been incorporated in the revised manuscript as Supplementary Figure 17 and text in the Results section, last paragraph before Discussion.

2. This work is over-generalizing the results of RT-PCR of two transposable elements to infer that many or all transposons are regulated in this manner. If the authors would like to make an argument on the whole-genome scale, RNA-seq is needed to quantify how many transposons are regulated by NDX.

The manuscript has been supplemented with further genome-wide small RNA-seq analysis and new mRNA-seq experiments aiming to measure global sRNA production from transposons and quantify transposon reactivation at the same time. We would like to note that since transposon silencing requires active transcription from TEs, it is difficult to differentiate siRNA production required for heterochromatinization from transcriptional reactivation of TEs. A further challenge is that mRNA-seq captures only a subset of activated TEs due to poly (A) selection.

With these premises, our small RNA-seq results show a clear increase in transposon-derived siRNA levels in the absence of NDX function, which was independent of pericentromeric, Knob, or arm association of TEs (**Revision Figure 2a**). We further observe a significant increase in siRNA expression over different functional categories of TEs (i.e. RdDM TEs and CMT2 TEs; **Revision Figure 2b**) such that higher expression is detected in the *ndx1-4* mutant relative to Col-0 in each case (*ndx1-4* vs. Col-0 comparison). (We note that siRNA levels from RdDM TEs were substantially higher than those from CMT2 TE both in Col-0 and *ndx1-4* backgrounds, which is expected from the known molecular nature of sRNA regulation by CMT and RdDM pathways, in agreement with published data (e.g. He L *et al.* (2021) Proc Natl Acad Sci U S A). Together, the above global changes indicate that most transposons are associated with increased siRNA expression in *ndx1-4* plants. The observed transcriptional changes were confirmed by stem-loop qRT-PCR at two representative transposons (LTR/Copia28 and DNA/MULE1; **Revision Figure 2c**), validating our sRNA-seq results.

Revision Figure 2. Global increase of siRNA-levels in the absence of NDX function. (a) The ratio of aligned siRNA reads (*ndx1-4* / Col-0) is significantly increased in pericentromeric TEs, Knob TEs and chromosome arm TEs relative to “negative control” regions (tRNA and rRNA genes) and the genome-wide average value (horizontal red line). Statistical significance: *** $p < 0.0001$ (Wilcoxon rank sum test). (b) The number of aligned sRNA reads show a significant increase in the *ndx1-4* mutant over the functional categories of TEs (RdDM TEs vs. CMT2 TEs). sRNA read counts were normalized to TE length. The siRNA levels from RdDM loci are also significantly higher than those from CMT2-only loci (both in Col-0 and *ndx1-4*), in line with published sRNA-seq data (e.g. He L *et al.* (2021) Proc Natl Acad Sci U S A). Statistical significance: * $p < 0.05$ (Wilcoxon rank sum test). (c) Stem loop rt-qPCR validation of increased sRNA production from Copia28 and MULE1 transposons (normalized to PP2A3 levels).

Herein we note that the above sRNA validation method (stem-loop qRT-PCR) is different from the conventional rt-qPCR, which is now better elaborated in the revised text. Stem-loop qRT-PCR relies on special stem-loop primers (instead of linear primers) combined with pulsed reverse transcription that specifically detects 21-24nt small RNAs with high sensitivity (Varkonyi-Gasic and Hellens (2010)). The main text and the Methods section are modified accordingly: “We used stem-loop qRT-PCR analysis to detect small RNA expression”... “Total RNA was reverse transcribed using an sRNA-specific primer and a reference-RNA specific primer according to the manufacturer’s instructions (NEB, E6560). First, the stem-loop RT primer was hybridised to the sRNA molecule and then reverse transcribed in a pulsed RT reaction. Next, the RT product was PCR amplified using an sRNA-specific forward primer and a universal reverse primer. Specific primers for LTR/Copia28 and MULE1 transposon-derived sRNA molecules and reference RNA were designed according to Varkonyi-Gasic and Hellens (2010).”

The above sRNA measurements were supplemented with new mRNA-seq experiments performed in Col-0 and *ndx1-4* seedlings to identify *de novo* transposon reactivation or repression (and find

chromatin regulators with altered expression levels that potentially affect siRNA biogenesis, as requested by Reviewer 2). The mRNA-seq results show that loss of NDX had a mild but significant effect on TE activity: 28 and 21 transposons were up- or downregulated in the *ndx1-4* mutant (DESeq analysis; $p < 0.05$; **Supplementary Fig. 7**). This may be an underestimate since mRNA-seq captures only a portion of active TEs due to poly(A) selection (most TEs lack a poly(A) tail). Notwithstanding, the number of reactivated transposons is comparable to the number of upregulated TEs detected in *cmt2* and *drm1/drm2* heterochromatin mutants⁹.

Revision Figure 3. Loss of NDX function weakly affects transposon activity detected by mRNA-seq. (left) Heatmap of differentially expressed mRNAs in Col-0 and *ndx1-4* seedlings detected by mRNA-seq. Statistical significance: $p < 0.05$. Colour heat is proportional to DeSeq2 log fold-change (LFC). (right) Distribution of differentially expressed TEs in transposon superfamilies. Retrotransposons (LTR/Copia, LTR/Gypsy, LINE1) represent more than 50% of the cases. We note that 20 TEs overlapped with protein-encoding ORFs and were not excluded from the analysis.

Based on the above results we conclude that NDX plays only a minor role in the transcriptional

reactivation of TEs (detectable by mRNA-seq), in contrast to its effect on siRNA expression (detectable by sRNA-seq). These new data have been incorporated in the revised manuscript emphasizing that NDX affects siRNA production and biogenesis pathways and not the reactivation of transposons.

At the request of Reviewer 2, we sought to find chromatin regulators that are differentially expressed in *ndx1-4*. We identified a group of genes that may play a direct or indirect role in the regulation of heterochromatin status. Overexpressed genes involve known RNAi factors such as 1) NRPD1b, NRPE3b, and NRPB/D/E9a, which represent the structural and regulatory subunits of Pol IV and Pol V, the core *trans* factors of RdDM; 2) RDR1, which participates in non-canonical RdDM⁵⁶; 3) IDN2, required for siRNA accumulation and binding to dsRNA and lncRNA⁵⁷; 4) AGO9, which is normally expressed in the ovule to interact with siRNAs transcribed from pericentromeric retrotransposons⁵⁸; 5) ROS1, which demethylates several genomic targets to restrict non-CG methylation activity⁵⁹. These RdDM genes are typically silent in seedlings (or expressed at low level) but become induced at high levels in the *ndx1-4* mutant. Downregulated chromatin factors include HTA4 (histone H2A), HON4 (linker histone like protein), and HMGB1 (high mobility group B1), which are involved in the assembly of nucleoprotein complexes. RT-qPCR validation of representative genes is shown in Supplementary Fig. 8 in the revised text. Misregulation of the above heterochromatin regulators are likely to contribute to the molecular phenotype of *ndx1-4*, however, causative relationships remain to be explored.

3. Many of the figures require additional description, clarity or controls. For example:
a. In Figure 1C, please show the no-tag ChIP track as a negative control.

The “no-tag” tracks have now been added to Figure 1c:

b. In Figure 3B-C, are the number of overlap between DRIP-seq R-loops and CHIP seq peaks expected by chance given datasets of this size, or is this an enrichment over the number expected by chance?

Yes, this is a statistically significant enrichment. The intersection of the two sets in the Venn diagram shows the number of overlapping DRIP and flag-NDX peaks. There are 326 DRIP peaks (out of 14124) that coincide with flag-NDX peaks, while 333 flag-NDX peaks (out of 2243) coincide with DRIP peaks. The two numbers are different since there are cases when multiple NDX peaks overlap with the same DRIP peak.

As the Reviewer noted, the number of overlapping peaks appears to be relatively small compared to the total amount of DRIP and CHIP peaks, however, this is still statistically significant over enrichment by chance ($*p < 0.001$, prop.test, >2-fold change). We modify the relevant paragraph in the revised text to emphasize this: *“This overlapping fraction is statistically significant compared to random enrichment ($*p < 0.001$, prop.test) and suggests that the NDX-stabilized R-loop model described for the FLC locus²² may still be true in some cases, though not in general.”*

c. In Figure 4D there is a close relationship to siRNA regions #6-9 and NDX binding. Please include a meta analysis of NDX binding (via the available CHIP data) to these regions of the genome such as the analysis in figure 3C. It will be important to determine if NDX binds all of these regions, or just a subset strongly.

The meta-analysis has been performed for sRNA groups #6-9 and, for comparison, for sRNA groups #1-5 (showing only weak association with NDX). sRNA clusters #6-9 were split into “arm” and “pericentromeric” loci according to their chromosomal position (**Revision Figure 4**). The analysis shows that NDX does not bind to sRNA #1-5 loci (as expected), nor does it bind to sRNA #6-9 loci located on chromosomal arms. In contrast, pericentromerically located sRNA #6-9 loci show a significant association with NDX. However, it is apparent that sRNA #6-9 loci exhibit heterogeneous binding affinity for NDX, as NDX binds strongly to a subset of these sRNA loci.

Revision Figure 4. Metaplot analysis of NDX ChIP signal distribution over sRNA loci 1-5 and 6-9. Rows represent the flag-NDX ChIP signal at the indicated genomic interval. The ChIP signal is scaled according to the scale shown on the right. sRNA loci 6-9 have been divided by chromosomal location as pericentromerically and arm associated. The NDX ChIP signal is preferentially enriched over pericentromerically located sRNA 6-9 loci, however, their binding affinity for NDX is heterogeneous.

The above meta-analysis has been incorporated in the the revised text (as

Figure 5c) to emphasize the heterogeneity of NDX association over sRNA loci #6-9:

“Nevertheless, NDX enrichment at class 6-9 sRNA loci was rather heterogeneous as NDX bound strongly to a subset of targets (Fig. 5c). Our metaanalysis shows that NDX does not bind to sRNA #1-5 loci (as expected), nor does it bind to sRNA #6-9 loci located on chromosomal arms, however, pericentromerically localized sRNA #6-9 loci bind significantly but heterogeneously to NDX (Fig. 5c).”

d. Figure 6B identifies regions that lose CHH/CHG methylation in *ndx* mutants AND bind NDX in wild-type. Please continue analyzing these regions to probe the mechanism of NDX activity. Particularly in Hi-C experiments and small RNA-seq experiments, what occurs at these regions that are likely the direct binding and action sites of NDX? More focus should be put on analyzing this “direct target” set in the manuscript.

We have supplemented our original ChIP peak-based DMR annotation (Revision Figure 5a; in the revised manuscript: Figure 8b) with a signal-based anchor plot analysis to show that NDX is enriched at or near the centre of hypo CHH and CHG DMRs (Revision Figure 5b). Then, hypo DMRs were categorized as ‘NDX-enriched’ and ‘non-enriched’ to assess possible sRNA expression changes (Revision Figure 5c) and chromatin architectural changes (Revision Figure 5d) at these direct and indirect targets of NDX activity. The results show that sRNA transcription changes detected at hypo CHH DMRs and hypo CHG DMRs are independent of NDX binding. This indicates an epistatic relationship between non-CG DNA methylation pathways and NDX such that the DNA methylase system functions independent of NDX binding to affect local siRNA transcription, and that the *ndx1-4* mutation plays an indirect role in siRNA production at hypomethylated CHH / CHGs. In contrast, NDX has a direct effect on chromatin compaction at hypo CHH DMRs (Revision Figure 5d) because NDX-enriched hypo-CHHs have a significantly higher compactness score than non-enriched hypo-CHH DMRs. This suggests that NDX binding at these sites directly promotes chromatin compaction, which is inversely changed in the *ndx1-4* mutant due to the absence of NDX (see Hi-C data). The latter data are in agreement with our Hi-C analysis showing that NDX promotes formation of intra- and interchromosomal interactions.

Revision Figure 5. Analysis of hypo CHH and CHG sites associated with NDX in wild type. a. Flag-NDX peak enrichment at DMRs. NDX is enriched over hypomethylated CHG and CHH regions. b. Anchor plot of flag-NDX ChIP signal over hypo CHH and CHG DMRs (red line) and random sites (blue, green, yellow lines). The NDX signal is enriched in the middle of the CHH DMRs and near the center of hypo CHG DMRs. c. sRNA expression changes at hypo CHH and CHG DMRs. Hypo CHH/CHG DMRs were classified as NDX enriched (blue) and non-enriched (red) based on their overlap with flag-NDX ChIP binding sites in Col-0 plants. Y axis represents sRNA-seq fold change (*ndx1-4* / Col-0). There is no statistically significant difference between NDX-enriched and non-enriched DMRs. d. Chromatin compactness changes of hypo CHH and CHG DMRs classified as NDX enriched (blue) and non-enriched (red). Y axis shows compactness scores calculated from Hi-C data. At hypo CHH DMRs there is statistically significant difference between NDX-enriched and non-enriched sites (** $p=0.0093$, Wilcoxon test).

DMRs there is statistically significant difference between NDX-enriched and non-enriched sites (** $p=0.0093$, Wilcoxon test).

The above data have been supplemented in the revised manuscript as Figure 8c and Supplementary Figure 11 and Supplementary Figure S16):

*“Since NDX binds directly only to CMT2 TEs (Fig. 4c), the above changes are likely due to the direct effect of NDX, whereas CHG methylation changes observed in RdDM TEs may be the indirect effect of *ndx1-4* mutation. Importantly, sRNA expression changes detected at hypo CHH/CHG DMRs appear to be independent of NDX binding, since there was no difference between hypo CHH/CHGs classified as “NDX-enriched” and “non-enriched” (Supplementary Fig. 11). This suggests that the *ndx1-4* mutation plays an indirect role in the production of siRNAs at hypomethylated CHH/CHG sites, and that the DNA methylase and RNAi systems can function without NDX binding (underscoring their epistatic relationship).”...*

*...“Nevertheless, chromatin compactness analysis⁶⁶ suggests that NDX has a direct effect on the condensation state of hypomethylated CHH regions (Supplementary Fig. 16) as NDX-enriched hypo-CHHs show significantly higher Hi-C compactness than hypo CHHs not bound by NDX (in wild-type plants). This indicates that NDX binding at these sites can directly promote chromatin condensation, which is inversely changed in the *ndx1-4* mutant due to the loss of NDX.”*

4. The analysis of transposons was not up to par for a paper that significantly focuses on their regulation. For example:

a. The authors describe how their ChIP overlaps with transposons from Fig 4A, but there is a lot more of this type of analysis that should be done. For example, which transposon types? All copies of that TE family, or only the ones in the centromere?

We appreciate the Reviewer’s suggestion that warrants further analysis of NDX association with transposons. We have performed an analysis on individual transposon families that showed significant

NDX-enrichment over a subset of DNA transposon families and retrotransposon families (**Revision Figure 6**). However, the above TE families are so diverse in transposition mechanism, internal structure, and chromosomal distribution that they prevent further functional interpretation of the results.

Revision Figure 6. DNA transposon and retrotransposon families enriched in NDX binding sites. Overlap ratios of observed NDX ChIP peaks, DRIP peaks and random peaks over transposon families. (Only TE families showing significant enrichment of ChIP or DRIP peaks are shown.)

We therefore performed a family-independent TE analysis on transposon families targeted by non-CG methylation pathways and compared TEs targeted by the RdDM system (RdDM TEs) to those controlled by the CMT2 pathway (CMT2 TEs) or both pathways (intermediate TEs) (Choi J *et al.* *eLIFE* 2021; **Revision Figure 7a**). These TE categories were further classified by chromosomal location as pericentromeric and arm-associated (**Revision Figure 7b**). The results show that NDX is primarily enriched at TEs regulated by the CMT2 pathway as well as the CMT2/RdDM common pathway and is depleted from RdDM-targeted TEs, regardless of pericentromeric or arm location. These functional relationships tie NDX to the CHH methylation pathway mediated by CMT2 and the common pathway regulated by CMT2 and RdDM.

Revision Figure 7. Enrichment of NDX binding sites over functional categories of TE families. Transposons targeted by the RdDM pathway (RdDM TEs) were compared to those controlled by the CMT2 pathway (CMT2 TEs) or both pathways (intermediate TEs). TE categories were further classified as pericentromeric and arm-associated by chromosomal location. Overlap ratios of observed NDX ChIP peaks and random peaks are shown over these transposon categories. n.a.: lack of statistical power due to very low peak count in the indicated category. NDX is enriched at CMT2 TEs and intermediate TEs, whereas it is depleted over RdDM TEs, regardless of pericentromeric or arm location.

b. It was not clear what “NAD” transposons are, or why they are important.

We apologize for not making this point clear enough. This is now made up for in the revised manuscript. *Arabidopsis* chromosomes form so-called nucleolus-associated domains (NADs) that are localized cytologically in the nucleolus (Pontvianne F *et al.* 2016. *Cell Rep*; Bersaglieri C *et al.* 2019. *Cells*). NADs are enriched in rDNA loci, telomeric regions, heterochromatic TEs (NAD TEs), and silent protein-coding genes sequestered in the nucleolus away from RNA Pol II activity. Studying NAD TEs by NGS techniques

is therefore potentially relevant since they can be used as a proxy to mark the nucleolus where NDX showed strong enrichment in our cytological measurements. Nucleolar enrichment of R-loop structures is also expected in *Arabidopsis* as well as in other species (Zeng C *et al.* 2021 *Mob DNA*), which warrants analysis of potential R-loop / NAD TE associations.

It seems that NAD transposons are enriched at both NDX ChIP sites and R-loops, but these two datasets do not overlap much. Please expand to clarify.

The association of NDX with NAD TEs does not differ from other TEs that are not related to the nucleolus, i.e. NDX binds equally to TEs located inside or outside the nucleolus. In contrast, R-loops show differential association with NAD TEs and NAD-independent TEs as the former group is enriched with DRIP peaks while the latter show depletion (Figure 4a in the revised manuscript). Thus, R-loops appear to associate with nucleolar TEs and have less preference for extranucleolar TEs. These associations are consistent with previous research that found high R-loop abundance in the nucleolus (Manzo SG *et al.* 2018. *Genome Biol*; Velichko AK, *et al.* 2019. *Nucleic Acids Res*; El Hage A *et al.* 2010. *Genes Dev*; Hegedüs É *et al.* 2018. *Nucleic Acids Res*; Szekvolgyi L *et al.* *PNAS* 2007).

Importantly, nucleolar TEs associated with NDX peaks or DRIP peaks are significantly colocalized with each other (**Revision Figure 8**), which suggests their functional interaction on a subset of nucleolar transposons. These relationships between NDX, R-loops, and nucleolar TEs are definitely worth exploring in follow-up experiments that we are planning to pursue.

Revision Figure 8. Overlap of NAD TEs associated with NDX and DRIP peaks. Left: observed overlaps; right: random (rnd) overlaps.

c. Most transposons in the pericentromere are not regulated by RdDM. Instead, they are regulated by the maintenance of CHG and CHH methylation via CMT3 and CMT2, respectively. The authors should split the transposons into their regulatory classes (Pol V/RdDM vs. CMT2/3) and determine if NDX functions on one or both of these classes. Since NDX functions on the pericentromere, I would predict that these are CMT3/2 regulated, and not RdDM.

We refer to our answer at *Point #4a* where transposons were classified according to their regulatory classes, as requested here. Based on this, NDX is enriched at TEs controlled by the CMT2 pathway as well as the CMT2/RdDM common pathway and is depleted from RdDM-targeted TEs, regardless of pericentromeric or arm location (**Revision Figure 7**). The revealed associations link NDX to the CHH methylation pathway mediated by CMT2 and the common pathway regulated by CMT2 and RdDM. Molecular changes at CMT2 TEs and RdDM TEs have been addressed in *Point #2* (**Revision Figure 2b**), where higher siRNA expression was detected in the *ndx1-4* mutant relative to Col-0 over CMT2 TEs and RdDM TEs. Nevertheless, CHH and CHG methylation levels were significantly decreased at CMT2 TEs

in the *ndx1-4* mutant, regardless of their pericentromeric or arm association (Figure 8f in the revised manuscript, see below), however, RdDM TEs showed reduced DNA methylation only in the CHG context located in chromosomal arms. (The mCG levels did not change in either TE class). Since NDX binds directly only to CMT2 TEs (Revision Figure 7), the above molecular changes are likely due to the direct effect of NDX binding to CMT2 TEs, while siRNA transcriptional and CHG methylation changes observed in RdDM TEs can be attributed to the indirect effect of the *ndx1-4* mutation.

The above associations and discussions have been supplemented into the revised manuscript.

Figure 8f in the revised manuscript. CMT2 TEs and RdDM TEs were classified by chromosomal location as pericentromeric and arm associated. Metaplots show average DNA methylation levels in CHH/CHG/CG contexts in Col-0 (black) and *ndx1-4* (red) backgrounds. CHH and CHG methylation is significantly decreased CMT2 TEs regardless of their arm- or pericentromeric location (mCG levels do not change). CHG methylation levels show reduction at RdDM TEs located in chromosome arms. (mCHH and mCG levels do not change.)

5. The Hi-C experiments do not separate the cause from consequence of *ndx* mutation. The authors argue that the molecular phenotype is caused by underlying chromatin changes, but this may be a consequence, rather than the cause.

The Reviewer is right: in our experiments we cannot separate the cause from consequence in terms of chromatin structural changes. We have now reworded the relevant sentence as follows: “Given the marked changes in siRNA transcription, R-loop formation, and DNA methylation in the *ndx1-4* mutant, we asked whether the observed molecular phenotype is related to underlying chromatin structural changes.”

6. The authors argue that the Hi-C changes are similar to those seen in other chromatin modifiers such as *met1* and *ddm1*. However, the molecular changes in *met1* and *ddm1* are significantly more severe compared to *ndx*. For example, the CHH methylation loss in *ndx* is very subtle (Fig 6E). Therefore, the authors need to directly compare the Hi-C changes, expression changes and methylation changes to those mutants to make such comparisons. It is likely that the authors will find that *ndx* has similar patterns, but at a much lower strength compared to *met1* and *ddm1*. I recommend a direct data comparison whenever the authors wish to compare them in the Results or Discussion section.

This is an interesting and important point that we intend to address. And we have indeed found that *ndx1-4* has a similar pattern to *met1* and *ddm1*, but with less strength. We analyzed published *met1* and *ddm1* specific Hi-C data from 10-day old seedlings and directly compared them to our *ndx1-4* Hi-C (a *clf28 swn7* Hi-C was also included in the analysis at the request of Reviewer 2). The results show that the Hi-C pattern of the *ndx1-4* mutant was similar to that of the *met1* and *ddm1* mutants, however, overall interaction frequency changes were milder in *ndx1-4* (**Revision Figure 9a**). Several new regions joined the KNOT structure in all three mutants that do not appear in Col-0 (e.g. the Chr1: ~ 8.8Mb region, shown in **Revision Figure 9b**); however, in the *ddm1-2* and *met1-3* mutants additional regions also interacted with the KNOT that are not present in *ndx1-4* (**Revision Figure 9c**). Therefore, chromatin structure changes observed in *ddm1-2*, and *met1-3* are more severe than in *ndx1-4*. The *clf28 swn7* mutant is expected to lack all H3K27me3 modification as CLF and SWN both control H3K27me3 (Lafos et al., 2011). Previous Hi-C analysis showed that spatial interactions of H3K27me3-enriched minidomains were reduced in the *clf-28 swn-7*, but no Hi-C change was found in the KNOT structure (Feng et al Mol Cell 2014). Our analysis gave similar results as the Hi-C pattern of the *clf28 swn7* mutant was different from all three mutants and the *de novo* KNOT interactions were completely missing from *clf28 swn7*. Therefore, in terms of genome organization, *clf swn* operates in a different pathway than *ddm1*, *met1*, and *ndx1-4*, which all showed highly similar KNOT interaction patterns. These associations further reinforce the link between NDX and DNA methylation pathways. The above data have been incorporated in the revised manuscript.

Revision Figure 9. Loss of NDX causes similar genome structural changes and DNA methylation mutants. rearrangement of chromatin organization. Differential Hi-C interaction matrix of *ndx1-4*, *ddm1-2*, *met1-3*, and *clf28 swn7* mutants versus Col-0. Red and blue colours show enrichment or depletion of Hi-C interactions in the indicated mutant, respectively. White colour indicates no change between mutant and Col-0. Lower panel: Regions showing statistically significant differences in Hi-C interactions identified by the Slitherine algorithm. Hi-C pattern of the *ndx1-4*, *ddm1-2*, and *met1-3* is highly similar, however, *ddm1-2*, and *met1-3* show stronger chromatin structure changes compared to *ndx1-4* as additional regions interact with the KNOT that are not

present in *ndx1-4*. Hi-C pattern of the *clf28 swn7* mutant appears to be different from all three mutants (e.g. de novo KNOT contacts are missing). b-c Representative KNOT contacts in *ddm1-2*, *met1-3*, *clf28 swn7*, and *ndx1-4* mutants. Panel a shows de novo formed KNOT contacts in the *ddm1-2*, *met1-3*, and *ndx1-4* mutants that are missing from Col-0 and *clf28 swn7*. Panel b shows *ddm1-2* and *met1-3* specific KNOT contact that are missing from *ndx1-4*, *clf28 swn7*, and Col-0.

In terms of DNA methylation changes, we compared the DNA methylome of *ndx1-4* plants to *met1* and *ddm1* supplemented with a large collection of BS-seq data obtained from DNA methylation mutants functioning in different pathways. Using the hcDMR pipeline (Zhang et al. 2018 PNAS), we compared these BS-seq libraries with more than 50 high-quality Col-0 controls and defined high-confidence differentially methylated regions (hcDMRs) in the CHH context. HcDMRs were then clustered with the S-MOD method (statistical measurement of overlapping of DMRs), allowing the identification of hierarchical relationships between *ndx1-4* and DNA methylation mutants (Revision Figure 10). Our analysis confirmed well-known functional relationships between the mutants (e.g., those operating in the RdDM and CMT2 pathways) and also revealed associations between *ndx1-4*, *drm2*, and *met1* for CHH methylation as they were grouped in a similar cluster (*ddm1* was not included in the latter group). However, we could not clearly determine if NDX influences non-CG methylation via the CMT2 or RdDM pathways, or, more possibly, by a less characterized independent pathway (e.g. Knot linked silencing). As DNA methylation pathways show significant redundancies, appropriate classification is not obvious in terms of NDX function. For instance, MET1 (the main CG methylase) and DRM2 (the main RdDM methylase) are both required to maintain CHH methylation in CMT2 targeted heterochromatin. The potential crosstalk between DRM2, MET1, and NDX remains to be explored by combining classical and molecular genetics.

Revision Figure 10. Clustering of overlapping hypo CHH DMRs identified by BS-seq in different DNA methylation mutants. The *ndx1-4* mutant (highlighted by red lines) clusters with the *drm2*, and *met1* mutants for CHH methylation (highlighted by brown lines).

Regarding the analysis of potential small RNA expression changes in the *met1*, *ddm1*, and *ndx1-4* mutants, no sRNA-seq data were found in mutant seedlings, which unfortunately prevented comparison of sRNA profiles.

Minor changes to be fixed

1. In the abstract it is claimed that Hi-C determines chromatin compaction. This is incorrect and should be fixed.

The phrase referring to chromatin compaction has been reworded in the abstract. (We note that the abstract had to be shortened to less than 150 words due to formatting requirements of *Nature Communications*)

2. The Introduction is lengthy, and in particular the length of the section on H3K27 should be reduced, as this is not assayed or really discussed beyond the introduction in the manuscript.

We reduced the length of the Introduction by removing a few sentences describing the role of H3K27 methylation. However, we believe that this section should not be completely omitted, as the H3K27me modification underpins the interaction between PRC1 and NDX, and the revised manuscript has now been supplemented with new *clf28 swi7* mutant Hi-C data that affects H3K27me3, which needs to be introduced.

3. Clear introduction and conclusion sentences to paragraph should be added to help guide the reader. For example, starting off a paragraph in the introduction “This possibility seems realistic given...” does not establish what “this” refers to.

We apologize for our incomplete wording. We have corrected the revised text as follows: “*The possibility of NDX-mediated genome structural changes seems realistic given that in animals PRC1/RING1B i) causes chromatin compaction through its non-enzymatic function^{25,26}, ii) mediates long-range promoter-promoter interactions between developmentally regulated genes²⁷, and iii) orchestrates oestrogen-induced enhancer-promoter looping interactions²⁸.*”

4. Please clarify if the FLAG- and GFP- tagged versions of NDX complement the mutant phenotype before using them for ChIP.

The flag-NDX and NDX-GFP constructs fully complement the *ndx* mutant phenotype, as shown in the paper of Sun Q *et al.* (Science 2013). The authors showed that transformation of FLAG-AtNDX (N-terminal tag fusion), or AtNDX-FLAG (C-terminal tag fusion), or AtNDX-eGFP (C-terminal fusion) complemented the molecular phenotype of an *ndx* mutant, and that defects in COOLAIR ncRNA expression at the *FLC* locus was rescued by expressing the tagged version of the NDX protein (see Supplementary Figures S3, S6, and S7 therein). All plant material used in our study was obtained from the Dean’s lab who had thoroughly characterized and validated the mutants in their landmark paper as references above.

5. For the DRIP sequencing, it was not immediately clear whether the authors had performed this experiment or downloaded the data from a previously-described work. Either way, please clarify if the DRIP-seq was performed in the same tissue type and developmental stage compared to the ChIP. This is important because they would be expected to overlap only if from the same tissue.

In our study a new DRIP-seq experiment was performed in 10-day old *Arabidopsis thaliana* seedlings, in the same tissue type and developmental stage as NDX ChIP-seq. Col-0 and *ndx1-4* samples were examined with or without RNaseH treatment. The DRIP protocol is described in detail in the *Methods* section.

6. It was unclear why siRNA1003 is used in Fig. 6? Is this a transposon, and if so, what type?

The siRNA1003 is not a transposon but a small RNA derived from the intergenic spacers (IGSs) of tandemly repeated 5S rRNA genes whose expression is controlled by the RdDM pathway (Blevins T *et al.* 2009. PLoS One; Douet J & Tourmente S. 2007. Heredity; Xie Z *et al.* 2004. PLoS Biol). We sincerely apologize for not going into more detail about this. The *Arabidopsis* genome contains about 1000 copies of 5S rRNA loci arranged in tandem arrays at the pericentromere of chromosomes 3, 4 and 5. The siRNA1003 probe used in our northern blot experiment represents the silenced 5S rDNA array located at the pericentromere of chr3 (cytologically at the chromocenter), just where NDX is preferentially enriched according to our ChIP-seq analysis. In addition, inactive 5S rRNA genes tend to be sequestered in heterochromatin at the periphery of the nucleolus (Layat E *et al.* 2012. Plant Cell Physiol), a compartment that also shows NDX positivity in our microscopic measurements (in line with Sun Q *et al.* 2013. Science). In addition, our new ChIP-seq analysis – performed at the request of Reviewer 2 – identified a preferential colocalization between the peripheral heterochromatin marker Nucleoporin 1 (NUP1) protein and NDX, further reinforcing the functional link between NDX and peripheral heterochromatin.

These associations make si1003 relevant for studying heterochromatin transcription in relation to NDX function. The northern blot result in Figure 9d in the revised manuscript shows that siRNA1003 expression is upregulated in the pericentromeric region of chromosome 3 where NDX is highly enriched and most non-CG hypomethylation occurs in the *ndx1-4* mutant. Therefore, NDX not only affects the sRNA expression profile of transposons, but also the activity of 5S rRNA genes located in pericentromeric regions. In the revised manuscript, we better emphasize the goal of studying siRNA1003 expression and its relation to NDX function, pericentromeric heterochromatin transcription, and DNA methylation.

MS text:

“Interestingly, the pericentromeric ncRNA siRNA1003, expressed from the silenced 5S rDNA array on chr3 (cytologically in the chromocenter), also showed increased siRNA levels by northern blot hybridization (Supplementary Fig. 4). This suggests that NDX not only affects the sRNA expression profile of transposons but also silences other gene types in pericentromeric regions.”

7. In Fig 6G, it is unclear what the browser shots refer to? The Results section says transcript levels, but if this is RNA-seq data, a lot more needs to be done with this data in this manuscript. The production of RNA-seq is not described in the methods section. See major point #2.

The original JBrowse snapshot has been supplemented with positional markers and landmarks to help the reader better identify the region. Description of the small RNA-sequencing (sRNA-seq) experiment was included in the original manuscript in the Methods section, which is now supplemented with the description of new mRNA sequencing (RNA-seq) and rt-qPCR measurements (also in the Methods section). See also our answers to major point #2 where additional genome-wide sRNA-seq and mRNA-seq analyses have been done.

8. Figure 1D leaves out knobs, NORs and other important features. In addition, it would be very interesting to introduce and bring in the different chromatin states into Figure 1D for overlap with their ChIP data.

The Knob region has been added to the annotation categories of Figure 5a. The result shows that NDX is preferentially enriched in constitutive heterochromatin regardless of centromere-proximal or arm association of this chromatin type. Regarding NORs, there is currently no assembled NOR-reference sequence to be used in our analysis. Sims *et al.* have recently sequenced about 400 units from NOR2 using a BAC-based strategy (<https://www.nature.com/articles/s41467-020-20728-6>) however, the identified contigs have not been assembled to a reference sequence. The exact length and nucleotide composition of NOR2 and NOR4 remain unknown, which unfortunately prevents an estimate of the extent of their NDX association.

9. Figure 2B is difficult to follow. I suggest putting different panel letters on the 5 different display items that are currently all labelled as “B”.

Now we use different display items to label each panel in Figure 2b.

10. Since much of the DRIP-seq data is negative with respect to overlap with NDX, I suggest confining the DRIP-seq analysis to a single main figure, rather than having it in several figures.

The current layout follows the logic and structure of the manuscript, in which we first compare the NDX ChIP and DRIP data, and then compare the Col-0 and *ndx1-4* mutant-specific DRIP profiles. It would be difficult to cram all these data into a single figure (and the text also needed to be significantly restructured), which is why we prefer to keep the original layout.

11. In Figure 4E, it was unclear to which category from part D each peak corresponds to.

Figure 4E has become Figure 5d in the revised text. The peaks have been connected with their sRNA classes, and the type of transposons with which they overlap was also indicated.

12. In all browser shots in figures, please add the surrounding annotation or something that helps the reader understand what type of a region of the genome this is. Currently they lack this information content. For example, in Figure 5C&H, what are these regions of the genome that make small RNAs?

All genome browser snapshots have been supplemented with landmarks and annotations to help the reader become more aware of the chromosomal region in question.

13. Figure 5A needs statistical testing

Figure 5a has become Figure 6a in the revised text. Statistical testing has now been added.

14. Hi-C data should be confined to one main figure plus other supplementals as needed.

Hi-C data are now shown in a single main figure (Figure 10) and four supplemental figures (Supplementary Figures 13, 14, 16, and 17).

Reviewer #2

In this study the authors explore the function of the transcription factor NDX that has previously been identified to play a role in ABA signaling and to directly interact with the PRC1 components RING1A and B (Zhu, Plant Cell 2020). NDX has also been reported to be involved in stabilizing R-loop structures at the FLC locus (Sun, Science, 2013). In contrast to these initial observations that could suggest a function in gene regulation, the authors find here by ChIP-seq that NDX is particularly enriched in pericentromeric heterochromatin and its binding sites are rather anti-correlated with positions of R-loops identified by DRIP-seq. Instead NDX is enriched at regions enriched in chromatin states 8/9 and at class 6-9 sRNA loci involved in RdDM. In *ndx* mutants, siRNA loci of class 4-9 and some transposons are upregulated. The authors also determine DNA methylation changes in the mutant by BS-seq and find hypo CHH/CHG sites co-localizing with NDX. Finally they show that by Hi-C that loss of NDX leads to more interchromosomal contacts.

This study brings new insights on the role of NDX in pericentromeric heterochromatin and refutes a more general genome-wide role in R-loop stabilization. It further establishes a link with siRNA production, DNA methylation and transposon control as well as higher-order genome organization. While this is an overall thorough study including several genome wide analyses, bringing important new insight, it remains vague how NDX affects these different chromatin features.

We sincerely thank the Reviewer for the positive and constructive feedback. As proposed, we have added new experiments and analyses that extended and confirmed our major conclusions. Specifically, we performed new small RNA-seq analysis to show that siRNA production is globally affected in the absence of NDX; we performed a new mRNA-seq experiment to identify chromatin regulators and siRNA biogenesis factors with differential expression in *ndx1-4* mutants; by rt-qPCR we validated a number of hits from the above mRNA-seq analysis; we complemented our Hi-C analysis with new confocal laser scanning microscopic measurements in Col-0 and *ndx1-4* seedling in order to compare DAPI-stained chromatin in single nuclei. We felt that these experiments would have a chance to close

this topic rather than open up new ones, as detailed below. Please note that we have considered *Nature Communications'* formatting guidelines (<https://www.nature.com/documents/ncomms-formatting-instructions.pdf>) and modified the original figures and text accordingly. The revised version contains 9 figures and 18 supplementary figures, which extend and confirm our major conclusions about the role of NDX protein.

Do the authors suggest, as the phenotypes are unlikely to involve R-loops, that the interaction with the PRC1 complex plays an important role? Indeed, several studies have suggested an ancestral role of Polycomb in transposon regulation (reviewed in D el eris, Trends in Genetics, 2021). Is a similar reactivation of transposons / siRNAs observed in *ring1a/ring1b* mutants?

Unfortunately, no sRNA-seq data are available in *ring1a/ring1b* seedlings, hence we could not compare them with *ndx1-4*-specific small RNA changes from transposons. However, based on our current knowledge, the role of RING1A/RING1B seems unlikely as PRC1 does not affect sRNA biogenesis. In this regards, it is noteworthy that the number of target genes coregulated by NDX and RING1A/RING1B is small (based on the mRNA-seq data of Zhu et al., Plant Cell 2020), which suggests that multiple and partially overlapping epigenetic mechanisms are used to control the targets of NDX and RING1A/RING1B activity.

The siRNA expression changes observed in *ndx1-4* plants have been further reinforced by new genome-wide sRNA-seq analysis incorporated in the revised manuscript. The analysis shows a clear increase in transposon-derived siRNA levels in the absence of NDX, which is independent of pericentromeric, Knob, or arm association of TEs (**Revision Figure 11a**). We further observe a significant increase in siRNA expression over different functional categories of TEs (i.e. RdDM TEs and CMT2 TEs; **Revision Figure 11b**) such that higher expression is detected in the *ndx1-4* mutant relative to Col-0 in each case (*ndx1-4* vs. Col-0 comparison). (We note that siRNA levels from RdDM TEs were substantially higher than those from CMT2 TE both in Col-0 and *ndx1-4* backgrounds, which is expected from the known molecular nature of sRNA regulation by CMT and RdDM pathways, in agreement with published data (e.g. He L et al. (2021) Proc Natl Acad Sci U S A). Together, the above global changes indicate that most transposons are associated with increased siRNA expression in *ndx1-4* plants. The observed transcriptional changes were confirmed by stem-loop qRT-PCR at two representative transposons (LTR/Copia28 and DNA/MULE1; **Revision Figure 11c**), validating our sRNA-seq results.

Revision Figure 11. Global increase of siRNA-levels in the absence of NDX function. (a) The ratio of aligned siRNA reads (*ndx1-4* / Col-0) is significantly increased in pericentromeric TEs, Knob TEs and chromosome arm TEs relative to “negative control” regions (tRNA and rRNA genes) and the genome-wide average value (horizontal red line). Statistical significance: ****p*<0.0001 (Wilcoxon rank sum test). (b) The number of aligned sRNA reads

show a significant increase in the *ndx1-4* mutant over the functional categories of TEs (RdDM TEs vs. CTM2 TEs). sRNA read counts were normalized to TE length. The siRNA levels from RdDM loci are also significantly higher than those from CMT2-only loci (both in Col-0 and *ndx1-4*), in line with published sRNA-seq data (e.g. He L *et al.* (2021) Proc Natl Acad Sci U S A). Statistical significance: * $p < 0.05$ (Wilcoxon rank sum test).

MS text: “We observed a significant increase in siRNA levels over RdDM TEs and CTM2 TEs such that higher expression was detected in the *ndx1-4* mutant relative to Col-0 (Fig. 6f; *ndx1-4* vs. Col-0 comparison). (We note that siRNA levels from RdDM TEs were substantially higher than those from CMT2 TE both in Col-0 and *ndx1-4* backgrounds, which is expected from the known mechanism of sRNA control, in agreement with published data⁵⁵.) Together, the above global changes indicate that many transposons are associated with increased sRNA expression in *ndx1-4* plants. The above siRNA changes were confirmed by stem-loop rt-qPCR at two representative transposons (Fig. 6g), suggesting that NDX is involved in transcriptional silencing of these loci.”

Are any other chromatin regulators among the genes misregulated by loss of NDX?

We performed a new mRNA-seq experiment in Col-0 and *ndx1-4* seedlings to identify potential chromatin regulators misregulated by the loss of NDX. The data recapitulate the mRNA-seq results published by Zhu *et al.* (Revision Figure 12) and are incorporated in the revised manuscript.

[redacted]

Revision Figure 12. Correlation of gene expression changes between two *ndx1-4* mRNA-seq datasets. Y axis: differentially expressed genes (DEGs) identified in the Zhu *et al.* dataset (Plant Cell 2020); *ndx1-4*/Col-0 log₂ fold changes. X axis: DEGs identified in the current dataset (Karanyi *et al.*); *ndx1-4*/Col-0 log₂ fold changes. The two datasets are significantly correlated (Pearson R = 0.81).

Our mRNA-seq analysis identified 1984 differentially expressed genes ($p < 0.01$) involving 864 up- and 1120 downregulated genes (Supplementary Table S8-9 and Supplementary Figures S7-9). GO term enrichment (AgriGO) analysis showed that upregulated genes were primarily involved in the formation of ribonucleotide complexes, cold-response processes and flower organ development, while downregulated genes were implicated in general stress response, post-embryonic development (e.g. seed formation), lipid storage/localization, and binding of RNA nucleotides. Importantly, we managed to identify chromatin regulators with differential expression (Revision Figure 13a-b) that might play a direct or indirect role in the observed siRNA changes in the *ndx1-4* mutant. The overexpressed genes involve known RNAi factors such as 1) NRPD1B, 2) NRPE3B, and 3) NRPB9A, representing the catalytic and structural subunits of RNA polymerase Pol V that is central to transcriptional gene silencing (NRPB9A is a common to Pol II, Pol IV and Pol V RNA polymerases; J. Gallego-Bartolomé *et al.*, Cell 2019; R. Haag *et al.*, Cold Spring Harb. Symp. Quant. Biol 2013); 4) RDR1, which governs siRNA-mediated antiviral gene silencing and participates in non-canonical TGS/RdDM processes (X.B. Wang, Proc. Natl. Acad. Sci. U. S. A. 2010; Cuerda-Gil, Nat. Plants. 2 2016); 5) IDN2, which is required for siRNA accumulation and binds to dsRNA and lncRNA (G. Böhmendorfer *et al.*, Plant J. 2014); 6) AGO9, which is normally expressed in the ovule to interact with 24 nt sRNAs produced by pericentromeric-located retrotransposons (V. Olmedo-Monfil *et al.*, Nature. 2010); 7) ROS1, which demethylates certain target gene promoters to restrict the activity

of TGS/RdDM (Z. Gong, T et al Cell 2002). These RNAi genes are typically silent in seedlings (or expressed in very small amounts) but become induced at high levels in the *ndx1-4* mutant. Downregulated genes include HTA4 (histone H2A), HON4 (linker histone like protein), and HMGB1 (high mobility group B1), which are involved in the assembly of nucleoprotein complexes. Rt-qPCR validation of some representative genes are shown in **Revision Figure 13c**. Misregulation of the above heterochromatin regulators are likely to contribute to the molecular phenotype of the *ndx1-4* mutant, however, causative relationships remain to be explored.

Revision Figure 13. Chromatin regulators that show differential expression in the absence of NDX. a-b: genome browser snapshots of upregulated (a) and downregulated (b) chromatin regulator genes. c. rt-qPCR validation of gene expression changed shown in panels a-b. mRNA levels were normalized with UBQ10 expression levels.

How do the authors envisage that NDX would control HC compaction?

We envisage that most molecular changes observed in the *ndx1-4* mutant are indirect consequences of the mutation. MS text: “Based on our data, the following model is proposed to summarize the function of NDX in heterochromatin homeostasis: (i) inactivation of NDX results in chromatin decompaction at highly condensed pericentromeric regions (by an as yet unknown mechanism), (ii) relaxed heterochromatin structure leads to het-siRNA accumulation and DNA methylation changes, (iii) consequently, a subset of chromatin regulatory genes and transposons become derepressed or repressed. Increased global R-loop formation is likely due to the collateral effect of NDX loss and chromatin conformational changes. Whether the NDX-mediated nuclear chromatin organization is

related to the CMT2 pathway (as may be suggested by some of our observation, Fig. 4c, 8f) or represents a completely independent route, needs to be further investigated. Genetic analysis of the above and other factors is expected to lead to a better understanding of heterochromatin homeostasis as the driver of genome organization and stability and its response to developmental signals and environmental stimuli.”

Can an NDX overexpressor rescue HC decompaction in other mutants such as *svuh456* or *atxr56*?

Overexpression of NDX alone does not cause a phenotype without ABA treatment (Zhu et al. 2020), suggesting that only a tiny amount of NDX is needed for its full functioning in the ABA-pathway. However, NDX overexpression in a *svuh456* triple mutant or *atxr56* double mutant background is likely to cause a severe chromatin phenotype. We are definitely interested in pursuing these experiments, however, mutant constructions followed by Hi-C analysis require a significant time investment (up to 1-2 years) and we feel that this workload is currently beyond the scope of our manuscript.

More specific comments:

Figure 2. I am a bit surprised that the authors look only in root cell nuclei, which are not the perfect cell type to investigate pericentromeric heterochromatin. Root cells have a big nucleolus and the chromocenters are not as well visible compared to other tissues. It would be good to see NDX-GFP enrichment also in other cell types - as here a clear enrichment in heterochromatin as would suggest the ChIP-seq results is not visible.

According to the Reviewer’s suggestion, we tried to recapitulate the GFP-NDX measurements in cotyledons (from 10-day-old seedling), however, only autofluorescent signal was detected that was presumably associated with the strong chlorophyll expression of this tissue type (**Revision Figure 14**). Chlorophyll-containing tissues therefore do not appear to be suitable for immunofluorescent analysis of NDX-GFP distribution. Nevertheless, we refer to a previous histochemical detection of NDX in cell types other than root cells by GUS staining (Sun Q et al 2013). In this method, a specific GUS-NDX staining pattern was detected in addition to the root cell nuclei, e.g. in embryos, young leaves, and flower buds, suggesting that NDX is ubiquitously expressed in dividing tissues. Our ChIP-seq experiment performed in seedlings represents the cumulative sum of NDX distribution in the above heterogenous tissue types revealing a clear enrichment in pericentromeric heterochromatin regions.

Revision Figure 14. NDX-GFP fluorescence cannot be assessed in cotyledons. Autofluorescence signal of chlorophyll from 10-day old cotyledons (NDX-GFP plant). The plant material was excited with the 488-nm line of argon-krypton laser and emission was detected through a 500-550-nm band-pass filter (left panel), chlorophyll autofluorescence is pseudo-coloured (middle). Plant material was excited with the 633-nm line laser and emission was detected through a 645-745-nm band-pass filter (right panel). Merged image.

From the ChIP-seq data, it seems that NDX is enriched in pericentromeric HC, but in the conclusion they suggest that it is closely associated with peripheral heterochromatin. Are they suggesting also silent euchromatin domains associated with the nuclear lamina? They could also compare their enrichment data with RE-ChIP data from Bi et al, where they have been looking at NUCLEOPORIN1 (NUP1) enriched chromatin.

Pericentromeric heterochromatin is usually located near the nuclear periphery in *Arabidopsis*. Since NDX shows a strong association with pericentric heterochromatin, lamina-enrichment and peripheral nuclear localization is expected, according to the Reviewer's assumption. We have checked the overlap of NUCLEOPORIN1 (NUP1)-enriched chromatin with the identified NDX binding sites and found a statistically significant overlap between the two features (**Revision Figure 15**; Supplementary Figure S3). A significant fraction of NDX peaks were located preferentially at the nuclear periphery, and *vice versa*, NUP1-enriched chromatin at the nuclear periphery preferentially coincided with NDX. These associations are in line with our microscopic observations suggesting that peripheral heterochromatin is enriched for NDX.

Revision Figure 15. Peripheral heterochromatin marker NUP1 coincides with NDX binding sites. 77% of NUP1 peaks colocalize with NDX, while 37% of NDX peaks overlap with NUP1. Both associations are significant compared to random associations (prop.test, *p<0.0001).

MS text: "peripheral nuclear localization was consistently observed and was further reinforced by the overlap between NDX binding sites and NUCLEOPORIN1 (NUP1)-enriched chromatin³³, which marks nuclear periphery (Supplementary Fig. 2). The NDX signal was weak in the nuclear interior, which typically coincides with euchromatin."

I would strongly suggest to also investigating NDX localization by immunofluorescence on the Flag-tagged version of the NDX protein. Often tags will alter the localization and the presence of a big fraction of NDX in the nucleolus might point in this direction. It is not clear whether the NDX nucleolus fraction is a functional one or just storage of non-functional protein? Can the authors show that it is also present in the Flag-NDX line? What do the authors suggest NDX binds to in the nucleolus, as there is no chromatin except active rDNA copies? Does it fit with their conclusion of a chromatin-binding factor?

In our microscopic measurements about 5-20% of cells showed NDX-GFP accumulation in the nucleolus while 80-95% were negative for nucleolar staining (this is now better emphasized in the revised text). Our observations fully recapitulate the results of Sun Q *et al.* (Science 2013), who detected nucleolar localization of NDX-GFP and showed that this was specific for the NDX protein (see a copy of the original figure below).

Sun Q *et al.* (Science 2013) - Figure S8/D. “atNDX protein is localized in the nucleus and nucleolus. GFP signals can be detected in nucleolus from AtNDX-GFP plants (arrowheads in a) but not VRN1-GFP plants (b)”

[redacted]

Nevertheless, we attempted to perform the suggested flag-NDX immunostaining on formalin-fixed permeabilized flag-NDX seedlings, but unfortunately failed to achieve a specific flag-NDX signal (Revision Figure 16). Although different internal labelling and permeabilization protocols were tested, our antibodies apparently did not properly enter the nuclei and / or the sample was degraded during the procedure. (In informal personal communications with a co-author of the Science paper, we learned that they also failed to perform successful microscopic labelling for the flag epitope.)

Revision Figure 16. Immunofluorescent detection of flag-NDX in formalin-fixed permeabilized flag-NDX seedlings. We failed to detect specific flag-NDX nuclear staining using anti-Flag-Alexa488 antibodies, as the fluorescent signal (green) accumulated outside the cells.

The above harsh conditions for sample preparation and labelling were very different from the native conditions for NDX-GFP detection in live cells, which is expected to better reflect the *in vivo* state of NDX distribution. Two independent laboratories obtained the same result for the nucleolar association of NDX-GFP, which in our opinion supports the reliability of our NDX-GFP data. In addition, the NDX-GFP distribution in our ChIP-seq experiment was consistent with the genomic distribution of flag-NDX, further validating our results on the chromosomal association of NDX-GFP. Regarding the relationship between rDNA copies and NDX, only a small fraction of rDNA units is transcriptionally active while most are inactive, so this does not contradict the repressive role of NDX in the nucleolus. Our fluorescence FCS and FRAP measurements showed similar kinetics between nuclear and nucleolar NDX, suggesting that these compartments have closely related NDX-binding properties. Whether the nucleolar fraction of NDX represents functional and/or non-functional (storage depot) populations is currently unclear and remains to be explored in further experiments. We supplemented the text as follows: “Nucleolar staining was visible in less than 20% of cells and it is currently unclear whether this fraction represents a functional or non-functional population that acts as a ‘storage depot’.”

Figure 6: How does the altered interactome in *ndx1-4* correlate with changes observed in *ref6* (Huang, Genome Research, 2020)? Can the authors show a more detailed comparison between *ndx* and *clf* or *swn* interactome?

We have analyzed *clf28 swn7* specific Hi-C data from 10-day old seedlings and directly compared them to our *ndx1-4* Hi-C (*met1* and *ddm1* Hi-C were also included in the analysis at the request of Reviewer 1). The *clf28 swn7* mutant is expected to lack all H3K27me3 modification as CLF and SWN both control H3K27me3 (Lafos *et al.*, 2011). Previous Hi-C analysis showed that spatial interactions of H3K27me3-

enriched minidomains were reduced in the *clf28 swn7*, but no Hi-C change was found in the KNOT structure (Feng *et al* Mol Cell 2014). Our analysis also showed that the Hi-C pattern of the *clf28 swn7* mutant was different from the *ndx1-4*, *met1* and *ddm1* mutants, which showed highly similar interaction patterns (**Revision Figure 17a**). Several new regions joined the KNOT structure in *ndx1-4*, *met1* and *ddm1* mutants that do not appear in Col-0 (e.g. the Chr1: ~ 8.8Mb region, shown in **Revision Figure 17b**); however, these *de novo* KNOT interactions were completely missing from *clf28 swn7*. Therefore, in terms of genome organization, *clf swn* operates in a different pathway than *ddm1*, *met1*, and *ndx1-4*, which all showed highly similar KNOT interaction patterns. These associations further reinforce the link between NDX and DNA methylation pathways. The above data have been incorporated in the revised manuscript.

Revision Figure 17. Loss of NDX causes similar genome structural changes as DNA methylation mutants. Differential Hi-C interaction matrix of *ndx1-4*, *ddm1-2*, *met1-3*, and *clf28 swn7* mutants versus Col-0. Red and blue colours show enrichment or depletion of Hi-C interactions in the indicated mutant, respectively. White colour indicates no change between mutant and Col-0. Lower panel: Regions showing statistically significant differences in Hi-C interactions identified by the Slitherine algorithm. Hi-C pattern of the *ndx1-4*, *ddm1-2*, and *met1-3* is highly similar, however, *ddm1-2*, and *met1-3* show stronger chromatin structure changes compared to *ndx1-4* as additional regions interact with the KNOT that are not present in *ndx1-4*. Hi-C pattern of the *clf28 swn7* mutant appears to be different from all three mutants (e.g. *de novo* KNOT contacts are missing). b-c Representative KNOT contacts in *ddm1-2*, *met1-3*, *clf28 swn7*, and *ndx1-4* mutants. Panel a shows *de novo* formed KNOT contacts in the *ddm1-2*, *met1-3*, and *ndx1-4* mutants that are missing from Col-0 and *clf28 swn7*. Panel b shows *ddm1-2* and *met1-3* specific KNOT contact that are missing from *ndx1-4*, *clf28 swn7*, and Col-0.

Can the authors also look at heterochromatin organization by an alternative approach? E.g. a FISH study of 180bp repeats to visualize changes in the organization of chromocenters?

We appreciate the Reviewer's proposal about the complementary application of FISH and Hi-C methods to study heterochromatin organization. FISH highlights 3D genome architecture within single cells (as opposed to the cell population based Hi-C approach), which can be useful to reach a balanced interpretation of Hi-C data. However, the FISH method applies an extremely harsh alkaline treatment and lysis condition as well as chaotropic agents (such as formamide) on the samples to allow access to the DNA strands for fluorescent probes. In addition, formalin crosslinks must first be eliminated in the FISH protocol, which is the most important point in proximity ligation methods. These are significant technical differences from the Hi-C protocol, which can artificially alter the nucleus and chromatin structure. Indeed, swelling and dispersal of chromatin were documented during FISH that lead to discordant results compared to Hi-C (e.g. Benabdallah, N. S. *et al.* Decreased enhancer–promoter proximity accompanying enhancer activation. *Mol. Cell* 2019).

Although the Reviewer raises an important point and we are planning to implement orthogonal approaches to validate our Hi-C, we feel that developing a reliable *in vivo* FISH protocol (that preserves native 3D genome organization) would involve a significant time investment that goes beyond the scope of the current study. Nevertheless, we have performed a microscopic measurement on Col-0 and *ndx1-4* samples to compare the physical parameters of nuclei (**Revision Figure 18**) and observed a significant increase in the size of DAPI-stained chromatin in *ndx1-4* nuclei. This suggests global relaxation of heterochromatin in the absence of NDX, consistent with our Hi-C results, which show an overall decrease in intra-chromosomal interactions (i.e. chromatin decompaction) within pericentromeric regions. We believe that 3D genome structures measured by Hi-C at tens of thousands of loci, supplemented by the above orthogonal microscopic approach, have the statistical power to make our conclusion reliable. The new microscopic data have been added to the revised manuscript as Supplementary Figure S15.

Revision Figure 18. Comparison of the size of Col-0 and *ndx1-4* nuclei. Root tips of 10-day old seedlings (Col-0 and *ndx1-4*) were stained with DAPI and analyzed by confocal laser scanning microscopy (CLSM). From each sample, 10 optical stacks were recorded, which were merged by z-project (max intensity). Representative images are shown on the left. Quantification of nuclear sizes (Ferret diameter) is shown on the right. Statistically significant difference is indicated (***p*<0.0001, Mann-Whitney U test).

Relevant MS text: “We propose that reduced intracentromeric interactions in the absence of NDX results in chromatin decompaction at pericentromeric regions (where NDX is enriched in wild-type plants), which strongly correlates with the location of *sRNA* expression changes and CHH/CHG

methylation changes observed in the ndx1-4 mutant. This model is reinforced by microscopic observations of Col-0 and ndx1-4 nuclei (Supplementary Fig. 15), which revealed a significant increase in the size of DAPI-stained chromatin in ndx1-4. (Although there may be several reasons for the increase in nuclear size, one of them is definitively chromatin relaxation.) Whether the above chromatin changes are directly or indirectly mediated by NDX remains unclear. Nevertheless, chromatin compactness analysis⁶⁶ suggests that NDX has a direct effect on the condensation state of hypomethylated CHH regions (Supplementary Fig. 16) as NDX-enriched hypo-CHHs show significantly higher Hi-C compactness than hypo CHHs not bound by NDX (in wild-type plants). This indicates that NDX binding at these sites can directly promote chromatin condensation, which is inversely changed in the ndx1-4 mutant due to the loss of NDX (see Hi-C data).

Finally, it should be noted that the chromatin binding of NDX does not scale with 3D chromatin compactness, i.e. NDX shows preferential enrichment in pericentromeric heterochromatin that is not due to the condensed state or high local DNA concentration of this chromatin type. To demonstrate this, we compared the Hi-C compactness of pericentromeric, Knob, and chromosome arm regions in terms of NDX binding (Supplementary Fig. 17). The analysis showed that the compactness of all three regions differ significantly such that the most condensed is the heterochromatic Knob region, followed by pericentromeres and chromosome arms. Pericentromeric and Knob regions appear to bind more NDX per unit length than chromosome arms, however, the Knob showed similar NDX enrichment as pericentromeric regions despite its greater compactness. This suggests that NDX binding is disproportionate to compactness. When comparing regions with similar compactness (by random sampling from the above regions), chromosome arms still showed significantly lower NDX enrichment despite their same condensation state (Supplementary Fig. 17c). It follows that NDX preferentially interacts with constitutive heterochromatin, regardless of local chromatin density.”

Minor comments:

Please indicate in the text and Methods whether the NDX fusion proteins are expressed from an endogenous promoter.

Now we indicate in the revised text (Results section) and in the Methods that the NDX fusion proteins are expressed from their endogenous promoter. We note that all plant material used in our study was obtained from the Caroline Dean’s lab who had thoroughly characterised and validated the constructs in their landmark paper (Sun Q *et al.* Science 2013).

Figure 3d: Color of lines does not correspond to legend.

Line colours have been adjusted to the figures.

Figure 3f: Could same colors be used as in d to ease understanding for the reader.

The colour of box plots has been modified accordingly.

Figure 4d: The sRNA6 peaks seem not to overlap with the position of the pericentromeric heterochromatin for chromosome 5?

Thank you for bringing this mistake to our attention. Centromere positions have now been aligned.

Does the increased KNOT interactions involve the NDX locus itself (with its T-DNA)?

This is a bright question that we have not thought about so far. The T-DNA insert inactivating the NDX locus indeed appears to be anchored to the KNOT structure in *ndx1-4* plants (Figure S13). This ectopic KNOT interaction is fully consistent with the recent results of Grob S & Grossniklaus U. (Genome Biol 2019) showing that transgene integration sites in T-DNA insertion mutants induce novel long-range KNOT interactions. The *ndx1-4*-specific T-DNA insert is no different in this respect from other T-DNA insertion sites and supports the proposed role of the KNOT to act as a transposon trap (Grob S *et al.* Mol Cell 2014). The revised text has been supplemented with the above piece of information and the figure below.

Figure S13. KNOT interactions in the absence of NDX function.

Diagonal: the five Arabidopsis chromosomes. Lower left triangle (below the diagonal): differential Hi-C matrix calculated from the ratio of *ndx1-4* (red) and Col-0 (blue) interaction frequencies in Juicer. Resolution: 250 kb. Upper right triangle (above the diagonal): scheme of KNOT interactions in the *ndx1-4* mutant. KNOT-associated regions (circles) and centromeres (squares) and are highlighted. The T-DNA insertion site inactivating the NDX locus (red line in chr4) also participates in forming an ectopic KNOT interaction.

REVIEWER COMMENTS

Reviewer #1 (Remarks to the Author):

In this re-review of the manuscript entitled "Nodulin homeobox (NDX) is required for heterochromatin homeostasis in Arabidopsis", the authors do an admirable job responding to my comments and adding data supporting data to the manuscript. In addition, I appreciate the accurate language the authors use to describe their results, such as a mild reactivation of TEs in the *ndx* mutant. The below requested changes describe only informatic analyses and small adjustments to the writing.

1. On line 69 the authors use the term "initiation phase of RdDM" and then describe Pol IV production of 24nt siRNAs. However, the initiation phase of RdDM uses Pol II transcripts (see work of Slotkin & McCue). Instead, in line 69 the more accurate phrasing would be to use the word "upstream siRNA-generating phase of RdDM...".
2. In the abstract and introduction, the authors use the term "het-siRNAs". These are 24 nt siRNAs generated from Pol IV. The authors should demonstrate that it is these Pol IV-derived / dependent siRNAs that are specifically altered in their *ndx* mutant. From the chromatin types identified, I believe these are, but the authors could overlap their *ndx* mutant analysis with available Pol IV-dependent siRNAs.
3. Related to points 1-2, Figure 6F is very informative. Please repeat this analysis breaking down the siRNAs by size into 21, 22, 23, 24 nt size classes. This will be very informative, as 21-22 nt 'easiRNAs' are associated with TE expression, while 24 nt siRNAs are usually generated by Pol IV and are het-siRNAs described above. Overall, the analysis of siRNA size needs to be improved and points 2-3 will alleviate these issues.
4. Around line 191, it is unclear how the photobleaching experiment adds to the result that NDX binding is to pericentromeric heterochromatin. Please add a sentence to explain how this microscopy result provides this information.
5. Line 281 needs revision. CMT2 sites may generate 21-24 nt siRNAs, but these siRNAs do not participate in CMT2-based DNA methylation.
6. I do not agree with the statement on line 268 that TE activation is not associated with PolyA mRNA production. This claim is not referenced by the authors and I generally do not believe it to be true. Autonomous TEs must generate proteins, so need to generate polyA mRNAs. In addition, mRNA-seq is a well-established method to analyze TE expression and activation.
7. In line 401, the word "required" may be too strong. CHG methylation can occur without NDX, so a word such as "influences" or "regulates" is better.
8. Line 594 the word "ecotypes" is used incorrectly. I believe the authors mean "genotypes" or "lines".
9. Lastly, a general note: There are gaps in the reference Arabidopsis genome, and most of these gaps are in gene-poor centromeric heterochromatin. These mainly are TE regions. Long-read genomes of Arabidopsis are available and papers describe what is missing from the reference genome (although these long-read genomes are not as well annotated as the reference genome). NDX binding or TE activation or any deep sequencing experiment that generates reads from these TE located in gaps will result in those reads mapping to other TEs, some of which may be out on the chromosome arms. So the pericentromeric localization of NDX that is described in this manuscript is likely an underestimate of true pericentromeric localization, because of the DNA sequence that is missing from the genome in these regions.

Reviewer #2 (Remarks to the Author):

The new version of the manuscript is improved by new data including small RNA and mRNA-seq and additional Hi-C analysis.

This reviewer is still a bit concerned by the localization of the NDX-GFP protein in nuclei relative to the ChIP-seq datasets. While I agree with the authors that the overlap between NDX-GFP and Flag-NDX

ChIP-seq data is clear and convincing, the preferential pericentromeric localization of NDX (Figure 1d) along the chromosome is expected to be reflected by a clear preferential staining of the chromocenters only and not nearly the complete chromatin as in Sun et al, 2013 and Figure 2a, therefore the suggestion to image NDX-GFP in other nuclei with clear chromocenter structures. As the authors could not reveal GFP staining in leaf cells, immunofluorescence staining with anti-GFP antibodies on paraformaldehyde-fixed, isolated nuclei from leaves or those tissues shown to express NDX (with the GUS construct) would be expected to be successful (FLAG antibodies can indeed be poorly adapted for IF).

As the response to the authors comments in the rebuttal letter regarding the relationship between rDNA copies and NDX localization in the nucleolus: Indeed, most of the 45S rDNA units are repressed, but most accumulate in form of chromocenters next to the nucleolus and not inside even though small rDNA foci within the nucleolus do seem to exist as well. Again root nuclei are quite peculiar for this feature, but the way the authors have now formulated the text in the manuscript is sufficient as it leaves the role of NDX in the nucleolus open for further studies.

The data on the overlap between NUP1 enriched domains and NDX signals are very interesting. Could the authors further subdivide the data into peaks within chromocenters/pericentromeric regions and chromosome arms (Bi et al: only 10%–20% of the regions on the chromosome arms are anchored at the nuclear periphery)? This reviewer was not necessarily suggesting that NDX should be enriched at lamina like domains or nuclear periphery, confusion likely arose due to use of the term "peripheral heterochromatin" together with the NDX-GFP staining that labels nearly all chromatin in the root cell nuclei (Figure 2a). I suggest to clearly distinguish between pericentromeric heterochromatin that is in general an essential part of the chromocenter structures and peripheral (likely repressed) (eu)chromatin regions belonging to the chromosome arms throughout the text.

Concerning the heterochromatin organization: This reviewer still believes that FISH is a good complementary technique to visualize specifically what is happening with the pericentromeric repeats and it can easily reveal enhanced or reduced clustering, but it can be time consuming to put into place if this technique is not available in the laboratory. Given the amount of data in this manuscript, this is maybe not necessary however this reviewer is not convinced about the usefulness of the new Figure S15. The authors have observed "... a significant increase in the size of DAPI-stained chromatin in ndx1-4 nuclei." Do they refer to chromocenters or the whole nucleus (which is DAPI stained)? From differences in nuclear diameter, one cannot deduce "global relaxation of heterochromatin". Given the images shown here, it clearly looks that loss of NDX has a major impact on nuclear and chromatin organization that would definitely warrant further characterization by imaging approaches (many nuclei, many different plants, preferentially with the heterochromatin marker), but this is maybe out of scope of this paper here and I feel that Figure S15 is too preliminary to be included in the manuscript.

Reviewer #1 (Remarks to the Author):

In this re-review of the manuscript entitled “Nodulin homeobox (NDX) is required for heterochromatin homeostasis in Arabidopsis”, the authors do an admirable job responding to my comments and adding data supporting data to the manuscript. In addition, I appreciate the accurate language the authors use to describe their results, such as a mild reactivation of TEs in the *ndx* mutant. The below requested changes describe only informatic analyses and small adjustments to the writing.

We would like to thank the Reviewer for his critical evaluation of our revised manuscript and for suggestions and comments that have contributed to clarify/improve data and advance the story.

1. On line 69 the authors use the term “initiation phase of RdDM” and then describe Pol IV production of 24nt siRNAs. However, the initiation phase of RdDM uses Pol II transcripts (see work of Slotkin & McCue). Instead, in line 69 the more accurate phrasing would be to use the word “upstream siRNA-generating phase of RdDM...”.

According to Slotkin’s paper (<https://pubmed.ncbi.nlm.nih.gov/27808230/>), the terms “initiation phase” and “effector phase” of RdDM have been replaced by “upstream” and “downstream” phases of RdDM.

2. In the abstract and introduction, the authors use the term “het-siRNAs”. These are 24 nt siRNAs generated from Pol IV. The authors should demonstrate that it is these Pol IV-derived / dependent siRNAs that are specifically altered in their *ndx* mutant. From the chromatin types identified, I believe these are, but the authors could overlap their *ndx* mutant analysis with available Pol IV-dependent siRNAs.

In the *ndx1-4* mutant, ~85% of differentially expressed sRNAs belong to sRNA classes 4-9 and thus depend on Pol IV activity (Hardcastle 2018). More specifically, differential sRNA expression in the *ndx1-4* mutant significantly shifts toward sRNA classes 4 and 5 such that their proportion doubles compared to non-differential ones. (We note that sRNA classes 4,5 are typically Pol IV and Pol V dependent, while sRNA classes 6,8,9 depend on Pol IV only (Hardcastle 2018)). These associations demonstrate that Pol IV-derived / dependent siRNAs are specifically altered in the *ndx1-4* mutant. This analysis was added to the manuscript (Supplementary Fig. 5) and description and discussion to the text.

3. Related to points 1-2, Figure 6F is very informative. Please repeat this analysis breaking down the siRNAs by size into 21, 22, 23, 24 nt size classes. This will be very informative, as 21-22 nt 'easiRNAs' are associated with TE expression, while 24 nt siRNAs are usually generated by Pol IV and are het-siRNAs described above. Overall, the analysis of siRNA size needs to be improved and points 2-3 will alleviate these issues.

We have performed the analysis, which show that 24 nt het-siRNAs are specifically upregulated in the *ndx1-4* mutant at both RdDM TEs and CMT2 TEs. This result further reinforces the link between NDX and Pol IV-dependent sRNA expression. The data are presented in Figure 6g (see below).

Length normalized sRNA-seq read count over TEs (RdDM and CMT2). 21 nt, 22 nt, 23 nt, 24 nt sRNA reads were analyzed separately. In the 24 nt category, there is a statistically significant difference between the expression status of *ndx1-4* and Col-0 samples (RdDM TEs: $p < 2 * 10^{-16}$; CMT2 TE: $p = 0.014$). (We note that RdDM TEs vs. CMT2 TEs differ significantly in each category (not indicated), as expected from the biological nature of RdDM.)

4. Around line 191, it is unclear how the photobleaching experiment adds to the result that NDX binding is to pericentromeric heterochromatin. Please add a sentence to explain how this microscopy result provides this information.

The FRAP experiment does not directly contribute to the result that NDX binding occurs at pericentromeric heterochromatin. We apologize that this wasn't clearly conveyed. Our most important statement regarding the NDX photobleaching experiment is that NDX recapitulates the very slow kinetics and tight chromosome binding characteristics of core histones (H3/H4) measured by Kimura *et al.* (J Cell Biol 2001) in a similar FRAP setting. The revised text is modified as follows: "Taken together, the above quantitative microscopic data obtained in living cells are consistent with our ChIP-seq results and extend them to different spatial resolutions and timescales, indicating that NDX is a chromatin-binding factor that is stably incorporated into chromosomes."

5. Line 281 needs revision. CMT2 sites may generate 21-24 nt siRNAs, but these siRNAs do not participate in CMT2-based DNA methylation.

The Reviewer is right and we have corrected the mistake according to the Hardcastle paper (arguing that "...The loss of methylation in the *ago4* and *dcl2/3/4* mutants in the CHG and CHH contexts demonstrates the dependence of methylation in the Pol V-dependent locus classes (4,5,7) on the RdDM pathway. Conversely, CHG and CHH context methylation in the Pol V-independent locus classes (6,8,9)

is clearly driven predominantly by the CMT3 and CMT2 pathways respectively and is largely independent of the RdDM pathway.”) The revised text is as follows: “Utilizing this sRNA database, NDX binding sites showed a particularly strong colocalization with class 6-9 sRNA loci (Fig. 5a-b) that code for centromeric and pericentromeric 21-24 nt siRNAs participating in CHH/CHG methylation via the CMT2/3 pathway (classes 6,8,9) or the RdDM pathway (class 7).”

6. I do not agree with the statement on line 268 that TE activation is not associated with PolyA mRNA production. This claim is not referenced by the authors and I generally do not believe it to be true. Autonomous TEs must generate proteins, so need to generate polyA mRNAs. In addition, mRNA-seq is a well-established method to analyze TE expression and activation.

Indeed, most autonomous TEs harbor polyadenylation signals to promote their transcription, however, such cis-regulating sequences may disappear in truncated TEs (10.3390/ biology11040488). We removed the following part of from the original sentence in line 368: “most TEs lack a poly(A) tail”, while the first part was retained („This may be an underestimate since mRNA-seq captures only a portion of active TEs due to poly(A) selection”) and supplemented with an appropriate reference (<https://doi.org/10.1186/s13100-021-00251-1>).

The cited work analyzed several publicly available RNA-seq datasets comparing two common methods for NGS library preparation (poly-A RNA selection vs. ribosomal RNA depletion) to get a real estimate of TE expression. The analysis revealed a significant decrease in the percentage of reads mapping to TEs when using the polyA selection method and the authors argued that the use of poly-A mRNA-Seq libraries might not be ideal to assess TE expression. However, it was also mentioned the polyA selection is still capable of retaining ~70% of all TEs.

7. In line 401, the word “required” may be too strong. CHG methylation can occur without NDX, so a word such as “influences” or “regulates” is better.

The sentence has been rephrased: “These associations suggest that NDX influences transposon CHH/CHG methylation in pericentric heterochromatin regions.”

8. Line 594 the word “ecotypes” is used incorrectly. I believe the authors mean “genotypes” or “lines”.

Corrected as follows: “The following *Arabidopsis thaliana* genotypes were used in our study:”

9. Lastly, a general note: There are gaps in the reference Arabidopsis genome, and most of these gaps are in gene-poor centromeric heterochromatin. These mainly are TE regions. Long-read genomes of Arabidopsis are available and papers describe what is missing from the reference genome (although these long-read genomes are not as well annotated as the reference genome). NDX binding or TE activation or any deep sequencing experiment that generates reads from these TE located in gaps will result in those reads mapping to other TEs, some of which may be out on the chromosome arms. So the pericentromeric localization of NDX that is described in this manuscript is likely an underestimate of true pericentromeric localization, because of the DNA sequence that is missing from the genome in these regions.

We deeply agree with the Reviewer's comment. Third-generation sequencing approaches are expected to increase the accuracy of TE annotations within repetitive regions (e.g.

<https://doi.org/10.1016/j.molp.2022.05.0>), upon which the precise topography NDX binding sites could be mapped.

Reviewer #2 (Remarks to the Author):

The new version of the manuscript is improved by new data including small RNA and mRNA-seq and additional Hi-C analysis.

We highly appreciate the Reviewer's work in evaluating our revised manuscript and his critical remarks, which were relevant and helpful to clarify/extend and better present our data.

This reviewer is still a bit concerned by the localization of the NDX-GFP protein in nuclei relative to the ChIP-seq datasets. While I agree with the authors that the overlap between NDX-GFP and Flag-NDX ChIP-seq data is clear and convincing, the preferential pericentromeric localization of NDX (Figure 1d) along the chromosome is expected to be reflected by a clear preferential staining of the chromocenters only and not nearly the complete chromatin as in Sun et al, 2013 and Figure 2a, therefore the suggestion to image NDX-GFP in other nuclei with clear chromocenter structures. As the authors could not reveal GFP staining in leaf cells, immunofluorescence staining with anti-GFP antibodies on paraformaldehyde-fixed, isolated nuclei from leaflets or those tissues shown to express NDX (with the GUS construct) would be expected to be successful (FLAG antibodies can indeed be poorly adapted for IF).

We have performed an additional microscopic analysis on etiolated NDX-GFP seedlings that is expected to reduce chlorophyll content. In leaf cells, we still observed some autofluorescence that prevented proper assessment of intranuclear NDX-GFP signal distribution (hence we omit these data from the manuscript), however, NDX was clearly located in nuclear foci.

Characteristic microscopic distribution of the NDX-GFP signal in *Arabidopsis* leaf cell nuclei. Seedlings were etiolated to reduce chlorophyll content.

In root tips, we could identify cells with a typical chromocenter structure (representing only a minority fraction) that showed strong enrichment for the NDX-GFP signal. Since chromocenters were not readily visible in most root cells, the dominant staining pattern remained the same as before (i.e. peripheral NDX localization with nucleolar signal in about 20% of nuclei). Both type of microscopic staining pattern of NDX nuclei was incorporated into the revised manuscript as Figure 1a/b and Supplementary Figure 2, and the text and discussion were amended accordingly.

Characteristic microscopic distribution of the NDX-GFP signal in *Arabidopsis* root cell nuclei. (a) Nuclear periphery and nucleolar enrichment, (b) chromocenter enrichment.

(The suggested anti-GFP antibodies did not work in our hands because of the same reason as the anti-Flag antibody (i.e. issues with sample permeabilization and harsh treatments.)

In addition, we would like to make a note on the data presented in Figure 1d that intends to show the density of flag-NDX peaks along five *Arabidopsis* chromosomes. Density equals the number of ChIP peaks falling to 1 Mb ranges (i.e. peak number/Mb), which is significantly enriched in pericentromeric/centromeric regions compared to chromosome arms (16 peaks/Mb in arms vs. 84 peaks/Mb in pericentromeres; relative enrichment in pericentric heterochromatin being 5,25-fold). However, the absolute number of NDX peaks in chromosome arms involves 81,28% of NDX total peaks, which is fairly consistent with the observed intranuclear staining pattern of NDX-GFP in single cells. Nevertheless, the fraction of NDX molecules that are not bound to chromatin are expected to cause some differences between microscopic and genomic analyzes, since ChIP only detects the chromatin-bound fraction while microscopy detects both the unbound and chromatin-bound fractions.

As the response to the authors comments in the rebuttal letter regarding the relationship between rDNA copies and NDX localization in the nucleolus: Indeed, most of the 45S rDNA units are repressed, but most accumulate in form of chromocenters next to the nucleolus and not inside even though small rDNA foci within the nucleolus do seem to exist as well. Again root nuclei are quite peculiar for this feature, but the way the authors have now formulated the text in the manuscript is sufficient as it leaves the role of NDX in the nucleolus open for further studies.

We appreciate the Reviewer's comment on the appropriate wording of the text regarding the nucleolar association of NDX.

The data on the overlap between NUP1 enriched domains and NDX signals are very interesting. Could the authors further subdivide the data into peaks within chromocenters/pericentromeric regions and chromosome arms (Bi et al: only 10%–20% of the regions on the chromosome arms are anchored at the nuclear periphery)? This reviewer was not necessarily suggesting that NDX should be enriched at lamina like domains or nuclear periphery, confusion likely arose due to use of the term "peripheral heterochromatin" together with the NDX-GFP staining that labels nearly all chromatin in the root cell nuclei (Figure 2a). I suggest to clearly distinguish between pericentromeric heterochromatin that is in general an essential part of the chromocenter structures and peripheral (likely repressed) (eu)chromatin regions belonging to the chromosome arms throughout the text.

As suggested, we classified NUP1 binding sites as pericentromeric and arm-associated, which revealed a preferential colocalization between NUP1 and NDX at pericentromeres but significantly less along chromosome arms.

Overlap of NDX and NUCLEOPROIN1 (NUP1) binding sites marking peripheral heterochromatin. In pericentromeric regions, 52.4% of NUP1 peaks colocalize with NDX, which is reduced to 27.1% in chromosome arms (prop.test, *p<0.0001). Flag-NDX peaks were extended from the peak summits to 500 bp (each peak was 500 bp). Pericentromeres were defined as CEN positions +/- 2.5 Mb.

In addition, pericentromeric heterochromatin and peripheral chromatin are now better distinguished in the revised text.

Concerning the heterochromatin organization: This reviewer still believes that FISH is a good complementary technique to visualize specifically what is happening with the pericentromeric repeats and it can easily reveal enhanced or reduced clustering, but it can be time consuming to put into place if this technique is not available in the laboratory. Given the amount of data in this manuscript, this is maybe not necessary however this reviewer is not convinced about the usefulness of the new Figure S15. The authors have observed "... a significant increase in the size of DAPI-stained chromatin in ndx1-4 nuclei." Do they refer to chromocenters or the whole nucleus (which is DAPI stained)? From differences in nuclear diameter, one cannot deduce "global relaxation of heterochromatin". Given the images shown here, it clearly looks that loss of NDX has a major impact on nuclear and chromatin organization that would definitely warrant further characterization by imaging approaches (many nuclei, many different plants, preferentially with the heterochromatin marker), but this is maybe out of scope of this paper here and I feel that Figure S15 is too preliminary to be included in the manuscript.

Supplementary Figure 15 has been removed from the revised manuscript.

REVIEWERS' COMMENTS

Reviewer #2 (Remarks to the Author):

The authors have addressed all my comments in a satisfactory manner. I recommend the manuscript for publication in its current form.